# Wearable Nano-Based Gas Sensors for Environmental Monitoring and Encountered Challenges in Optimization

**DOI:** 10.3390/s23208648

**Published:** 2023-10-23

**Authors:** Sara Hooshmand, Panagiotis Kassanos, Meysam Keshavarz, Pelin Duru, Cemre Irmak Kayalan, İzzet Kale, Mustafa Kemal Bayazit

**Affiliations:** 1Sabanci University Nanotechnology Research and Application Center (SUNUM), Tuzla, Istanbul 34956, Turkey; 2The Hamlyn Centre, Institute of Global Health Innovation, Imperial College London, South Kensington, London SW7 2AZ, UK; p_kassanos@hotmail.com; 3Department of Electrical and Electronic Engineering, Imperial College London, South Kensington, London SW7 2AZ, UK; 4Faculty of Engineering and Natural Science, Sabanci University, Istanbul 34956, Turkey; pelinduru@sabanciuniv.edu (P.D.); ikayalan@sabanciuniv.edu (C.I.K.); 5Applied DSP and VLSI Research Group, Department of Computer Science and Engineering, University of Westminster, London W1W 6UW, UK; kalei@westminster.ac.uk

**Keywords:** wearable nanosensors, gaseous pollutants, gas-sensing nanomaterials, environmental detection, sensor performance optimization

## Abstract

With a rising emphasis on public safety and quality of life, there is an urgent need to ensure optimal air quality, both indoors and outdoors. Detecting toxic gaseous compounds plays a pivotal role in shaping our sustainable future. This review aims to elucidate the advancements in smart wearable (nano)sensors for monitoring harmful gaseous pollutants, such as ammonia (NH_3_), nitric oxide (NO), nitrous oxide (N_2_O), nitrogen dioxide (NO_2_), carbon monoxide (CO), carbon dioxide (CO_2_), hydrogen sulfide (H_2_S), sulfur dioxide (SO_2_), ozone (O_3_), hydrocarbons (C_x_H_y_), and hydrogen fluoride (HF). Differentiating this review from its predecessors, we shed light on the challenges faced in enhancing sensor performance and offer a deep dive into the evolution of sensing materials, wearable substrates, electrodes, and types of sensors. Noteworthy materials for robust detection systems encompass 2D nanostructures, carbon nanomaterials, conducting polymers, nanohybrids, and metal oxide semiconductors. A dedicated section dissects the significance of circuit integration, miniaturization, real-time sensing, repeatability, reusability, power efficiency, gas-sensitive material deposition, selectivity, sensitivity, stability, and response/recovery time, pinpointing gaps in the current knowledge and offering avenues for further research. To conclude, we provide insights and suggestions for the prospective trajectory of smart wearable nanosensors in addressing the extant challenges.

## 1. Introduction

Due to rapid industrialization and urbanization, the world is facing the problem of severe air pollution. This is witnessed by the increasingly large amounts of toxic and pollutant gases released into the environment, one of the negative aspects of the industrial revolution. The demand to improve public safety and quality of life due to the development of different industry sectors and the issues related to air pollution are significantly growing. These pollutants usually contain toxic gases such as CO, SO_2_, NO_2_, and H_2_S that can harm humans, their surrounding atmosphere, and the environment through ozone depletion, acid rain, and the greenhouse effect [1]. The harmful effects of such gases on human health are so severe that industrial workers exposed to such gases need a safety gas alert system capable of detecting and quantifying harmful chemicals and gases. Detecting dangerous gases is becoming increasingly important for indoor and outdoor air quality monitoring in houses, office buildings and factories, public safety, and mining and energy industries [2].

Gas sensors are also widely used to measure greenhouse gases. Furthermore, studies are being undertaken to detect some of these gases in breath, including nitric oxide (NO), ethane, and NH_3_, for diagnostic applications. They also act as biomarkers for different pathologies [3,4]. The detection of volatile organic compounds (VOCs) is thus of significant importance [5]. There is a tremendous need for simple and portable sensors that can measure gases sensitively, selectively, and easily in real time to analyze gases efficiently. As a result, simple, on-site, real-time monitoring of air quality and pollutant content is vital for protecting human health from poisonous and dangerous gases. Meanwhile, there is also a need for such devices to safeguard human health in various situations, including food protection and public security. As a result, the demand for developing small-sized, lightweight, and portable sensing devices has drastically increased. The global gas sensors market is projected to reach USD 2.1 billion by 2027, growing at a compound annual growth rate (CAGR) of 8.9% [6]. For this purpose, numerous sensing techniques, such as calorimetric, thermoelectric, optical, and chromatographic methods, have been applied [7]. However, most of these techniques have disadvantages, such as relatively high cost, low sensitivity and selectivity, poor design, and a need for additional equipment; some even lack portability [7].

In terms of their mechanical properties, sensors can generally be categorized as either rigid, flexible, or stretchable depending on the properties of the main substrate of the sensor. All three categories can be used in wearable applications, depending on the requirements and specifications (e.g., lifetime, duration of use, and mechanical perturbation that can be tolerated) of the application and the body area on which they are intended to be worn. Flexible and stretchable sensors can be composed of several rigid parts interconnected with flexible/stretchable electrical interconnects in an island-bridge approach. Rigid substrates and devices can be thinned to make them flexible to some extent, while use of intrinsically rigid devices can be worn on body areas not susceptible to mechanical deformations (bending) such as the lower back, the head, and areas away from joints. In addition, making a device sufficiently small compared to the scale of mechanical deformation and motion can render a rigid device wearable and insensitive to the motion of the body. For example, see Section 4.4 on microelectronics approaches. When device scales are larger than the body curvature, and not much mechanical deformation due to body motion is expected, flexible devices are needed. However, when large deformations are expected as in joints and during exercise routines and sports, stretchability is required. In addition, large rigid devices, due to the significant difference in their mechanical properties with skin and the underlying tissue, will lead to injuries for the wearer, while the rigid device will also become damaged and will malfunction. Consequently, when the overall sensing system size cannot be sufficiently small, the development of low-cost, flexible, or stretchable wearable and high-performance flexible gas sensors is necessary. Wearable sensors can also be integrated into clothing or accessories, in addition to being worn directly on the skin, depending on the needs of the intended use. In the past ten years, major developments have occurred in the wearable sensor research field [8]. The market for wearable technology has experienced exponential growth in recent years. Wearable sensors allow the non-invasive monitoring of human activity and health. This has given access to vast amounts of information and new knowledge. The most popular wearable devices are ear-worn devices, smartwatches, and wristbands. These devices can easily connect to a smartphone via Bluetooth or a wireless network, enabling real-time data transmission, cloud computing, and big data analytics for large sensor networks and data, and remote monitoring [9]. With computational capabilities, density and integration, and microelectronic components’ power efficiency, improving on-node real-time information processing is becoming a reality. Wearable technologies can potentially improve occupational health and safety management, making real-time health and environmental data available to users, inspectors, and personnel responding to emergencies. This information can pertain to air pollution, indoor air quality, exposure to aerosols, or the identification of biohazardous agents in the workplace, allowing informed decisions to be made [10]. Recent advances in smart wearable electronics and sensors are allowing more precise, less noisy, pervasive, and non-intrusive environmental data collection that can consistently monitor human physiology and environmental factors and are contributing to the rapid development and evolution of the Internet of Things (IoTs) that is currently underway [11].

The realm of gas sensing has been extensively explored in several reviews [12,13,14,15]. Yet, a comprehensive examination of wearable gas-sensing devices, along with their associated challenges, remains conspicuously absent, particularly from a holistic standpoint. This gap is noteworthy, especially as the future trajectory is evidently leaning towards the integration of gas sensing with smart wearable personalized technologies. Notably, there has been a review on the recent advancements in flexible room-temperature gas sensors grounded on metal oxide semiconductors [16]. Furthermore, cutting-edge methodologies solely centered on metal-oxide-based heterostructures for room temperature gas sensors have been discussed [17]. Additionally, a detailed analysis has been undertaken regarding gas sensors and the dynamics that impact their sensing mechanisms, with a special emphasis on metal oxide semiconductors [18]. The uniqueness of our paper lies in its broad perspective, bridging the existing literature gap and highlighting the novel integration of wearable technology in the gas-sensing domain.

This paper presents a cutting-edge review of advancements made since 2020 in the realm of smart wearable (nano)sensors tailored for the environmental monitoring of gaseous pollutants. Notably, this comprehensive survey underscores the pioneering nature of these wearables in tackling key challenges associated with their development and commercialization. These challenges encompass sensor sensitivity, selectivity, wearability, integration, miniaturization, cost implications, power consumption, and the quest for high-performance, biocompatible power supplies. Such power sources must closely align with the sensing device in terms of weight, flexibility, durability, and other vital attributes [19].

This review is systematically structured into distinct sections for ease of understanding:Gas-Sensing Materials: This section delves into a variety of materials, including 2D nanostructures, carbon nanomaterials, conducting polymers (CPs), nanohybrids, and metal oxide semiconductors (MOSs).Wearable Substrates and Electrodes: Here, different substrates such as paper-based (PB) mediums, polymeric materials, textiles, and stretchable electronics have been explored that can serve as pivotal components in wearable sensors.Sensor Types: This section covers an in-depth analysis of diverse sensors—from colorimetric, chem-optical, and electrochemical, to transistors and chemiresistors. Emphasis is laid on the critical role these sensors play in monitoring environmental gaseous pollutants, including but not limited to ammonia (NH_3_), nitric oxide (NO), nitrous oxide (N_2_O), nitrogen dioxide (NO_2_), carbon monoxide (CO), CO_2_, H_2_S, sulfur dioxide (SO_2_), ozone (O_3_), hydrocarbons (C_x_H_y_), and hydrogen fluoride (HF). The significance of deploying proficient detection methods for these pollutants is accentuated.Challenges in Gas Sensor Optimization: Distinct from existing literature, this section offers a meticulous discussion on challenges that have remained largely unexplored. Topics such as circuit integration, real-time sensing, repeatability, power consumption, gas-sensitive material deposition, and stability, among others, are examined in detail.

The uniqueness of this review paper lies in its holistic approach, addressing numerous facets of gas sensor technology in a singular, comprehensive document. We envision that this will significantly benefit researchers and industry experts, steering them towards pioneering the next generation of gas sensors. In the following sections, various materials, device architectures, and sensing approaches are reviewed and are discussed and presented with regard to gas-sensing parameters that mainly include sensor response, sensitivity, response/recovery times, selectivity, detection limit, and stability.

## 2. Sensing Materials

Various sensing materials, including 2D nanostructures, carbon nanomaterials, CPs, nanohybrids, and MOSs, have been reported to allow a reliable detection system for the advanced monitoring of gaseous pollutants. These are discussed in the following sections.

### 2.1. 2D Nanostructures

The latest developments in the use of two-dimensional (2D) nanostructures for sensors for monitoring gaseous pollutants are discussed in this section. These 2D nanostructures have gained broad interest owing to their superior electronic and material properties and high photo-response. Numerous studies have shown the exceptional physical/chemical properties of 2D nanostructures and their wide range of applications in healthcare, defence, environmental monitoring, and food safety. These 2D nanostructures, such as graphene, graphitic carbon nitride (g-C_3_N_4_), poly-types of transition-metal dichalcogenides (TMDC or TMD) materials (for example, MoS_2_ and WS_2_), and oxides of group B elements such as MoO_3_, WO_3_, and MnO_2_, can have exceptional electrical, chemical, physical, and optical properties, with high electrical conductivity, electron mobility, and a large surface area. These are attributes that are highly beneficial in sensing applications [20], allowing the use of these materials in photoelectrochemical, field-effect transistor (FET), fluorescence, surface-enhanced Raman scattering (SERS), and colorimetric biosensors [21].

Graphene, hexagonal boron nitride (h-BN), and metal dichalcogenides, MX_2_ (M, transition metal; X, chalcogen atom), have dominated the spotlight. For instance, the application of 2D titanium carbide MXenes for detecting various gases pollutants is proposed by Simonenko et al. to take advantage of lowering response and recovery times, increasing sensitivity and stability, and improving sensitive functionality in humid atmospheres [22]. The latest advancements in device integration, microfabrication, and material and device processing make it possible to integrate materials such as Ti_3_C_2_T_x_ MXene-loaded 3D-substrates into lab-on-a-chip systems, which can analyze multiple analytes in the gas form [23]. Additionally, layered 2D nanomaterials, so-called layered metal disulfides (LMDs) such as SnSe_2_, an anisotropic binary layered material with a hexagonal structure, have been used in the gassensing field as they offer unique electronic characteristics, high sensitivity, rapid response, good selectivity, and low operating temperature. Chemiresistive SnSe_2_ sensors have demonstrated a significant response of 60% to 1 ppm NO_2_ at room temperature (RT) [24]. Despite these attributions that make them an ideal candidate for wearable gas sensors compared to ZnO, their performance is limited in harsh environmental conditions [25,26,27].

### 2.2. Carbon Nanomaterials

Developing sensitive nanomaterials with a rapid response to detect and quantify low concentrations of gaseous pollutants is a big challenge. Carbon nanomaterials have shown great potential to be efficient sensing materials due to their exceptional characteristics like great sensitivity, high solid–gas interactions, and a superior surface-to-volume ratio [28]. Optimizing key factors such as the physical structure and surface architecture of the deposited carbon nanomaterials on the electrodes/substrates (e.g., porosity, thickness, and interface properties) should be deeply considered to achieve the best device performance [29]. On the other hand, two other key factors affect the sensitivity of carbon nanomaterial-based gas sensors: surface area and electrical conductivity. These factors can be improved through the chemical modification of the surface (e.g., surface functionalization with other nanoparticles, metal atoms, polymers, ionic liquids, enzymes, etc.) [28]. Carbon-based nanomaterials are generally classified based on their geometric structures into three types of fullerenes (spheres as in C_60_ (soccer ball) or ellipsoids as in C_70_ (rugby ball)), carbon nanotubes (CNTs, tube-shaped), and graphene (single sheets of carbon atoms) [30]. Gas sensors using such materials and their derivatives have been widely demonstrated [31].

#### 2.2.1. Fullerenes

With 60 π-electrons in their structure and high electron mobility, fullerenes have earned prominent notice in sensing applications due to their exceptional charge transport and separation, unique photophysical properties, and high electrical conductivity. They tend to improve polymeric sensors’ overall sensitivity and sensing performance for health care and environment monitoring [32]. Particularly in the gas-sensing area, the curvature effects of C60 make it a top candidate for the physical adsorption of small molecules such as CO_2_ via van der Waals interactions [33].

Some approaches have been verified to improve fullerenes’ surface reactivity and adsorption capacity. In a recent study, boron-functionalized C60 fullerene decorated with Sc was studied to adsorb CO_2_ efficiently. The boron atoms chemically doped with C60 (C_48_B_12_) make it a powerful electron acceptor to bond strongly with Sc atoms. The large energy barrier provided by this metal-incorporated system prevents aggregation even at critically high temperatures. Moreover, Sc atoms, the lightest transition metal atoms, offer a larger capacity for reversible CO_2_ adsorption at ambient temperature in the final form of Sc-decorated C48B12 due to polarization and charge transfer effects. Each decorated Sc atom is capable of capturing four CO_2_ molecules resulting in a 52% of gravimetric density. This system can also be employed efficiently to separate CO_2_ from its mixtures with N_2_, H_2_, and CH_4_ [33]. In the case of chemiresistive gas sensing, providing a chemically heterogeneous nanocomposite can affect the hole density of the fullerenes and, consequently, increase the gas adsorption. In another study, a chemiresistive sensing composite of brominated fullerene and single-walled carbon nanotubes (C_60_Br_24_/SWCNT) was proposed [34]. While SWCNTs bring about a conductive composite, Br atoms make it capable of bonding with H atoms of gas molecules through medium-strength halogen bonds. Different SWCNTs loadings (2, 5, 8, and 10 wt%) were tested. Based on results and DFT calculations, the best sensing performance was demonstrated by halogenated C_60_ composites mixed with SWCNT (10 wt%) compared to pristine fullerene composites (Figure 1). At RT, a trace amount (50 ppb) of H_2_S was selectively detected in the presence of CO or NO with a superb response of 1.75% [34].

#### 2.2.2. Carbon Nanotubes (CNTs)

CNTs, a family of 1D nanomaterials, have been widely used in gas-sensing applications due to their hollow core, high electrical conductivity, chemical inertness, and high surface area. CNTs perform like p-type semiconductors since oxidation states and defects are formed in their synthesis and purification steps. However, they have sites with a few defects in their nanoscale architecture, which can be a drawback [35]. The interactions between CNTs and the target gas are either van der Waals or donor–acceptor. Gas-adsorption-induced charge transfer has been verified as the primary sensing mechanism for CNT-based gas sensors. As CNTs bridge the sensor electrodes, each CNT performs as a conductive channel, capable of adsorbing the gas species. Different approaches have been described to form these CNT channels, including single CNT bridging, dielectrophoretic (DEP) assembly, and dispersed CNT network formation [36]. SWCNTs have been used in ultra-sensitive FETs to monitor ppm to ppb amounts of gases due to their capability to be applied in miniaturized devices under low temperatures. A remarkable device response of 10 ppm to 100 ppb was achieved for detecting NO_2_ gas using a SWCNT-FET platform (Figure 2). This superior low limit of detection (LOD) is due to the increase in the Schottky barrier modulation generated by the smart configuration of the sensor in which an individual semiconducting SWCNT is directly connected between the interdigitated source and drain electrodes. As it is the first and most sensitive unfunctionalized carbon-nanomaterial-based gas sensor operating at RT, it can offer valuable knowledge for constructing the next generation of CNT-based gas sensors [37].

Semiconducting CNTs have also been decorated with tin dioxide (SnO_2_) nanoparticles via a two-step dielectrophoretic assembly, first to form CNT channels between the comb-shaped electrodes, and then to attach SnO_2_ nanoparticles onto the CNT surfaces. This sensor could successfully detect trace amounts of NO_2_ gas (10 ppb) due to the p-n hetero-junctions between the CNTs and the SnO_2_ nanoparticles, which caused a sharp increase in the resistance [38].

A sol–gel-based composite film of multi-walled carbon nanotubes (MWCNTs) in an alumina (Al_2_O_3_) matrix has revealed a high gas-sensing response (7.3%) to CO_2_ concentrations from 450 ppm to 50 ppm. The change in resistance was considered an indicator for the gas measurements. Due to the charge transfer mechanism between MWCNTs and gas molecules, a considerable increase occurred in the resistance level upon CO_2_ exposure. Furthermore, UV light and heat were applied as different recovery modes, resulting in fast desorption times of CO_2_. This fast-response (~50 s) and stable sensor can be used in multiple gas-analyte-sensing applications as an economical and reliable method [39]. As another example of the application of functionalized SnO_2_/CNT-based gas-sensing materials, a highly sensitive copper-nanoparticle-coated SnO_2_/CNT composite film was used to detect H_2_S at RT. While CNTs are applied to transmit charge, the p-CuO/n-SnO_2_ interface causes a high resistance in the sensing materials by creating a charge-carrier depletion layer. Selectivity of the sensor in 40 ppm H_2_S with a faster maximum response of 19% was verified in the presence of the same gas amount of NH_3_ (4.1%), CO (0.2%), and SO_2_ (0.1%). In the presence of different concentrations of H_2_S, the sensor showed stable performance, outstanding repeatability, and a considerable decrease in resistivity. The sensor also revealed a fast response of 4 min for low concentrations of H_2_S (10 ppm) with a sensitivity of 4.41 and a recovery time of 10 min [40]. Combining single- (75%) and multi- (25%) walled CNTs has recently been employed to detect NH_3_ and NO_2_ using impedance spectroscopy. Functionalizing these CNTs with pyrene led to a more sensitive film to both gases than the bare forms. Sensing layers reveal different impedance responses upon exposure to different gases. In addition, since a wide range of frequencies is required for analyzing the signal responses of the films to the gases, it is possible not only to identify but also quantify the concentration levels of the gases [41].

#### 2.2.3. Graphene

Various production methods have been reported for graphene-based gas sensors over the last few years owing to graphene’s benefits, including superb optical transparency, conductivity, mechanical strength, fast charge transfer, and low surface area. However, the lack of defective sites in pure graphene and its reduced surface area due to simple aggregation may limit the recovery speed of the sensor and its stability in humid conditions. Thus, to achieve the next generation of commercialized graphene-based gas sensors, different ways of optimizing the use of graphene should be investigated. Graphene is considered the basic structure for other carbon nanomaterials, which can be stacked up to form graphite, wrapped up to produce fullerenes, and rolled to shape nanotubes [42]. Surface modification techniques have effectively enhanced the gas-sensing properties of graphene. In addition, FETs have also been proven to be valuable platforms for fabricating ultra-sensitive graphene-based gas sensors [43].

Hydrophobic polymer brush insertion was applied to graphene FETs as a surface treatment to rapidly detect gaseous pollutants of NO_2_, NH_3_, and CO_2_ [44]. This ultrathin surface modification layer boosted graphene-sensing performance for NO_2_ to 5.3 times more with a LOD of 4.8 ppb by covering the charged impurities on the SiO_2_/Si substrates. C and Si atoms belong to the same group with similar chemical properties, with only a larger atomic radius for Si atoms than carbon. Thus, by doping Si atoms into the C network of graphene, the atomic arrangement becomes more crowded, which leads to the modification of its chemical properties by changing the electronic structure of graphene. As the ideal performance of Si-doped graphene gas sensors for NO and NO_2_ was verified theoretically through ab initio DFT; experimental studies were also conducted to approve the whole concept. In another study, active sites of Si atoms were well-doped into the carbon network of graphene nanosheets in a high-temperature annealing process at 400 °C between graphene oxide (GO) and tetraethoxysilane (TEOS) [45]. The prepared sensing material revealed unique selectivity and sensitivity to NO_x_ with a response value of 21.5% (50 ppm NO_2_) and response and recovery time of 126 s and 378 s, respectively. In addition, these chemiresistive sensors allowed the detection of a wide NO_2_ gas concentration range from 300 ppm to 18 ppb [45]. Graphene has also been employed as the source electrode in fabricating a vertical graphene field effect transistor (VGr-FET) for the gas-phase sensing of ammonia, isoprene, oxygen, and water vapor [46]. Thin layers of C_60_ as the semiconductor film and Al as the drain electrode were coupled with graphene to make a graphene/C_60_/Al structure. Analyzing the I–V response curves of the heterojunction diode model, the energy barrier height between graphene and polycrystalline fullerene was determined. The lowest LOD of 86 ppb was achieved for NH_3_ compared to isoprene (420 ppb). This is due to the donor nature of the adsorbed NH_3_ on p-type graphene, which causes a positive gating effect in the VGr-FET. The authors recommended their facile economical sensor as a promising quick-response device for diagnostic applications in the early stages of severe lung cancer [46].

### 2.3. Conductive Polymers

Polymers are macromolecules with repeating structural units united by covalent connections. Although traditional aliphatic polymers are usually insulators, polymers can also be electrically conducting. The conjugated backbones of CPs are their primary building blocks. Due to the delocalization of electrons in a continuously π-overlapped orbital along the polymer backbone, CPs, which are also referred to as conjugated polymers, possess some of the most interesting electrical and optical properties. These properties are a result of the polymer backbone [47]. As a result of their outstanding chemical and environmental stability, biocompatibility, and unique optical and electrical properties, CPs have attracted considerable attention in a wide range of scientific domains, such as chemistry, biology, electronics, and materials science [48].

The attractive and distinctive optoelectronic features of CPs allow them to be exploited for numerous applications, including gas detection. Flexible gas sensors that are constructed out of CP-sensing materials have drawn a lot of attention due to their many advantageous properties. These properties include RT operation, tunable conductivity, flexibility, environmental stability, low cost, and various functionalization options [49]. A conjugated arrangement with alternating single and double bonds, or a conjugated system linked with atoms which generate orbitals for a continuing orbital overlap, appears to be an essential requirement for a polymer to conduct inherently. For polymers to be electronically conducting, charge carriers and the orbital system that enables these carriers to move freely are necessary, similar to the free electron movement associated with high conductivity in metals [50]. Since the 1980s, CPs that are solution-processable aromatic polymers such as poly(p-phenylene) (PPP), polypyrrole (Ppy), polyaniline (PANI), polythiophene (PT), poly(3,4-ethyelenedioxythiophene) (PEDOT), and their derivatives have been utilized as active layers [51]. There are several ways to modify CP conductivity to achieve desired results. Doping modifies a polymer’s structure chemically. Alternatively, CPs can be mixed with additives, including metals, semiconductors, acids, or surfactants. In a study, an NH_3_-sensing composite was prepared by coating polymer sulfonic-acid-doped PANI with bacterial cellulose (BC) nanofibers and co-doped with dodecylbenzene sulfonic acid (DBSA) and poly (2-acrylamido-2-methyl-1-propane sulfonic acid) (PAMPS) [52]. BC/PANI-DBSA/PAMPS was successfully synthesized using an in situ chemical oxidation polymerization technique. This synthesis method produced a profound structure of a nanoscale PANI layer connected to the BC backbone, as shown in Figure 3a. The homogenously coated three-dimensional network structure enabled faster gas adsorption/desorption and electrical signaling.

Furthermore, a macromolecular sulfonic acid, PAMPS, made closer connections between the crystal islands because of various interactions. Additionally, more sensing sites were created on the surface of the PANI, which enhanced the structural stability of the DBSA doping and strengthened the conduction signaling of the material. The produced chemiresistive sensor based on this macroscopic film had a simple device configuration. It displayed a sensing response (defined as the ratio of the device’s intrinsic resistance to that in the presence of the target gas) of 6.1 up to 100 ppm, high selectivity, and rapid response/recovery time of 10.2 s/8.6 s for 100 ppm with a LOD of 200 ppb to NH_3_ at RT. This easy and versatile method for building sensing materials on nanofiber templates has potential in NH_3_-exposed areas.

The effectiveness of a nanohybrid sensor based on PANI, silver oxide (Ag_2_O), and GO (PANI-Ag_2_O-GO) was evaluated for NO_2_ sensing [53]. The dispersed graphene and PANI/Ag_2_O solutions in deionized water were combined to make the composite sensor. The resulting solution was then dropped and dried over platinum microelectrodes attached to alumina ceramic chips. A computerized characterization instrument was used to determine the sensor’s capabilities (Figure 3b–d). A data-collecting system recorded the readings after introducing the target gas to the sensor via the gas cylinder. When subjected to 25 ppm NO_2_ at 100 °C, the results showed that the PANI-Ag_2_O-GO sensor (5.85) had a two-fold higher response than the PANI (2.5) and PANI-Ag_2_O (3.25) sensors. The charge transfers between the adsorbed NO_2_ gas molecules and the nanohybrid surface were attributed to the increased gas-sensing response of the nanohybrid sensor (PANI-Ag_2_O-GO). On the surface of PANI-Ag_2_O-GO, the adsorbed NO_2_ molecules decomposed into nitric oxide (NO) or dinitrogen oxide (N_2_O). The formation of a π–π conjugation system between PANI and Ag_2_O/GO aided the dissociation of NO_2_ molecules, resulting in a large electronic cloud and excellent electron transport channel. As a result, the proposed sensor can be used to detect NO_2_ gas at low temperatures [53].

A capacitive-type nanohybrid sensor for detecting CO_2_ gas was investigated and constructed using Ppy and copper phthalocyanine (CuPc) [54]. Initially, Ppy was made from its monomer using a single-step reaction of chemical oxidative polymerization. This reaction was carried out in hydrochloric acid (HCl) and used ammonium persulfate (APS). Then, an in situ chemical oxidative polymerization process produced the nanohybrid (Ppy-CuPc) at a temperature between 0–5 °C with or without the inclusion of a cationic surfactant called cetyl-trimethyl ammonium bromide (CTAB). Incorporating CTAB into the Ppy-CuPc mixture led to an interconnected network of nanofibers resulting in higher response and sensitivity than Ppy-CuPc nanohybrids and Ppy. The CTAB-based enhanced sensing performance of Ppy-CuPc nanohybrids was attributed to the porous structure of the sensing layer facilitating the diffusion of gas molecules, the high surface-to-volume ratio of the nano-fibrous structure of Ppy, the highly interconnected network of Ppy nanofibers, and the synergistic effects of the properties of individual components of Ppy and CuPc [54].

Another example is a Ppy-modified TiO_2_ deposited on Cu-interdigitated electrodes that was exploited as a chemiresistive CO gas sensor with high sensitivity and rapid response even at ambient temperature. Ppy/TiO_2_ was directly synthesized via an in situ chemical polymerization. The CO response measurements were carried out on active Ppy/TiO_2_ films by exposing them to CO gas of 1–320 ppm at RT. The sensor’s demonstrated response and recovery times were fast (36 and 38 s, respectively) and dependent on the film composition [55].

### 2.4. Nanohybrids

Detection of low traces of harmful toxic gases using nanohybrid sensors is increasing due to their low cost, feasibility, low maintenance, simple structures, and capability to sense various types of gases [56]. The conjugation of organic and inorganic nanomaterials has introduced a new class of hybrid nanomaterials [57]. In contrast to single-component nanomaterials, nanohybrids offer versatile chemical and physical functionalities that are advantageous in various fields [58]. Nanohybrid sensors benefit from increased conductivity, porosity, catalytic activity, and optical and electrical potential, with a low LOD for gases even at ppm concentrations [57,58,59].

Semiconductor-based gas sensors such as ZnO, SnO_2_, NiO, and CuO have attracted significant attention owing to their good sensitivity, cost-effectiveness, and capability to detect a wide range of toxic gases. However, semiconductor-based gas sensors suffer from low selectivity, surface defects, weak adsorption potential, and low anti-corrosive properties [60]. Nanohybrid-based sensors offer a new paradigm to attain selectivity with enhanced sensitivity. Polymers such as PT, polyacetylene (PA), PEDOT, poly (phenylene vinylene) (PPV), PANI, and Ppy are among the CPs with potential use in gas sensing as their conductivity changes upon exposure to gas molecules [61]. Utilizing such polymers has the benefits of high functionality, low cost, excellent stability, rapid response time, remarkable recovery, facile synthesis, and high surface area. However, poor sensitivity, selectivity, response, and recovery restrict their gas-sensing capabilities. Incorporating semiconductors and inorganic nanomaterials in polymer-based nanohybrid sensors, such as metal oxides and metal oxide semiconductors, enhances their sensitivity, stability, and response by increasing the surface area and allowing the detection of a wide range of gases [62,63]. Generally, immiscible polymers are without specific interactions; therefore, developing miscible composite polymers is of great concern. For instance, polymeric hybrids may possess weak bonding interactions like hydrogen bonds and van der Waals forces between organic and inorganic parts.

In contrast, few organic/inorganic hybrids contain strong chemical bonds at the interface. These interactions depend on nanoparticle size, dispersion form, shape, and size distribution. To synthesize polymeric/inorganic nanohybrids sensors for detecting harmful gases such as hydrogen (H_2_), NH_3_, H_2_S, CO_2_, CO, NO_2_, and liquefied petroleum gas (LPG), different methods such as physical, chemical, and electrochemical techniques have been used [64,65,66].

### 2.5. Metal Oxide Semiconductors

MOS materials have been extensively used in gas sensing due to their simple structure, simple and established manufacturing process, compatibility with microelectronics manufacturing, low-cost, excellent response, and long-term stability [67]. As a result, MOS materials and their use in gas sensing have been extensively studied over several decades. The use of metal oxide gas sensors goes back to 1962, and it was realised that the resistance of such films changes in the presence of CO_2_, toluene, and propane gases at a specific temperature [68]. This chemiresistive characteristic of semiconductor materials has made them ideal candidates for gas sensing. Initially, MOS gas sensors suffered from high power consumption and complicated fabrication processes, which hindered their application. However, in 1986, gas sensors based on SnO_2_ were commercialized [69,70].

The adsorption/desorption of gas molecules onto the surface of semiconductor-based sensors leads to changes in electrical parameters, which determine the presence of analyte molecules in the environment. Subsequently, the analyte’s composition and chemical structure and the semiconductor materials’ morphology significantly impact the operation of chemical gas sensors. Therefore, the right sensing material choice is critical in detecting gaseous compounds in the environment [71,72,73,74]. The use of MOSs to detect SO_2_ gas in wearable sensors is increasing. MOSs such as ZnO, SnO_2_, and In_2_O_3_ are relatively inexpensive, highly sensitive, and easy to fabricate [75]. Nanostructured ZnO-based MOS gas sensors, representative of n-type semiconductors, offer certain advantages including non-toxicity, cost-efficiency, rapid response, and stability. These attributes make them notable candidates in the realm of gas sensing, especially for wearable devices. The 1D ZnO nanostructure-based sensors have demonstrated the ability to detect gases like NO_2_, NH_3_, CH_4_, H_2_, and C_3_H_8_, though the sensitivity largely depends on specific operating temperatures [76,77]. However, it is crucial to highlight that the choice of the best material is influenced by application-specific requirements. Comparing the performance of ZnO with other metal oxide materials under standardized conditions would be imperative for a comprehensive analysis. In addition to ZnO-based 1D nanomaterials, advancements in gas sensing also focus on other one-dimensional semiconductors, such as Pd-doped 1D tungsten oxide nanowires [78]. Overall, metal oxide nanowires have been gaining attention for their stable chemical properties and considerable surface-to-volume ratio. By increasing the surface area, these studies have shown enhanced adsorption of analyte molecules, subsequently improving the gas-sensing response of the device. For a holistic understanding, it is essential to consider parameters like sensor temperature and gas concentration when evaluating the magnitude of the response.

Moreover, it has been realized that the electrical properties of chemiresistive structures are highly affected by the adsorption and desorption of gas molecules when the size of the sensing material is proportional to its Debye length. Additionally, recent advancements in the field of two-dimensional nanomaterials such as metal–organic frameworks (MOFs), graphene, and TMDs, with a thickness of a few atomic layers to tens of nanometers, due to unique physical (large surface area), chemical, and electronic properties, have made them suitable structures for the development of functional devices [79,80,81]. As reviewed elsewhere, MOSs (e.g., CuO, ZnO, TiO_2_, SnO_2_, In_2_O_3_, etc.) have also been exploited as a sensing interface in surface acoustic wave (SAW)-based detection of gases such as NO_x_, NH_3_, and H_2_S [82]. However, nanostructured n-type TiO_2_ has gained extensive interest because of its exceptional advantages of nontoxicity, wide bandgap (~3.1 eV), cost-effectiveness, and high chemical stability, attributions that are pivotal in gas sensing [83,84].

## 3. Wearable Substrates and Conducting Electrodes

There has been a lot of interest in flexible sensors due to their potential application in wearable chemical-sensing applications [85], especially in the disciplines of sports [86], health [87], medicine, and workplace safety [88]. Real-time monitoring of the body’s condition and surrounding environment enables a rapid response during an accident [89]. Since they may significantly affect people’s well-being, the gases produced as a byproduct of natural and industrial processes must be identified and monitored. A vital requirement for the practical use of wearable sensors is determining which substrate best matches the requisite wearable and flexible features. Therefore, numerous attempts have been made to design smart, flexible substrates for wearable applications. So far, different PB, polymeric, textile, and stretchable electronic substrates have been proposed for wearable sensing platforms. However, their characteristics should be carefully examined before choosing one for a specific application, as each offers different electrical and mechanical properties and production costs. At the same time, their processability can also be an issue [90].

### 3.1. Paper-Based (PB)

A PB substrate can also be employed as a starting substrate for manufacturing flexible and wearable gas sensors. Paper is becoming more popular in electronic devices as a flexible, low-cost, lightweight, tailorable, environmentally friendly, degradable, and renewable material. Since they degrade quickly, they can reduce the quantity of electronic garbage created for future generations. In recent years, numerous PB sensors have been described for wearable applications. PB gas, humidity, and strain sensors are among the most widely researched PB sensors. In general, electrodes are required for PB sensors. When compared to other substrates, PB substrates have an increased surface roughness and the existence of holes, both of which prevent them from being compatible with high-quality thin film deposition and transfer. Compared to other substrates, PB electrodes may be made using several processes, including printing, magnetron sputtering, conductive tape pasting, and pencil drawing [90]. Electrode production techniques such as printing, and magnetron sputtering can be applied on various substrates. Simple, low-cost, solvent-free processing methods such as pencil drawing and conductive tape pasting, which take advantage of the rough surface of the paper, can be used to make the electrodes of PB sensors. The sensing materials employed to construct PB gas sensors include carbon compounds and derivatives, MOS, and organic materials. Among these sensing materials, carbon compounds and their derivatives are particularly helpful. Because paper is not resistant to heat, traditional MOS gas-sensing materials that perform well at high temperatures are challenging. Furthermore, PB substrates are unsuitable for detecting VOCs due to their high hygroscopicity. This is because absorbed humidity affects conductivity more than the presence of the target VOCs [91]. PB moisture sensors have been demonstrated for tracking respiration rates and patterns [92]. These sensors are based on the propensity of paper to absorb moisture from its surroundings. Gas sensing has a wide range of applications, and it would be simple to print and attach a sensor like this to a human body.

Write printing is a method of printing that involves applying a solution of functional components to a surface with pens or other writing implements. When expelled from the nozzle tip, functional ink for write printing must harden fast. Printed ink should meet electrical, mechanical, and thermal standards to produce high-performance wearable sensors. For example, write printing technology has created PB wearable gas sensors of nearly “zero cost” [90]. These PB wearable gas sensors can achieve sufficient precision, ecological sustainability, and durable operation under high relative humidity (RH) settings due to appropriate printing materials and structural design.

A simple, low-cost, light hydrocarbon gas sensor has also been proposed based on bending a PB cantilever generated by polymer swelling [93]. A smartphone camera was employed for data reading. The functional layer of the sensory cantilever was made of polyethylene film, the adhesive layer of the sensory cantilever was made of double-sided sticky tape, and the substrate of the sensory cantilever was made of weighing paper. The hydrocarbons used to investigate the sensor’s performance were Xylene, n-hexane, benzene, ethylbenzene, and toluene (BTEX). The sensor demonstrated an excellent linear response to hydrocarbon content and a wide detection range, low detection, and rapid response performance.

### 3.2. Polymer-Based Sensors

Polymeric materials have gained popularity in sensing applications due to their processibility, durability, and low cost [57]. A recent study on polymeric sensors focuses on a nanocomposite of polypyrrole and TiO_2_ to detect volatile gas analytes such as ammonia, acetone, and ethanol. The distinguishing characteristic of this nanocomposite is that it is wrapped in poly-methyl methacrylate. Synthesis of polypyrrole was achieved through chemical oxidative polymerization of the pyrrole monomer. Gaseous analytes were detected with TiO_2_ nanoparticles in the composite structure. TiO_2_ is advantageous as it can endure temperatures as high as 600 °C, is abundant, has good stability, and is easy to process [94]. After the individual components of the composite material were prepared, the ternary nanocomposite was designed through physical methods such as grinding and mixing. The stoichiometric ratio of TiO_2_ to polypyrrole was taken as [1:1]. This ratio is critical as it helps obtain a composite of strong adsorption after 30 min of manual grinding. The gas-sensing performance of the nanocomposite was assessed with three different gases, as mentioned previously. The composite material was subjected to an air atmosphere to simulate real application conditions. The resulting composite material shows the highest sensitivity to NH_3_. Although this study does not propose a wearable solution for gas-monitoring technologies as it is a fiber-optic sensor, it is nevertheless easy and fast to fabricate, and it is a promising technology and approach for future wearable sensor technologies [94].

A wearable chemiresistive CP-based NH_3_ sensor was developed by constructing porous and neural-network-like electrospun films [95]. The intriguing characteristic of this sensor is that Au/Ppy was employed. The structure of the developed material (3D interconnections), the porous structure, and the synergistic effects between Au and Ppy nanoparticles make a sensitive sensor for NH_3_ detection and monitoring. The neural-network-like structure of the composite provides a practical pathway for signal transferring throughout the film, resulting in the effective transfer of electric signals from NH_3_-Ppy active sites to the working electrodes. The porous and hollow capsule-like structure is the key to fast and responsive detection of NH_3_. This structure allows for fast doping and re-doping between NH_3_ and Ppy capsules. This is especially important for detecting NH_3_ in the atmosphere quickly and effectively. Furthermore, the synergistic effects between Ppy and Au nanoparticles are essential for the number of active sites for NH_3_ to bind and to be detected by the sensor. This unique system promises a great future in wearable sensor technology applications due to its enhanced mechanical and electronic properties [95].

Another area of polymeric material research focused on fabricating cellulose acetate (CA)-based nanofibers and nanofilms [96]. The study aimed to detect H_2_S, which is crucial as it leads to respiratory problems [97]. The study’s novelty was using an environmentally friendly CA and an ionic liquid (glycerol) to facilitate charge transfer within the composite matrix. Ppy, a CP, and tungsten oxide (WO_3_) doping achieved the sensor’s conducting properties. Chemiresistive sensing nanofilms were prepared by solution casting, whereas nanofibers were prepared by electrospinning. The final sensor was fabricated and assembled by placing films/fibers between two plates of copper (bottom) and stainless steel (top). Stainless steel was chosen as the top contact as it is resistant to the corrosive effect of H_2_S. Heatproofing of the sensor was accomplished by using a conductive silver paste covering all the layers. Assessment of the H_2_S sensing capabilities of the sensor was carried out with a mass flowmeter, subjecting the sensor to H_2_S flows at a fixed rate and different concentrations (1–50 ppm). In addition, the sensor’s performance was tested under varying temperature conditions. This allows the testing of the sensor under different weather conditions. Test results revealed that nanofilm and nanofiber sensors could operate above 20 °C and have a minimum LOD of 1 ppm. In addition, the sensors showed a fast response time, nearly half a minute, with a quicker response time of 22.8 s. The sensors’ distinctive reproducibility properties, long-term stability, and low humidity dependence make them great candidates for application in outdoor and indoor atmospheres [96].

A selective sensor for NO_2_ detection was developed with a triazine-based 2D organic polymer [98]. A covalent triazine framework (CTF) was designed with a nanoscale thickness and intrinsic and periodic pore structures. In addition, this material possesses abundant functional groups, aiding the detection process. The outstanding property of this design is the ultrahigh sensitivity to NO_2_ in the atmosphere. The results showed that the response time of the material is between 35–47 s, and sensitivity to NO_2_ is at 452.6 ppm^−1^. Assessment of the gas-sensing performance of this triazine-based 2D organic polymer (T-2DP) was carried out together with a sensitivity assessment. Before the tests, T-2DP sheets were assembled into a thin film to act as the chemiresistive gas sensor. Various targets were used in these tests, including inorganics and organics. Some inorganic targets were NO_2_, NH_3_, and SO_2_, and the organic ones include methanol, ethanol, and acetone. It is crucial to mention that the detection and monitoring tests were carried out at RT. Sensing performance was evaluated using the variations in relative conductance of the material before and after exposure to the target gas. The highest response was observed for NO_2,_ with a response of 505% [98].

On the other hand, the material showed extremely low sensitivity to H_2_S (9.8%). In addition to sensitivity testing, response time was also tested, and similar results were obtained, strengthening the basis for NO_2_ detection. Response time was dependent on the gas concentration, and, upon NO_2_ exposure, response time was measured as 47 s for 150 ppb and 35 s for 5 ppb. Tests also revealed that the sensing behavior of T-2DP is reversible and reproducible. Even after 15 cycles, only a 4% variation in detection performance was observed during exposure to 1 ppm NO_2_. Finally, the mechanical durability of the sensor was tested by performing a bending test. The sensor was bent for multiple cycles with an angle of 45°. The 2D nanosheets provide inherent flexibility to the material; thus, the sensor displayed continuous constant conductance during bending. As a result, NO_2_ detection was observed to be stable for 500 cycles with only a 2.5% degradation after 1000 complete cycles. Overall, T-2DP was proven to be an excellent selective sensor for NO_2_ due to its reproducibility, durability, and sensitivity, making it a promising solution for the wearable detection of atmospheric pollutants [98].

### 3.3. Textiles

Textiles are a suitable substrate for the realization of wearable gaseous pollutant sensors, as such an approach would render the integration of such sensors with clothing trivial. Several characteristics make textiles attractive. These include their durability, breathability, and washability. Due to their availability, silk and cotton are the most commonly preferred textile fabrics [99,100]. Cotton fabrics have gathered significant attention due to their high flexibility, low cost, high moisture absorption, strong mechanical strength, good biocompatibility, and biodegradability. They can also be integrated into smart clothing [101]. Furthermore, encapsulation is essential in wearable sensor technologies regarding atmospheric conditions. Encapsulation of electronic sensors ensures a waterproof coating and acts as a shielding layer for human skin [102].

Graphene shows excellent performance in gas sensing; however, it also displays poor adhesion when used on textile materials. Research addressing this gap used amyloid nanofibrils to fabricate graphene-based electronic textiles (e-textile) [103]. Amyloid nanofibrils were used as a bio-inspired adhesive, promoting adhesion between the graphene flakes and the cotton yarn. Graphene flakes provide active sites for NO_2_ molecules to absorb, enabling the detection of pollutant gas in the environment. These interactions induce an increase in the electrical conductance of graphene. Figure 4a shows a schematic illustration of the fabrication process of different composite textile yarns.

Response characteristics of the e-textile yarns were measured through exposure to a constant concentration of NO_2_ (100 ppm) over 15 min, and electrical current differences were plotted (ΔI = I − I_0_). Among all textile composites examined, RGO/FBLG/CY had the best monitoring performance regarding NO_2_. In addition, the sensitivity of all materials was investigated by exposing them to NO_2_ between the concentration range of 0–100 ppm. The results show that, similarly to the results in monitoring performance, RGO/FBLG/CY achieved the highest sensitivity of 1 ppm, whereas the following best composite showed sensitivity between 3–5 ppm. RGO/FBLG/CY has the best performance among all five examined materials because amyloid nanofibrils retained the best affinity for GO flakes, as the GO flakes provided large binding sites for NO_2_ molecules. The selectivity tests were performed, and RGO/FBLG/CY showed a high response to NO_2_ and no reaction for N_2_, acetone, CO, CO_2_, NH_3_, and NO. The negative response to NH_3_ exposure can be attributed to the electron donor characteristics of NH_3_ on the rGO flakes, which results in decreased electrical conductivity. Finally, the humidity test revealed that RGO/FBLG/CY does not exhibit any change in electrical conductivity under humidity.

Consequently, RGO/FBLG/CY is a promising composite e-textile material for environmental gas monitoring with reliable performance. A schematic of the assembly and operation of the gas chamber is shown in Figure 4b, into which the e-textile gas sensor was embedded before NO_2_ exposure. Flow rate and concentration of NO_2_ were controlled through an MFC (Figure 4c). As shown in Figure 4d, the red LED light is not lit before NO_2_ exposure while it is turned on after exposure to NO_2_ (10 ppm) for 4 min (Figure 4e).

As graphene shows significant NO_2_-sensing characteristics, various study platforms exploited this material. Another textile application for NO_2_ detection was achieved by developing a dopamine–graphene hybrid with highly conductive and flexible properties [104]. As previously discussed, graphene needs an adhesive agent to be used in textile applications. In this study, dopamine was employed as the adhesive biomolecule. Dopamine’s adhesive properties come from the structure of catechol and amine groups. E-textile yarns were exposed to NO_2_ to test their feasibility in gas monitoring. Electrical current differences (ΔI) were plotted during the exposure (100 ppm). The plot shows that the electrical currents of DGY increased exponentially over time, whereas rGO-coated yarn displayed almost no increase. For both materials, saturation time was calculated, and DGY had a response time of ~1.5 min, whereas rGO-coated yarn had a 4 min response time. The materials’ sensitivity was characterized by exposing them to various concentrations of NO_2_ ranging between 0–100 ppm and measuring their electrical conductivity. Although both materials display a linear increase, DGY showed greater sensitivity than the rGO-coated yarn, indicating the fabrication of a superior material regarding sensitivity to NO_2_. This is one of the key characteristics provided by dopamine, as rGO flakes are attached more to DGY than rGO-coated yarn. The high sensitivity of DGY allows detection at levels as low as 0–10 ppm NO_2_. Reproducibility tests were also performed to assess the performance of the fabricated DGYs under repeated applications. Tests were conducted under 10 ppm NO_2_ (exposure limit) and 2 min of exposure time, followed by 3 min of N_2_ gas. This cycle was repeated five times for both samples. These results conclude that DGY can be applied for wearable sensing applications, as it is reliable and durable. In addition, washability tests were conducted to confirm DGY’s use for human applications. Like the reliability test, washability was tested in five cycles using soapy water. After five cycles of washing, DGYs did not display a significant decrease in electrical conductivity (2.96% variation). Fabricated DGYs offer excellent durability, washability, and sensing characteristics, making them great candidates for wearable e-textiles for gaseous pollution monitoring systems.

Regarding wearable textile sensors, colorimetric detection is also applied. Colorimetric detection provides easy and fast detection of gaseous pollutants in the environment. A study regarding colorimetric detection was conducted for the simultaneous detection of NH_3_ and HCl gases [105]. Combined detection of gaseous pollutants is essential and should be further investigated. There are also textile sensor applications for the dual detection of NO_2_ and NH_3_ in the literature. For instance, polyester textile was employed in the sensor. The sensor was graphene-based, and the detection of gases from human breath was aimed [106]. Overall, colorimetric detection of these gaseous pollutants is critical as they are colorless and spread quickly.

### 3.4. Stretchable Electronics

Standard electronics and microelectronics are primarily based on thick, rigid, and fragile substrates. For example, traditional printed circuit boards (PCBs) typically use flame retardant (e.g., FR4) or other substrate materials. At the same time, bipolar junction transistor (BJT), bipolar complementary metal-oxide-semiconductor (BiCMOS), and CMOS technologies are traditionally based on silicon wafers. Wearable applications necessitate a move away from these technologies and into devices and systems that are flexible and ideally stretchable. The human body and skin are not flat, rigid surfaces; they are in constant motion and soft, flexible, and stretchable. The inconsistency between the mechanical properties of skin and electronic devices can lead to many issues; for example, the wearable device might get damaged due to the continuous motions and can fail or become detached from the skin or the wearer can also become injured from prolonged device use. Depending on the sensing modality, it can also lead to recording noise signals and motion artefacts. Other issues are related to the wearability and intrusiveness of the devices, which can create emotional stress for the wearer or a feeling of intrusiveness.

In contrast, recorded data may be biased by the wearability factor, as the device itself may prohibit the wearer from performing daily tasks [107]. First-generation wearable devices have become widely available over the last ten years. However, these are rigid devices that are limited in functionality. It has, thus, become obvious, driven by current market and societal needs, that advanced next-generation technologies are needed to improve wearable devices’ wearability, form factor, un-intrusiveness, and ubiquity. Flexibility and stretchability are key factors in this direction. Technologies and devices can be divided into techniques and devices realized through standard clean-room-based microfabrication and additive manufacturing approaches. Microfabrication techniques (thin-film deposition methods, photolithography, etc.) can lead to high-quality and performance devices with high resolution. However, they require high-cost, complex fabrication tools and facilities. Additive fabrication techniques (stencil screen, inkjet, extrusion-based printing, laser carbonization, etc.) are lower-cost simpler technologies, which, however, are limited in resolution [107,108,109,110]. Many devices have been demonstrated using such techniques, including strain/pressure sensors, interconnects, and electrodes. Hybrid technologies exploiting the advantages of each of these families of technologies but also in combination with standard rigid technologies have emerged to address the challenges of wearable applications, particularly since the performance and level of integration of traditional microelectronics technologies cannot yet be rivalled. For example, the island-bridge approach can be used to allow some components and the electrical interconnects to be highly flexible and stretchable and rigid components to sit on islands, thus allowing a hybrid implementation [110].

As the substrates of on-body types of wearable gas sensors are in direct contact with the skin surface, they must be light, thin, flexible, and, most importantly, breathable to avoid airflow blockage and inflammation resulting from sweating. A sufficient stretchability provides appropriate stability to the sensor during movements. However, conventional flexible substrates such as polyethylene terephthalate and polyimide have low stretchability in the flat film form, so their suitability for epidermal electronics is low. Similarly, PB and textile substrates are unsuitable for manufacturing high-density circuits. They also cannot be used for the deposition of high-quality sensing materials. Therefore, designing advanced stretchable substrates with proper stability and durability becomes crucial in fabricating wearable sensing devices [111]. Various studies have reported the advancement of nanomaterial-based stretchable substrates and their application in stretchable electronics [112,113]. However, the major challenge lies in the ability of a device to be stretchable and, at the same time, allow electronic circuits and devices to retain electrical function and performance even under strain. Various unconventional materials have been used to construct functional device components and electronic circuits to overcome this issue. Poly(dimethyl siloxane) (PDMS) and Ecoflex (types of platinum-catalyzed silicones) are the two most common soft substrates used in the fabrication of stretchable electronics [114]. Other popular candidates as elastic substrates include poly(styrene-co-ethylene butylene-co-styrene) (SEBS)- and polyurethane (PU)-based elastomers [115]. A thin layer of soft PDMS and a strip of Ecoflex has also been tested to produce a stretchable, low-temperature, and ultrasensitive MoS_2_/rGO-based sensor for detecting NO_2_ at 10 ppb. The nanocomposite was integrated directly into a conductive laser-induced graphene pattern to avoid using an interdigitated electrode. Due to the enhanced contact with the gas-sensing nanomaterials, these stretchable porous 3D patterns yielded exceptional selectivity and robust operation upon mechanical deformation. Moreover, the rGO/MoS_2_ nanocomposite provides a controlled highly specific surface area and overcomes the drawbacks of pristine MoS_2_ in NO_2_ sensing, including low conductivity [116] (Figure 5).

## 4. Sensor Types

Conventional equipment to detect quantitatively, characterize, and classify different gases are based on gas chromatography (GC), mass spectrometry (MS), ion mobility spectrometry (IMS), pellistors, semiconductor gas sensors, or electrochemical devices [4]. Although these methods of gas detection are stable and reliable, their use outside laboratories are limited due to their massive power consumption, cost, and size [117,118]. As the demand for portable gas sensor devices with low power consumption and accurate sensing performance increases, alternative methods, such as the cantilever-based, capacitive-based, thermometric, optical, field-effect transistor, solid-state electrochemical, and chemiresistive or colorimetric gas sensors, have gained a lot of attention [118].

### 4.1. Colorimetric

Colorimetric-based gas sensors can detect the presence of gaseous analytes and the concentration of the analytes through chemical reactions, resulting in color changes in the sensor [118]. The use of colorimetric gas sensors, such as thread-based washable textile gas sensors, is increasing due to their low power consumption and low LOD, low cost, selectivity, and multiplexity [119]. Additionally, the color changing of these sensors is easy to visualize and, hence, is intuitive, attributions that make them ideal for application in monitoring of air quality, personal exposure tracking, assessment of food quality, detection of hazardous chemicals, and breath analysis. In addition, using a smartphone camera to analyze their response quantitatively is another attractive characteristic. Colorimetric sensors typically employ various organic compounds for capturing target molecules, such as dyes, fetal organic complexes, and polymers [118]. Colorimetric sensors operate based on various mechanisms. One is based on the ring opening and closing reactions [117]. Various dyes and organic compounds of aromatic structures with several conjugated π-systems can be used as colorimetric sensors based on the ring opening and closing reactions. This leads to a change in the number of conjugated π-bindings.

Change in the number of conjugated π-systems or lengthening/extending of a conjugated system leads to a shift in the absorbance spectra of a material; i.e., the wavelength of the photon captured is affected. Thus, specific photons are captured, leading to a measurable change in the material’s color and, therefore, the detection of a target analyte. Another mechanism exploited for colorimetric sensors is based on functional groups. These are specific moieties that can enrich the characteristic features of organic compounds and increase reactivity [117]. One such example is organic compounds with hydroxyl groups that can oxidise quickly. Reaction with a target gas changes these functional groups to other functional groups, leading to a detectable material color change. In a different colorimetric sensor mechanism, ligand exchange is exploited. Electronic transitions in the energy level of d-orbitals define the color of transition metal complexes. Ligands around transition metal ions can affect these. If they are changed or substituted, the interaction between the d-orbitals of the transition metal and the electron cloud of the ligand is affected, changing the electrons’ state, thus leading to the absorption of different wavelengths of light and, hence, to a color change. Another common approach is based on phase transition. Various inorganic compounds can react with gases in a catalytic reaction that leads to phase transition and, thus, a change in color. One example is iron oxidized in air to produce iron oxide [117]. Though, the real-world applications of colorimetric gas sensors are limited by the interference of humidity and their variability. To tackle the interference of humidity, the use of hydrophobic substrates is suggested. Proposed materials to serve this purpose include reverse-phase silica thin-layer-chromatography plates, polyvinylidene fluoride membranes, and polyethylene terephthalate films. However, the main drawback of colorimetric sensors is that they are limited to ionic compound detectors, making them less ideal for effectively sensing gaseous hazardous and toxic substances [120,121,122,123,124].

### 4.2. Optical

Optical-based gas sensors have the advantages of fast response (allowing rapid real-time detection) with minimum drift (owing to absorption of gas molecules at a specific wavelength) and extreme gas specificity and sensitivity without disturbing the gas sample. If they are appropriately designed, cross-response to other gases can be eliminated, which makes optical gas sensors inherently reliable [125,126]. Furthermore, optical gas sensors can distinguish various gas species by comparing their unique gas optical fingerprints. Optical gas sensors, thus, exhibit remarkably high selectivity, with exceptional physical and chemical stability compared to chemiresistive metal oxide sensors. There are several different optical techniques available that are suitable for gas sensing. These include non-dispersive infrared (NDIR) sensing, photoacoustic spectroscopy (PAS), tunable diode laser absorption spectroscopy (TDLAS), and spectrophotometry, some of that are discussed below.

NDIR gas sensing is a popular optical technique. The gas sample is kept in a small compartment, and an IR light source is passed through it, reaching an IR detector. The recorded spectrum provides indicative peaks at different wavelengths corresponding to absorption bands specific for different analytes [127,128,129]. This arises from vibrations of the gas atoms and allows the selective detection of multiple gases simultaneously, provided there is no overlap in their fingerprint absorption spectra. Most toxic gases demonstrate strong absorption in the mid-IR (2.5–14 μm), about 100 times more than in the near-IR region [127]. The stability/drift of the light source is recognized as a factor that can negatively impact measurement precision in these systems, together with noise in the detector, humidity, and the optical path required [127,129]. When analyzing gas mixtures, a long optical path is needed that can consist of multiple detectors, optical beam splitters and filters, and other components, increasing the overall system size, complexity and cost [128]. While such systems have become small and more portable, their wearable implementation can be challenging.

A multi-pass multi-channel system was proposed for multi-gas detection [130]. Using a single broadband incandescent light source and three optical path lengths of 24, 36, and 48 cm, three different gases could be detected: CO_2_, CH_4_, or N_2_O with sub-ppm resolution, combined with H_2_O and CO_2_ for monitoring with a 1 ppm resolution. Three wavelength-selective detectors were mounted to one of the two facing mirrors of the optical cell, separated by a complete propagation cycle between the spherical mirrors and at each path length. The device’s response time was measured to be in the region of 15 s. Measurement precision was found to be better than 0.1 ppm for CO_2_ and N_2_O, about 0.4 ppm for CH_4_, and 2 ppm for H_2_O. In addition, the system’s power consumption was below 0.5 W, and the device size was equal to 4 cm × 3 cm × 3 cm.

Spectroscopic measurements using quantum cascade laser (QCL) technology are also typical [128]. These are stacked atomic-level thickness semiconductor lasers in the mid-IR, focused using several lenses. However, they require vacuum pumps, alignment of their optics, chambers, and overall large, costly embodiments, making them unsuitable for wearable applications. Therefore, QCLs have been combined with PAS, tunable IR laser differential absorption spectroscopy (TIL-DAS), wavelength modulation spectroscopy (WMS), and quartz enhancement, among others, to reduce their overall size and improve portability [128]. In one approach, an external-cavity QCL (EC-QCL) was developed for the simultaneous detection of CH_4_, N_2_O, and H_2_O vapor using off-beam quartz-enhanced photoacoustic spectroscopy (OB-QEPAS) [131]. The excitation wavelength was scanned over the three absorption lines of CH_4_ (1260.81 cm^−1^), N_2_O (1261.06 cm^−1^), and H_2_O vapor (1261.58 cm^−1^) by tuning the grating of the EC-QCL with a piezoelectric actuator. The impact in the generation of QEPAS signals due to light absorption by CH_4_ and N_2_O was investigated in the mid-IR region around 8 µm. Water vapor has a beneficial influence, which enhanced the QEPAS signals by three times for CH_4_ in the air and by 20% for N_2_O in the air compared to those in dry samples. It is, thus, essential to monitor the water content in measured samples when using this method. This approach achieved LODs of 98 ppbv for CH_4_, 12 ppbv for N_2_O, and 750 ppmv for H_2_O vapor in humidified gas mixtures. When applying a real-time Kalman filter to improve measurement precision by a factor of ~4 while keeping the same temporal resolution, these numbers changed to 60 ppbv for CH_4_, 10 ppbv for N_2_O, and 0.07% for H_2_O in the measurements of 1.99 ppmv CH_4_ and 312 ppbv N_2_O humidified with 2.8% H_2_O vapor.

Optical-fiber-based gas sensors also operate at RT. One such example is the multi-mode interference of single-mode—no-core—single-mode fiber (SNS) structure with MoS_2_ nanosheets as the sensitive film presented in [126] and allowed measurements of formic acid from 0 to 250 ppm in 50 ppm steps. Another optical approach is quartz-enhanced photoacoustic spectroscopy (QEPAS) and its variant beat-frequency quartz-enhanced photoacoustic spectroscopy (BF-QEPAS) using a quartz tuning fork (QTF) as a resonant acoustic transducer operating at the mid-infrared (MIR) with a quantum cascade laser. BF-QEPAS allows rapidly and simultaneously obtaining the resonance frequency and Q-factor of the QTF and the trace gas concentration. For example, examining the adsorption–desorption effect and optimizing the modulation depth and frequency has enabled an ammonia detection limit of 9.5 ppb with an integration time of 3 ms with BF-QEPAS [125].

Mid-wave infrared (MWIR) absorption spectroscopy is another gas-sensing method worth mentioning, as it is suitable for mass-producing compact and inexpensive devices and systems [132]. A MWIR graphene photodetector was recently proposed that is integrated into silicon-on-sapphire (SOS) slot waveguides for on-chip absorption spectroscopy measurements of gases. Simulations indicated that the detection of CO_2_, CH_4_, and N_2_O with an LOD below ppb levels (0.14, 0.72, and 0.52 ppb, respectively, at 2 mW source power) could be achievable with the proposed device [132].

Additionally, incorporating functional nanomaterials such as polymers, noble metal/metal oxide composites, graphene, and its derivatives into optical gas sensors can enable gas interaction with the sensitive evanescent field of the transmitted optical mode by the adsorbed gas molecules [133,134]. However, sophisticated interferometers or resonators such as Mach–Zehnder interferometers (MZI) and Bragg gratings are needed to correct the sensor output based on the wavelength shifts. This can considerably increase the design and fabrication cost [135]. As an alternative, fiber-optic-based gas sensors such as fiber tips and tapered/D-shaped fibers are relatively more straightforward in design and fabrication [4].

Generally speaking, optical approaches are not the most suitable sensing techniques for wearable applications, due to their overall size, especially when considering a complete system, their power consumption, and aspects related to wearability (flexibility, stretchability, mechanical robustness, etc.). However, they are discussed herein for completeness and in an effort to stimulate future research into the wearability of such sensing approaches.

### 4.3. Electrochemical

Compared to traditional instrumentation approaches, which require specialized apparatus and complicated protocols, electrochemical sensors have several distinct advantages when it comes to detecting environmental contaminants. As a result, they have been applied in various applications, including clinical diagnostics, food safety and quality, biological analysis, and environmental monitoring. These advantages include ease of use, low cost and power consumption, high miniaturization, and relatively simple instrumentation [136]. When it comes to wearable sensors, biocompatibility is an additional consideration. Electrochemical gas sensors are relatively more specific than other methods to detect individual gases and have ppm- or ppb-level sensitivity. However, similarly to semiconductor-based sensors, they suffer from cross-response issues and have a limited lifetime [137,138,139]. In addition, they are susceptible to temperature fluctuations; thus, the temperature should be kept constant and ideally known [140].

Like other types of sensors, the electrode materials should have high conductivity. However, they should also have an excellent electrochemical potential window and catalytic activity. The electrochemical interaction at the analyte–electrode interface leads to an electrically detectable signal proportional to the concentration of the analyte that was engaged in the process. There are several different electrochemical methods. The most commonly used in gas-sensing applications are cyclic voltammetry (CV, where a cycling triangular voltage excitation is applied to the electrochemical cell and a resulting current is measured) and amperometry, where a constant potential known to facilitate a redox reaction in the studied system is applied, and the resulting current is again measured. Typically, a three-electrode cell is used in these electrochemical techniques, composed of a working (WE), a counter (CE), and a reference electrode (RE). The CE, typically made of Pt, should ideally have a surface area 10 times greater than the WE, so its interfacial impedance does not become a significant part of the measurement. The RE is typically a standard Ag/AgCl electrode. If the current can be made small and the ohmic drop between CE and RE can be made infinitesimal by placing the electrode very closely together through their miniaturization, then two electrodes, a WE and a CE, can also be used [128,141]. Depending on the applications, CV can be exploited as a first step to identify the redox potential required for amperometric measurements, which can serve as the main detection method, as it is a simpler method. It is important to note that different gases have different redox potentials. For example, O_2_ and CO_2_ have less negative reduction potentials than N_2_O, while N_2_ has higher [128].

Amperometric gas sensors are often based on electrochemical fuel cell technology. Catalysts that are highly loaded with Pt are typically used to catalyze the oxidation reaction of the targeted gases at the WE and the oxygen reduction reaction at the CE [142]. Nevertheless, Pt is expensive, and another disadvantage is its poisoning by various gases when used in ambient conditions, which hampers sensor performance over time. Other common electrode materials include Rh, Ru, Pd, Au, Ir, and Ag. Nevertheless, carbon-based materials are also common. Typically, the WE is either within a sensing chamber or is itself porous. In any case, some kind of porous diffusion barrier is often employed. Increase of the applied potential leads to an increase in the reaction rate until it reaches a plateau, where the diffusion flow of the target gas is maximized. Another electrochemical method that can be exploited for gas detection is potentiometry. This method requires the use of only two electrodes and relies on the measurement of the difference in the polarization potential between two electrodes, one indifferent (RE) electrode whose potential is ideally always stable and unaffected by changes in its electrochemical environment and a WE, where the actual detection of target analytes takes place.

In contrast to amperometry and CV, in potentiometry, there is no current flow. The open circuit potential formed at the electrode interfaces is defined by the Nernst equation and it is, thus, logarithmically related to gas concentration. It is, thus, less sensitive to low concentrations, in contrast to amperometric sensors, where the limiting current is linearly related to the gas concentration. One common potentiometric gas sensor example is the lambda sensor employed for the detection of oxygen in exhaust gases with a solid oxide electrolyte (YSZ, discussed further below). Another type of sensors are mixed potential gas sensors, where multiple reactions take place, leading to a non-Nernstian response that does not reach equilibrium as in potentiometric sensors, and a mixed potential is observed. This method is typically used for measurements in non-equilibrium gas mixtures containing H_2_, CO, CO_2_, hydrocarbons, SO_2_, NO_x_, NH_3_, and H_2_S [143]. Primarily portable and benchtop mixed potential sensors have been demonstrated [144,145,146,147,148]. An investigation towards wearable and room-temperature realizations of such gas-sensing devices could be a promising research direction. In addition to the above, there are also mixed amperometric/potentiometric sensing systems composed of two cells, as a means of improving overall system sensing performance. Finally, electrochemical impedance spectroscopy (EIS) gas sensors are another approach often used, where the electrochemical cell is excited with a sinusoidal voltage of varying frequency. EIS is typically used with semiconductor-based and solid-oxide-based sensors.

At this point, it is also important to highlight that the two or three electrodes are spatially separated. A conductive medium is required to close the circuit and allow the current flow or to establish the polarization potentials at each electrode. This is typically achieved through an electrolyte, which can be either in liquid or solid forms, with the latter preferred for portable and wearable instrumentation. These are known as solid-electrolyte-based electrochemical gas sensors. Some solid electrolytes demonstrating ionic conductivity include AgS, Cu_2_S, 85 mass% zirconia and 15 mass% yttria (YSZ), or fluorites such as ZrO_2_ that is stabilized with Y, Yb, Ca, and Sc oxides, or their mixture, CeO_2_-based materials doped with lanthanides, and lanthanum gallate (LaGaO_3_)-based materials, LaYO_3_ and SrZrO_3_, and BaCeO_3_ and yttria-doped BaCeO_3_, although these can be properties encountered at elevated temperatures (e.g., 600 °C), unsuitable for wearable applications [143]. Care must be taken to increase the selectivity and specificity of a particular gas. Several approaches have been exploited to achieve this. One approach involves the use of two electrochemical cells. In the first, a reduction potential is used that is below that for the target gas, the reduction potential of which is used in the second cell. This potential can reduce other species in the mixture. The difference in the recorded current can thus indicate the target gas concentration. Alternatively, reducing interfering gases can be used to completely remove them from the gas mixture, such that sensing of the target gas can be facilitated in a subsequent chamber. Another approach involves the choice of the electrolyte used. Scavengers of interfering gases can be exploited in one- or two-compartment devices to eliminate them from the measurement. For example, scavengers of O_2_ include alkaline ascorbate solutions and diphenylphosphine in propylene carbonate. Several recent electrochemical gas sensor examples were recently reviewed elsewhere [143].

One recent example is a platform for the selective electrochemical sensing of nitrite based on SnO_2_/Pt/Ti/SiO_2_/Si [149]. The sensor was produced by depositing a material composed of SnO_2_ onto a substrate consisting of SiO_2_/Si. Synthesis of SnO_2_ was accomplished through the microwave irradiation method (MW), and the coating of SnO_2_ on the substrate was found to generate free electrons. By depositing a Ti/Pt interlayer on the substrate, the adherence of the SnO_2_ coating can be further improved. In addition to that function, this interlayer served as the metallic contact electrode layer. The sensor exhibited the highest level of electrochemical sensing when exposed to nitrite. In addition, it can be utilized to analyze nitrite levels in samples of beer, milk, municipal water, and mineral water. A linear response and a lower LOD of approximately 10–400 μM and 1.7 (±0.1) μM were achieved. The MW approach provides a route for synthesizing SnO_2_ that is quick and reproducible. In addition, the MW-based SnO_2_ coatings had a higher conductivity and sensitivity than those created using the traditional sol-gel process.

### 4.4. Transistors

In these types of sensors, sensing is achieved through a transistor, mainly using the MOS FET. One advantage of transistor-based sensors is that they have the potential for high miniaturization and a high level of integration with all the necessary readout electronics in the form of a standalone integrated circuit. Consequently, they are highly suited for wearable applications. Furthermore, they have the potential to overcome limitations found in other technologies concerning size, power consumption, dynamic range, and sensitivity [150]. On the other hand, although semiconductor-based gas sensors are susceptible at the low ppm level, they suffer from changing humidity levels which can cause drift and cross-response to other gases [151,152,153]. The subject and fundamental operation of the various approaches for FET-based sensing were recently reviewed [154], and, thus, here, we will focus on recent developments. Briefly, we will summarize the basic strategies proposed for FET-based gas sensing. As it will be further elucidated, there are two main approaches to achieving FET-based gas sensing. Sensing can either be performed (a) via the gate or (b) directly through the semiconducting channel.

#### 4.4.1. Gate Sensing

Often, detection is performed at the gate of a MOSFET. For example, one of the transistors in a differential input pair of an instrumentation or operational amplifier can be a gas-sensing transistor, allowing the direct measurement of an analyte by transforming a change in transistor gate potential relevant to a reference potential into a proportional drain current based on the device’s IV characteristics. Gas sensors are often based on a resistance change from gas-oxide redox reactions on transition metal-oxides [150]. Alternatively, instead of directly reacting with the metal oxide, the detected gas reacts with the surrounding oxygen, improving stability for long-term use. A sensing film is required to enhance sensitivity and selectivity. Various 2D materials have been proposed for chemiresistive-type NO_x_ gas sensors. These include TMDs, graphene, and black phosphorous [150].

As discussed elsewhere in detail [154], gate-based FET sensing can be achieved via multiple approaches. The earliest approach is based on the catalytic metal-gate FET [155]. Detection is based on changes in the dipole moment formed at the interface between the metal gate and the gate insulator. This dipole layer is formed through the catalytic reaction at the gate that leads to penetration of the gas molecules through the metal. This modulates the threshold voltage and, thus, the channel current of the device. However, the applicability of this approach is limited by the molecular size of the target gas. To address this, the suspended-gate FET (SGFET) was proposed. This structure employs an air gap that allows gases of any molecular size to be detected. However, the SGFET is challenging to manufacture; thus, various variations of this architecture have been proposed. One is the capacitively coupled FET (CCFET), which can reduce the operating voltage required and increase sensitivity through enhanced gate coupling [154].

An air gap is still present, but, in contrast to the SGFET, a portion of the gate is in contact with the gate oxide. Essentially, there is a capacitor, with one of its plates covered with a sensing layer and a floating potential, coupled to the gate electrode of the FET. The other has a well-defined potential and acts as the source electrode. Adsorption of gas molecules on the floating gate changes its potential, changing the channel current. Finally, the horizontal floating-gate FET (HFGFET) is the most recent and probably the most common approach that combines many of the aforementioned approaches [154]. A control and a floating gate are horizontally fabricated in the same place. The control gate is directly connected to the sensing layer, and the floating gate is insulated. The sensing layer sits on top of both, allowing a higher coupling ratio between these two layers. As a result, gases of any molecular size can be sensed, and, since there is no need for an air gap, the manufacturing challenges faced by the SGFET and CCFET are eliminated. In addition, like the catalytic gate FET, the HFGFET is more compatible with standard CMOS processes. A similar approach has recently been exploited, as explained below.

SnSe_2_ is a TMD with a high surface-area-to-volume ratio, and it has been used in CMOS-compatible gas sensors. Recently, a fully CMOS-technology-compatible gas sensing system was presented [150]. This work proposed floating-gate devices composed of two inputs. This highly integrated CMOS chip was designed and fabricated with a standard commercial 28 nm CMOS technology node and post-processed for incorporating a SnSe_2_ film. Figure 6a illustrates the proposed approach, and the technology nodes’ metal layers and the interdigital electrodes used. The proposed structure was composed of a reading gate (RG), a sensing gate (SG) used for the detection of NO_2_, and a floating gate (FG). The potential in the SG is coupled with the FG, which alters the FET’s channel current. Changing the SG-to-RG coupling ratio adjusts the coupled potential. An interdigital metal–oxide–metal (MOM) capacitor structure was used to couple the FG, RG, and SG, as shown in Figure 6a,b. Such a structure possesses a relatively high aspect ratio, providing a strong capacitive coupling effect that further enhances sensitivity. Several readout circuit approaches were proposed. The first involved a potential divider formed with a standard resistor and a SnSe_2_-sensing film. The varying potential was coupled to the FGMOS circuit of Figure 6c with the RG set to 0V, leading to an overall sensitivity of 15.42%. Different sensing gate coupling ratios lead to a diverse range of measured currents as a function of gas concentration (Figure 6d). In another approach, the threshold voltage was exploited to fix the channel current through a self-balanced circuit (Figure 6f) comprised of the FGMOS transistor and an operational amplifier (opamp) in negative feedback feeding the RG potential. This approach achieved a 102 mV ppm-1 sensitivity in the 1–25 ppm NO_x_ range. RT operation was conducted by optimizing the coupling capacitors and the sensing film thickness. At the same time, process variations could be calibrated by adjusting the sensing range of the circuit through the reference voltage in the opamp-based circuit of Figure 6e,f. The 3 × 3 and 16 × 16 gas sensing arrays with peripheral circuits were demonstrated to detect multiple gases simultaneously, showing a unique and complete standalone system on chip with all the required sensing and control (timing and decoder) electronics, including calibration methods [150].

Gas-sensing FETs often perform sensing through the semiconducting channel. However, this may impact the intrinsic properties of the device, hindering its operation as an active circuit element. Thus, isolating it from the gas and performing detection at the gate allows us to exploit the stable properties of the FET for processing the gas-depended signal arising at the gate. This was demonstrated by using a CNT film as the semiconducting channel, and a Y_2_O_3_ dielectric layer, with the detection of HCHO, achieved at a Pd/Au catalytic metal gate and leading to a 20 ppb LOD at RT between 20 to 800 ppb. The device has a hole mobility of 1812 cm^2^/Vs. The high sensitivity at RT is attributed to the amplification achieved with the device.

Nevertheless, at RT, the sensor does not fully recover to baseline values and drifts upwards at 18.5% over time. However, operating the sensor at 150 °C allows complete recovery (97%), a lower LOD of 10 ppb, improved linearity from 0.86 to 0.99 (linear correlation coefficient, R), and minimization of the baseline drift. The specific operating temperature and gate material allow selectivity towards HCHO. The humidity did not affect the device operation (up to 80% RH), while the device also demonstrated long-term stability in 20-day measurements at 80 ppb [155]. Work function variations due to gas adsorption on the gate change the channel current. When physisorption takes place, a dipole layer is formed on the gate, increasing the work function of the gate, and shrinking the conductive channel due to additional holes induced around the semiconducting CNT surface, leading to a smaller channel current. At elevated temperatures, where chemisorption occurs, the current trend is reversed. Due to the difference in chemical potentials between the Pd/Au gate and the gas, chemical reactions take place. The gate will dissociate chemisorbed HCHO molecules to yield hydrogen that diffuses between the gate and the dielectric forming a dipole layer. This broadens the conductive channel and, thus, increases the channel current. The stacked Pd-Au gate plays a crucial role in the device’s performance; Au is the chemical sensitizer, while the synergistic effect of the bimetallic structure enhances sensitivity by promoting the chemical reaction of HCHO [155] (Figure 7).

#### 4.4.2. Channel Sensing

Alternatively, the transistor channel can be sensorized, allowing a direct change in the drain-source current due to a transduction mechanism for a defined gate potential. To ensure that the measured response is only due to the channel, all other areas of the device, especially source and drain metal contacts, must be sufficiently passivated. Much of the research focuses on developing novel transistor architectures and materials for the transduction mechanism. One material that holds great promise in many electronics and sensing applications is graphene. As a transistor channel material, it has also been explored for gas sensing. One such approach involved the development of n-doped graphene with ethylene amines that increase the device’s carrier mobility and creates specificity and sensitivity in the channel current to oxidizing gases [156]. Upon exposure to oxidizing gases such as NO_2_ at a fixed 0 V gate voltage, the channel current decrease depends strongly on the number of amine functional groups in various ethylene amines. Non-destructive vapor-phase molecular doping under applied heat with diethylenetriamine (DETA) has been reported to lead to a high response, recovery, and long-term sensing stability to NO_2_ when compared to other doping agents such as ethylenediamine (EDA) and triethylenetetramine (TETA). An average LOD of 0.83 parts per quadrillion (ppq, 10^−15^) was reported due to the attractive electrostatic interaction between electron-rich graphene and electron-deficient NO_2_ [157]. Oxidizing gas molecules of NO_2_ or SO_2_ introduced charged impurities leading to an enhanced decrease in the channel current. On the other hand, the absence of such a strong interaction between DETA-n-doped graphene and NH_3_, combined with different effects, leads to insensitivity to NH_3_ [156].

InGaZnO (IGZO) is a promising material for thin-film transistor realization, especially for flexible transistors on thin, flexible substrates [157]. IGZO has also been exploited for gas sensing. A top contact bottom gate IGZO transistor with a 10^7^ on/off ratio and a charge carrier mobility of 7.6 cm^2^V^−1^s^−1^ has been used for sensing NO_2_ [158]. In the presence of NO_2_, measurable changes in the threshold voltage (V_Th_) and drain current (I_D_) of the device were observed due to the adsorption of the gas onto the IGZO channel. Adsorption due to the high electronic affinity of NO_2_ and its electrophilic nature leads to the oxidation of IGZO and a resulting charge carrier depletion. Illumination with a blue (450 nm) LED at 1 mW/cm^2^ restored the device to its condition before NO_2_ gas exposure [158]. The required illumination duration depends on the NO_2_ gas concentration. The exposed (unpassivated) sensing IGZO was diode-connected, generating a current for a set V_DD_ proportional to the NO_2_ concentration. The rest of the transistors were insulated to ensure their characteristics remained unaltered in the presence of NO_2_. A simple NMOS current mirror was realized with two passivated IGZO transistors copying the sensing IGZO current into a second branch. A third passivated IGZO was diode-connected and set to a VDD1 voltage to create a competing current that would trigger an inverter when the NO_2_ concentration exceeded an established limit. This, thus, created a 1-bit detector. Copying the current to branches with a different current, thus, allows thresholding at different levels and, thus, increases the system’s sensing resolution (bits) to detect various gas levels [159]. Power consumption was in the μW range for the complete readout system, which did not require additional amplifiers or analogue-to-digital converters. The LOD of the IGZO RT sensor was 100 ppb [158].

Organic gas-sensing FETs (OFTs) have also been proposed, where gas sensing is achieved through the exposure of the organic semiconductor and the corresponding change of drain current due to a specific gas. Doping of the organic semiconductor, change of its dipole moment, or change in the morphology of the organic film are various mechanisms that have been proposed to explain gas sensing with such devices [159]. One advantage of organic field effect transistors is that they can be operated in RT and can be solution-processed using inexpensive methods (e.g., spin-coating) and fabricated on flexible plastic substrates. Nevertheless, a common barrier to their widespread use arises due to the deterioration of these devices by humid air. In n-type organic semiconductors, humidity can generate electron traps at the edge of the conduction band. At the same time, in p-type materials, it acts as a dopant that increases the off-current of the device [159]. Typically, OFTs are encapsulated to address this issue. However, this is not possible in gas-sensing applications, as the channel needs to be exposed, and, thus, they will be exposed to humidity. This was recently addressed with the use of molecular additives of tetracyanoquinodimethane (TCNQ) and 4-aminobenzonitrile (ABN) in an indacenodithiophene-co-benzothiadiazole (IDT-BT) semiconductor. This approach eliminates moisture adsorption, leading to a reversible, stable response for NH_3_ sensors that can operate for over 5 h continuously and in humid air for 16 days [159]. Detection is performed by the n-type doping effect of the IDT-BT channel by amine in the NH_3_ gas.

A bottom gate architecture, which allowed the organic channel to be exposed and on the top of the device (W/L = 1 mm/100 μm), was implemented using a flexible plastic PEN substrate. The bare IDT-BT sensor demonstrated a sensitivity of 20%, a recovery of 37%, and high degradation in the air. The IDT-BT:ABN achieved a sensitivity of 17.53% in 100 ppm NH_3_ and 100% recovery in N_2_ and the air, and it reached the most extended lifetime of 84 days in the air. Finally, the IDT-BT:TCNQ sensor achieved a 23.14% sensitivity at 10 ppm NH_3_ gas and a 100% recovery in N_2_ and air, characteristics that were maintained for over 16 days and after 1200 bending cycles. Thus, IDT-BT:TCNQ OFET sensors achieved the best sensing performance and air stability. Greater reliability and air stability with the ABN and TCNQ additives were achieved due to the replacement of moisture present in the IDT-BT films generated from the solvents used during film deposition that can remain even after the annealing steps. In examinations regarding specificity, the sensors did not yield a distinct response in the presence of CO, CO_2_, and NO_2_, apart from NO [159].

SWCNTs acting as a gas-sensitive semiconducting channel in SWCNT-FETs have also been demonstrated [37,160]. As described previously, detection is primarily attributed to modifying the Schottky barrier at the SWCNT/electrode junction or doping the SWCNT [37,160]. SWCNT-FETs have enhanced reactivity to electron-donor gases, such as NO_2_ [160]. Such gas-sensing SWCNT-FETs have been manufactured on flexible plastic substrates for wearable applications [160]. Controlling the number of SWCNTs and, thus, their density, as well as their length, and the removal of metallic tubes by applying selective electrical breakdown, allow control of the semiconducting properties of the channel. Several isolated individual SWCNTs connected between source and drain interdigitated electrodes lead to an enhanced response when compared to using a single SWCNT (limited NO_2_ active site) or large randomly deposited dense networks (junctions between SWCNTs and or metallic tubes hinder or suppress the effect of the Schottky barrier modulation). Such devices can have on/off current ratio of 10^5^ [37].

NO_2_ gas sensing leads to measurable changes in the depletion-mode current and the threshold voltage. The latter should not be used, as trapped charges may modify it. An exponential relationship with a second power law relationship in a log–log plot has been found. Adsorption of NO_2_ leads to an increase in the work function of the Pd electrodes and, thus, a reduction in the barrier height to hole carriers, leading to an increase in the off current as a function of NO_2_ concentration [37]. Characterizing the transfer characteristics of such devices can be challenging since the channel current can vary within the range of 10^−12^ to 10^−3^ A [160].

In addition, looking at the SWCNT-FET as a standalone device is challenging since many device characteristics can change simultaneously, such as the shape or slope of the transfer characteristics, the threshold voltage, the drain current, and the on/off currents. It is, thus, a tedious procedure to monitor and interpret all these parameters. This comes with challenges in using these devices as a standalone element, requiring large power-hungry implementations. To address this, the gas-sensing transistor was implemented as part of an electronic circuit, an inverter, and an elementary digital circuit [160]. In this way, the readout implementation and interpretation of the results are greatly simplified. This inverter combined two SWCNT-FETs, a p-type SWCNT-FET sensing device and an n-type SWCNT-FET, resulting in a complementary structure (CMOS), and, thus, the output to be monitored is now a voltage rather than a current that was sensitive to NO_2_. Detection was, therefore, performed by monitoring the change in the amplitude of the square wave output of the inverter. The n-type device was rendered insensitive to NO_2_ within the 0.6 to 10 ppm range through the deposition of a thick PEI hermetic sealing film, so detection is only due to the p-type device. The sensing device is fully reversible. However, the time required for it to return to baseline values depends on the amount of NO_2_ it is exposed to; at 10 ppm, this is 5 h. This can be reduced either through heating or UV exposure. As the gas concentration increased, a decrease in the output square wave amplitude was observed. The sensitivity and LOD of the inverter depended on the supply voltage with an optimum of 4 V leading to an LOD of 0.57 ppm and the smallest error, and similar behaviors for 6 V. Due to the higher sensitivity at these supply voltages, a 2 V supply should be used for high gas concentrations as the other examined supplies would saturate [160].

Another recently proposed approach involves the use of MoS_2_. This atomic layer material can take several forms based on its processing. Chemical exfoliation to isolate MoS_2_ through lithium intercalation is one approach to obtaining this auspicious monolayer material. Charge transfer from lithium-ion to monolayer MoS_2_ leads to a phase transition from a direct bandgap semiconducting trigonal prismatic (2H) to a metastable conducting octahedral (1T) co-ordination [161]. However, 1T-MoS_2_ is thermodynamically unstable and transforms into the 2H form at elevated temperatures. The metastable 1T phase has higher conductivity for charge transfer and adsorption energy for NO_x_, making it suitable for gas sensing. At the same time, it leads to smaller contact resistances in FET devices. The 2H has a large 1.8 V bandgap impeding gas adsorption, but it is thermodynamically more stable and has a chemically active surface and edge sites for catalytic activities [161].

Consequently, one approach could involve combining both phases into a 1T/2H MoS_2_ heterostructure to exploit the advantages of gas sensing of NO_2_. Gas-sensing behavior is highly dependent on the 1T/2H ratio, which has to be tuned via thermal annealing to a ratio of 0.67 that yields the best performance of a sensitivity of 25% at 2 ppm and a response time of 10 s [161]. Then, 300 nm of SiO_2_ served as the gate oxide, and a bottom gate top channel architecture was once again used to allow direct exposure of the channel to the gas. The Si substrate served as the device’s back gate, and the MoS_2_ was deposited between the interdigitated source/drain Au electrodes. Thermal annealing was used to convert 1T to 2H. Nevertheless, even at RT, there is a 2H phase present in the material (about 30%). Raising the temperature to 100 °C increases that to 61%, and it further increases to 96% at 300 °C. This p-type device arises from water and/or oxygen adsorption during the monolayer fabrication steps and through the heterostructure. Thermal annealing at 100 °C also improves the electrical contact between the monolayer and the electrodes by reducing its resistance. However, further temperature increases increase the device resistance due to the lower conductivity of the 2H phase. Thermal annealing at 100 °C also improves the on/off current ratio, improving the device performance (from 1.29 to 1.47). Sensor sensitivity is maximized at that annealing temperature, and the response times are improved (20 s at 2 ppm). At 25 ppb NO_2_, representing the LOD, more than 1% sensitivity is achieved through monitoring the channel current. The recovery time, however, reaches the worst performance compared to other annealing temperatures. At RT, the device demonstrated high selectivity to NO_2_ [161].

Large arrays of MoS_2_-based FETs have also been demonstrated [162]_._ In contrast to chemical exfoliation, which has poor reproducibility and leads to films of poor uniformity, radio frequency magnetron sputtering was used with a subsequent step of thermal sulfurization for the MoS_2_ layer of the transistors. In the presence of NO_2_ at RT, the device exhibited a reduction in mobility and a positive threshold voltage shift; the channel current is reduced with increasing NO_2_ concentration due to the decrease in the conductivity of the semiconducting channel. A W/L of 20 μm/7 μm was used, and the devices demonstrated a threshold voltage of 13.5 V; thus, the V_GS_ of 0 V pushed the device into the subthreshold operating regime, where MoS_2_ FETs typically exhibit a higher response. V_DS_ was kept at 1 V. An on/off current ratio of 2 × 10^6^ was also achieved. In addition, a response of 0.5% was achieved for 1 ppm NO_2_, representing this device’s LOD, which also demonstrated rapid recovery and good reproducibility when comparing the response of nine identical devices. Measurements up to 256 ppm were demonstrated. This approach was used to create large active-matrix arrays of gas-sensing FETs with 7 × 6 pixels, each being individually addressed with a MoS_2_ switch. A local bottom gate separated from the back gate was used to control the devices fabricated with an 80 nm-thick aluminum oxide (Al_2_O_3_) gate dielectric. An encapsulating 20 nm thick SiO_2_ dielectric layer was used to insulate and protect the MoS_2_ channel of the matrix switches to prevent them from being exposed to NO_2_ and, thus, also change [162].

Hybrid MoO_x_/graphene FETs have also been proposed, exploiting shadow mask techniques as a low-cost manufacturing approach. NH_3_ sensing at 12 ppm was demonstrated with such FETs [163]. Graphene channels of 2 mm × 1.5 mm were realized with 3 nm Mo layers covering the graphene by different percentages from 25% to 100% in 25% steps. Oxidation of Mo took place naturally in the air. Ohmic contact was formed between graphene and S/D contacts, and a p-type behavior was achieved with all devices; this is maintained through the range of applied voltages from −75 to 100 V. Increasing the graphene coverage by MoO_x_ increases the relative change in the channel current when the devices are exposed to a set NO_2_ concentration while also affecting their recovery. The difference is more pronounced and faster at higher applied voltages and the better the recovery. No heating or illumination is required for RT sensing and recovery. A −18.1% response, recovery time of 356 s, repeatability of 12 cycles at 12 ppm after 3 min exposure to NO_2_, a LOD of 310 ppb, a sensitivity of 0.1% ppm^−1^, and complete recovery, all at RT, were achieved with the 100% covered device. After a 55-month air exposure period, sensing response decreased from −18.1% to −3.52%, but responses were still observable. Humidity also seems to affect the response and recovery time of the device [163].

### 4.5. Chemiresistors

Chemiresistive sensing is the most popular sensing approach. The use of chemiresistors has been accelerated in recent decades due to their relatively suitable amenability for inexpensive portable devices. Gas-sensing approaches achieving higher accuracy are typically more expensive and complicated and may even be unsuitable for wearable applications. The mechanism of gas response is based on the change in surface resistance due to gas interaction [164]. In most printed gas sensors, chemiresistive structures are employed to detect trace gas amounts due to their response principles to gas analytes through changes in channel conductance. The structure of a chemiresistor contains two pairs of electrodes linked together by a film of sensing materials, e.g., metals or semiconductors. Interdigitated electrodes (IDEs) are often employed to boost the sensing response. A known constant current or potential can be applied across electrode pairs. The resulting voltage or current is then measured upon exposure to gaseous analytes to obtain the film’s resistance through Ohm’s law, which is a function of a target gas’s concentration [90].

The MOS family, including materials such as SnO_2_, ZnO, In_2_O_3_, TiO_2_, and WO_3_, is frequently used in chemiresistive gas sensors. While CeO_2_, Fe_2_O_3_, CdO, and CuO are also employed in various sensor applications, their primary use in chemiresistive gas sensors is often as catalytic additives rather than primary sensing materials.

These materials can be deposited using a wide variety of manufacturing methods, such as chemical vapor deposition (CVD), hydrothermal growth, co-precipitation, thermal evaporation, the sol–gel method, pulsed laser deposition (PLD), radio frequency (RF), and direct current (DC) sputtering. The operational sensing principle of MOSs (based on the oxygen adsorption and desorption model) is better understood than with other, more complex materials. At the same time, they also typically offer a higher sensitivity and faster response and recovery times. MOSs exhibits a change in resistance when exposed to a target gas, arising from oxidative interactions with the negatively charged adsorbed oxygen [128]. Whether the MOS is n-type or p-type will define the polarity of this change. Under normal atmospheric conditions, atmospheric O_2_ is adsorbed or chemisorbed on the material’s surface. An electron-depletion zone is formed due to the extraction of electrons from the conduction band arising from the adsorption of O_2_. Consequently, in n-type material, reducing agents will release back electrons to the conduction band, decreasing electrical resistance. On the other hand, in a p-type material (e.g., CoO and NiO), O_2_ adsorption results in the formation of a hole accumulation region. Exposure to reducing analytes will then lead to an increase in resistance. The opposite occurs when the material is in the presence of oxidative agents [128].

In summary, n-type materials in the presence of an oxidizing agent will demonstrate a resistance increase, while, in the presence of a reducing gas, they will demonstrate a resistance decrease; p-type materials in the presence of an oxidizing agent will demonstrate a resistance decrease, while, with a reducing agent, they will demonstrate a resistance increase. Oxidizing gases act as electron acceptors while reducing gases act as donors. Between the two types, n-type MOSs (such as materials based on SnO_2_, In_2_O_3_, ZnO, TiO_2_, and WO_3_) are more common, possibly because they lead to stronger responses. A drawback of MOSs is the requirement for operation in high temperatures (e.g., 100 to 300 °C). The adsorbed oxygen requires high temperatures to become an oxygen anion, so it can migrate and remove electrons, thus changing the material’s conductivity. To improve the performance of chemiresistive sensors, the types of sensing materials employed, and the device fabrication methods should be developed via novel strategies [11]. Using photoexcitation or metal−organic frameworks and noble metals or organic-coated MOSs can reduce the required operational temperature and modify the sensing mechanism. The MOS crystal structure, grain size, porosity, the existence of catalysts, the manufacturing approach, the sensing film thickness, the nano-structuring, and the use of nanomaterials of the material can also influence sensitivity, reaction rate, selectivity, and operational temperature [128,165].

Another approach is using two or more metal oxides to form a heterojunction interface. For example, amorphous materials synthesized in RT with a higher carbon content, or a NW-structure have shown improved performance. Some examples include nanocrystalline Mg_0.5_Zn_0.5_Fe_2_O_4_ for N_2_O, graphene−CeO_2_ for CO, MWCNTs/WO_3_ for SO_2_, In_2_O_3_ nanocubes, Ti_3_C_2_T_x_ MXene composites for methanol sensing, photoactivated MOS for NO_2_, CuFe_2_O_4_ for ethanol, ZnSnO_3_ for H_2_S, and p-type Na_0.44_MnO_2_ nanoribbons for ethanol [128]. Another example to detect toxic amounts of NO_2_ is a chemiresistive gas sensor based on a polyethyleneimine (PEI)-functionalized SWCNTs thin film on a SiO_2_ substrate, with improved selectivity and sensitivity towards strong electron-rappeler atoms [166]. A repeatable response with a sensitivity of 37% was observed for the modified SWCNTs compared to the bare ones in the studied concentration range. SnO_2_ NW-based stripe (1–100 μm stripes) chemoresistors grown through CVD on micro-hotplates for CO and NO_2_ detection with a gas response increase by a factor of 500 [167]. Graphene-based sensors realized by drop-casting graphene-based nanomaterial solutions on silver interdigitated electrodes (7 mm × 13.4 mm overall area composed of 7 + 7 digits with a width of 210 μm) were able to detect several gases (NH_3_, (CH_3_)_2_CO, C_2_H_6_O, (CH_3_)_2_CHOH), and NaOCl) in the ppb range [168]. Five different inks were each dispensed on top of the electrodes. These were a graphene dispersion (1 mg/mL in DMF), graphene nanoplatelet (1 mg/mL, water dispersion), Fe_3_O_4_/graphene nanocomposite (10 mg/mL acetone dispersion, Fe3O4 NPs size: 5–25 nm), CoPt/graphene nanocomposite (10 mg/mL acetone dispersion, CoPt NPs size: 2–5 nm), and TiO_2_/graphene nanocomposite (10 mg/mL, acetone dispersion, TiO_2_ NPs size: 10–40 nm). Principal component analysis of the sensor responses allows discrimination of the different gases, while linear discriminant analysis demonstrated the ability of the sensor array to recognize unknown data with a high accuracy.

### 4.6. Self-Powered Triboelectric Gas Sensors

Triboelectric nanogenerator-based (TENG) self-powered gas sensors have gained interest as they enable the development of portable gas sensors that do not require an external power source. These sensors require no maintenance and are able to be miniaturized and portable [169,170]. Self-powered active gas sensors function simultaneously as an energy source and sensor. The TENG-based self-powered wearable gas sensors response is due to changing the TENG output as the triboelectric charge density varies upon gas exposure [171]. Since triboelectricity involves temporary electrostatic charge, which is a common phenomenon in most materials, the use of various material options is possible. Triboelectricity is simply a friction-generated static electrical charge [172]. Metal oxide semiconductors and conducting polymers are commonly used as sensing materials in the fabrication of TENGs [173].

Self-powered high-performance MXene-based flexible wearable sensors driven by triboelectric–electromagnetic nanogenerators are developed as multifunctional detection systems for gas and movement monitoring [174]. The Ti_3_C_2_T_x_ MXene/Ag-based TENGs fabricated by microelectronic printer and electrospinning devices performed successfully on the finger and knee with high selectivity to ethanol with a long-term stability up to 30 days. The sensor response to ethanol was also ~25 higher than that of chemiresistors [174].

To obtain fully flexible wearable gas-monitoring TENGs, a good combination of high performance and miniaturized circuits is required, which makes it challenging to develop with low power consumption in environmental gas measurements. To overcome this challenge, a wearable rechargeable and self-powered polyacrylamide hydrogel patch has recently been developed for remote wireless detection of NO_2_ with remarkable performance. With an ultrahigh selectivity and sensitivity in high humidity and at subzero temperatures, this sensor revealed outstanding performance (for sub-ppm) compared to existing self-powered NO_2_ sensors [175].

Due to the importance of miniaturization in the development of self-powered gas sensors and in an effort to achieve real-time monitoring, an RT high-performance self-powered TENG sensor was developed for the detection of formaldehyde [176]. A surface modification strategy was applied to exploit the functionalized triboelectric layer as the active sensing layer followed by incorporation of the seed layer of the Ag electrode. As the first self-powered TENG-based formaldehyde sensor, superior selectivity, and ultra-fast response time of 5 s to a sub-ppm level were achieved, which are superior to those of state-of-the-art formaldehyde sensors [176]. In summary, the research on self-powered TENG sensors for environmental gas monitoring is still in its early stage, and the selection of sensing materials, device stability, device design, and other aspects offer significant scope for new developments.

## 5. Environmental Gaseous Pollutants Monitoring

This section discusses several critical gaseous contaminants, highlighting their importance and effect on humanity and the planet.

### 5.1. Ammonia (NH_3_)

NH_3_ is a dangerous gaseous chemical with serious health consequences. Exposure to excessive amounts of NH_3_ can result in life-threatening effects. It is an odorless, water-soluble, and poisonous gas that can pollute the environment and cause severe lung conditions [177]. It has a high solubility in water-rich environments and contact with its vapors causes acute irritation of the eyes, mucous membranes, and respiratory system. Despite the essential industrial role of NH_3_ in petrochemical, plastics, textiles, explosives, and nitric acid manufacture, it is one of the most hazardous gaseous pollutants. The USA National Institute for Occupational Safety and Health has defined legal safety limits for NH_3_ exposure, with a concentration limit of 25 ppm for long-term exposure of 8 h and 35 ppm for short-term exposure of 15 min. However, prolonged exposure to higher concentrations of NH_3_ gas can cause significant injuries or even death [178]. Thus, it is vital to identify low levels of this gas by providing sensors with superior precision and performance, specifically in small dimensions for environmental purposes. Furthermore, NH_3_ is reported as a general marker in many illnesses (e.g., kidney failure), as well as an index for food quality monitoring (e.g., fish) since some bacteria produce it as part of their metabolic processes [179]. Various sensing approaches based on various nanomaterials have been proposed to detect NH_3_. Such examples have already been discussed in previous sections, and a few more examples are highlighted here. Another example is a chemiresistive RT graphene quantum dot-based gas sensor that has shown a quick response to NH3 with a linear concentration range of 10 to 500 ppm [180]. After chemisorption of oxygen species in the air on the surface of the OH-modified sensing film, NH_3_ vapor molecules react with these oxygen species and transfer electrons to the n-type sensing surface. This electron transfer decreases the resistance by increasing the density of the electron carrier. Here, edge functionalization of graphene quantum dots with hydroxyl groups by drop casting improves the sensor’s performance with high selectivity to NH_3_ gas [180].

Three-dimensional maze-like graphene nano-sheets with improved structure and thickness were prepared from two-dimensional graphene nanosheets by plasma-enhanced CVD and applied for the chemiresistive RT detection of NH_3_ gas. Due to H_2_ plasma etching, the surface area and the number of defect sites were enhanced, which led to an excellent performance of this platform with small-sized crystalline sensing nanomaterials [181]. Trace amounts of NH_3_ gas (10 ppm) were also measured at 40 °C on a polyaniline nanofiber/SWCNT composite. Compared to pristine SWCNT-based sensors, higher repeatability and long-term stability were observed for the chemiresistive sensor [182]. The importance of the functionalization of GO sheets in the sensing performance of NH_3_ sensors has also been investigated. Meta-toluic acid was modified on GO thin films via esterification, leading to a lower sensor resistance from RT to 600 °C than bare GO sheets. Sufficient sensor response of 32.7% and a response time of 10 s were obtained for 100 ppm NH_3_ gas [183]. The application of non-covalent functionalized pyrrole/phthalocyanine with MWCNTs has also been verified in ultrafast detection (11.7 s) of NH_3_ gas with a recovery time of 91.8 s. The improved chemiresistive sensing performance of the platform to 50 ppm NH_3_ gas at RT was ascribed to the synergistic optimization and recombination of pyrrole, phthalocyanine, and MWCNT [184]. The performance of ZnO-based chemiresistive sensors towards NH_3_ has been efficiently improved by MWCNT decoration compared to bare ZnO sensors. As the porous space between ZnO nanoparticles is filled with CNTs, the roughness of the available contact surface area for gas adsorption increases, consequently improving the sensor’s gas detection capability. High and stable selectivity and sensitivity of the sensor to NH_3_ in the presence of CH_4_ and CO were also verified at low concentrations of NH_3_ (10 and 20 ppm). The sensor architecture and sensing mechanism, as well as the experimental setup used, are illustrated in Figure 8. The sensor achieved a response of 1.022, a response time of 13.687 s, and a recovery time of 107.109 s [185].

Tunable diode laser absorption spectroscopy is well-known as a promising fast-response optical gas-sensing technique with sufficient selectivity and sensitivity. In a recent survey, the effect of three different carbon-based nanosensing materials, including CNTs, GO, and reduced graphene oxide (rGO), coated on a quartz tuning fork (i.e., the detector of the system), was studied by measuring trace amounts of NH_3_ gas. This low-cost sensing device revealed the minimum LOD of 7.27 ppm for NH_3_ gas at optimized conditions for the rGO-coated sensor [186].

As one of the best organic n-type semiconductors, non-toxic sulfur-containing C_60_ fullerene derivatives have been applied as green gas-sensing materials in organic field-effect transistors for NH_3_ detection. This rather sensitive device could respond to 1 ppm NH_3_ after a 7% increase in the source-drain current. The response and recovery time of the sensor were ~30 and 120 s, respectively. This simple, environmentally friendly approach can also be designed and developed to fabricate flexible electrical circuits [187]. In a different approach, a sensor was made up of the organic semiconductor poly(3,4-ethylene dioxythiophene): poly(styrene sulfonate) (PEDOT: PSS), electrochemically deposited iridium oxide particles (IrO_x_), and an agarose hydrogel sheet that aids the electrochemical sensing mechanism of the proposed device [188]. The working principle is based on the potentiometric response of IrO_x_, which is embedded within the organic semiconductor and responds to local pH variations in the hydrogel, modulating the doping state of the semiconductor. This is a significant departure from the majority of current NH_3_ gas sensors. The hydrogel interface is the most critical aspect of the sensor construction for achieving a reversible and selective response. This is because it is here that the gaseous analyte dissolves and absorbs reversibly, causing pH shifts. Its composition, acid-base characteristics, and shape were all finely tuned to achieve self-healing properties, the desired porosity, substrate adherence, and humidity stability.

Furthermore, the material’s self-healing characteristics were achieved by finely tuning the material’s acid–base properties. On a flexible plastic foil, the PEDOT: PSS/IrO_x_/hydrogel sensor was successfully constructed and tested in a wearable configuration with wireless communication to a smartphone. The sensor could be implemented on a flexible plastic foil due to the dependability of the analytical response, the simple two-terminal structure, and the low power consumption of 0.1 mW. With a sensitivity of 60 ± 8 μA decade^−1^ in a wide concentration range of 17–7899 ppm, the wearable sensor displayed robustness to mechanical deformations and good analytical results [188].

### 5.2. Nitric Oxide (NO)

One of the most hazardous gaseous pollutants is nitric oxide (NO). It contributes to acid rain deposition and participates in ozone layer depletion. It is formed by lightning in thunderstorms and combustion systems, such as car engines and various industrial processes. Exposure to even infinitesimal concentrations (ppb-ppm) of NO can significantly impact the environment and human health. NO is a colorless gas, and, along with CO, it is known as a silent killer and irritates the skin, eyes, and mucous membranes. At the same time, it also increases the risks of respiratory and cardiopulmonary diseases [189,190]. The USA Occupational Safety and Health Administration (OSHA) has defined a safety limit at the workplace of 25 ppm over an eight-hour workday; above this limit, it reduces the oxygen-carrying capacity of hemoglobin. Above 100 ppm, it becomes immediately dangerous. NO is a signaling molecule in several physiological and pathological processes in humans and other mammals. It can diffuse freely across cell membranes; it is, for example, a cardiovascular signaling molecule. It increases vasodilation and blood flow while playing a strong role in angiogenesis. In almost all types of organisms, it is a byproduct. As a diagnostic biomarker, its measurement in exhaled breath can be exploited for diagnosing digestive diseases and inflammation in the stomach (gastritis, hepatitis, and colitis), liver transplant rejection, cystic fibrosis, encephalopathy, helicobacter pylon digestive cancer, and respiratory diseases, such as chronic obstructive pulmonary disease (COPD) and asthma [191]. Hence, converting NO into harmless nitrogen species through catalytic reactions has gained attention [192,193]. Therefore, the production and development of gas sensors has become a significant concern for researchers and scientists. Carbon structures such as fullerene (C_60_), graphene, graphene nanoribbons, graphene quantum dots, and CNTs have been proposed for sensing CO, NO, and NH_3_ gases [194]. Although various allotropes of carbon, such as armchair graphene nanoribbons, zigzag graphene nanoribbons, graphene quantum dots, and graphene sheets, have promising capabilities concerning gas adsorption, similar to graphene and CNTs, it has been shown that functionalization of carbon allotropes (doping) with Sc, Ti, V, Cr, and Al remarkably improves their sensing ability [195]. In one such example, an SWNT-based gas sensor that operates at RT was proposed [191]. The SWCNTs were formed between Au electrodes and were functionalized with en-APTAS. A chemiresistive change of 28.64% with a 100-ppb concentration of NO gas was achieved, presenting no sensitivity to CO and VOCs. An SWCNT dispersion was spray-coated on a semiconductive wafer that was first oxygen-plasma-treated and then coated with APTES. The device was then coated with en-APTAS and then washed in ethanol. Sputtering Au through a shadow mask enabled the patterning of interdigitated electrodes above the sensing film. In a different study, graphitic carbon-doped SnO_2_ nanosheet-wrapped tubes were proposed for chemiresistive ppb-level RT NO sensing [196]. This approach enabled a response of 256.3 towards 1 ppm NO at 50 °C, a short recovery time of 42 s, an LOD of 50 ppb, and long-term stability. The chemiresistive sensing film was manufactured using wooden hydrangea petals as a biotemplate.

In another chemiresistive approach, iron oxide nanorods prepared by oxidizing a stainless-steel mesh by a wet oxidation technique were demonstrated [197]. This sensor operated at 250 °C, where, with a 1 mV applied voltage, NO sensing between 0.5 ppm and 2.75 ppm was demonstrated. The device response was found to be between 1.49% for 0.5 ppm and 15.2% for 2.75 ppm, with a response and recovery time of less than 10 s. Another exciting approach involved pulsed laser ablation in a liquid environment and subsequent thermal annealing to synthesize CuO ligand-free nanoparticles [198]. These were then drop-casted on an interdigitated electrode structure on an alumina substrate. The produced chemiresistive sensor demonstrated a fast (response time of 2.5 s, recovery time of 278 s) and selective NO sensor, with a sensitivity of 0.0044 R/Ro ppm^−1^ and an LOD of 10 ppm, when operating at 50 °C.

### 5.3. Nitrous Oxide (N_2_O)

N_2_O is a toxic gas that finds widespread use in medicine, specifically in surgery and dentistry, due to its anesthetic and analgesic properties. It is commonly known as ‘laughing gas’ due to its euphoric effects upon inhaling it, which can cause slight hallucinations. As a result, it is also used as a recreational substance. Nevertheless, it is neurotoxic and can cause irreversible neurological damage, while it has also been associated with DNA damage. It is a non-flammable and colorless gas with a sweet smell and taste; it is stable at RT and easy to store. At elevated temperatures, it is a powerful oxidizer, and it is thus used in motor racing and rockets as a propellant. It is also extensively used as a food additive as an aerosol spray propellant for cooking sprays and aerosol whipped cream. N_2_O is known to be a primary ozone scavenger in the stratosphere, contributing significantly to global warming, especially since it is the third most important long-lived greenhouse gas. CO and N_2_O are critical greenhouse gases that play a crucial role in air pollution, with N_2_O being ~300 times more damaging and having a longer lifetime in the atmosphere than CO_2_ [128].

N_2_O is primarily generated in industrialized agriculture, animal farms, and through excessive use of synthetic fertilizers [199]. Its concentration in the atmosphere increases annually by approximately 1 ppb, and, in 2020, it reached a value of 333 ppb. About 40% of emissions between 2006 and 2016 are estimated to be due to human activity, primarily from emerging economies’ industrial and agricultural sectors. The above highlight the necessity for developing reliable methods to monitor N_2_O. Currently, N_2_O is analyzed using several methods, such as infrared laser spectroscopy using QCL, gas chromatography, (MEMS)-based approaches, NDIR adsorption, and chemiresistive and electrochemical methods (e.g., amperometry) [128]. Gas chromatography, however, requires the removal of moisture and particles and dilution steps that increase measurement uncertainty, making this approach unsuitable for real-time measurements of N_2_O [200]. There are several commercial NDIR N_2_O gas sensors. N_2_O has its highest IR active absorption wavelength at 4.5 μm, the lowest possible perturbation energy required to identify it. As a result, an LED can be used. Nevertheless, the absorption wavelengths of NO at 5.3 μm and CO at 4.3 μm lead to interference and overlap in the recorded spectra, thus leading to ambiguities in the measurement. In another optical approach, PAS has been combined with a 4.53 μm wavelength QCL to enhance its sensitivity and to achieve a minimum detection limit of 28 ppbv in 1 s and a measurement precision of 34 ppbv [201]. The performance was improved by humidifying the gas sample and reflecting the transmitted laser light back to the PAS cell. Kalman adaptive filtering improved measurement precision by 2.3 times without affecting the time resolution of the measurement. In a different approach, Zifarelli et al. achieved a minimum LOD of 6, 7, and 70 ppm at 100 ms of integration time for CO, N_2_O, and CO_2_, respectively, via the implementation of a quartz-enhanced photoacoustic spectroscopy (QEPAS) system [202].

As discussed previously, electrochemical amperometric sensors require a polarization potential for reducing a target gas and measuring the resulting current related to its concentration. Different gases have different redox potentials, and, as mentioned before, O_2_ and CO_2_ have less negative reduction potentials than N_2_O, while N_2_ has higher. Moreover, N_2_O has a potential higher than that of dioxygen [200,203]. Consequently, various methods must be applied to improve specificity. Several of these were discussed earlier and included using multiple compartments where different reduction potentials are applied or electrolytes that act as scavengers for the interferants. Different materials can be used for the sensing cathode of the electrochemical cell for N_2_O detection. These include silver and indium, the most common choice due to their catalytic activity toward N_2_O reduction. Other choices include palladium, tetraaminophthalocyanato cobalt (II), and metal porphyrins [128]. TCN (II)–KOH/rGO deposited by drop casting on carbon felt has demonstrated a capability of detecting N_2_O at different concentrations and increased gas flow rate compared to the GC sensors [203]. The sensor achieved a wide sensing range from 1 ppm to 16 ppm and an LOD of 1 ppm. The electrochemical cell was composed of a Pt counter electrode and an Ag/AgCl reference electrode that was positioned within a 0.5 M H_2_SO_4_ solution. The carbon-felt-based working electrode was placed in a different chamber separated by a Nafion membrane where the gas was introduced. The above discussion considered direct detection of N_2_O; however, indirect detection through the reduction of O_2_ produced through the decomposition of N_2_O at temperatures between 500–700 °C is another approach [128]. This can be facilitated by using a solid-state electrolyte such as yttrium-stabilized zirconium, which is not a suitable approach for wearable applications due to the very high temperature required.

### 5.4. Nitrogen Dioxide (NO_2_)

NO_2_ is a highly reactive gas and one of the most prevalent atmospheric gaseous pollutants, ranking second among the hazardous pollutants produced by automobiles. It is mainly produced by fossil fuel combustion, automobile exhaust, power plant emissions, and industrial activities. It is considered one of the most hazardous environmental and biological pollutants. It can cause severe respiratory problems, such as asthma, lung inflammation, and increased susceptibility to respiratory infections, even at short-term exposures and low (ppm) concentration levels. Some non-respiratory system problems caused by NO_2_ exposure are diabetes, hypertension, and heart and cardiovascular diseases [204]. It is also a threat towards animals, plants, and the environment, and also one of the primary factors producing photochemical pollution and acid rain [4]. Therefore, it is of utmost importance to develop high-performance gas sensors that can precisely monitor concentrations of NO_2_ at the parts per billion (ppb) level. MOS gas sensors, such as those based on ZnO, SnO_2_, TiO_2_, In_2_O_3_, MoO_3_, and WO_3_, have garnered significant attention for their low cost, high sensitivity, and abundance [205,206,207,208]. MOSs commonly requires elevated temperatures, which enable them to achieve considerable sensing performance. However, this also results in higher power consumption, making their use impractical for portable applications like wearable devices.

On the other hand, two-dimensional (2D) metal sulfides have emerged as high-performance and power-saving materials for gas sensing but have received relatively little attention. However, doping of 2D metal sulfides has been shown to enhance gas interaction properties, particularly at RT. For example, Cheng et al. synthesized three-dimensional (3D) micro-combs from 2D N-doped In_2_S_3_ via hydrothermal synthesis, resulting in an RT reversible NO_2_ gas sensor [209]. They have shown that N-doping significantly enhances the electronic band structure of In_2_S_3_, which enables hybridization with NO_2_ molecular orbitals. In addition, gas sensors based on chemiresistive tin oxide (SnO_2_) have received widespread attention due to their high oxygen vacancies, intrinsic non-stoichiometry, broad bandgap, high charge carrier mobility, and high thermal and chemical stability [210]. However, their application, mainly in wearable devices, is restricted by elevated operating temperatures and high-power consumption [211,212]. Various strategies have been employed to address these issues, including doping, phase-controlled modifications, nanostructuring, and heterojunctions with metals and metal oxides. For example, a low-temperature SnO_2_ sensor has been developed by doping Ni into the SnO_2_ nanoparticle matrix, which yielded a high response at a low concentration level of 3 ppm NO_2_ at RT [213,214]. The utilization of different dopants, such as Pd, has allowed low detection concentrations (0.35–5 ppm) of NO_2_ at 100 °C. Additionally, researchers have demonstrated that doping SnO_2_ thin films with rare earth elements (Y and Er) produces a considerable amount of oxygen vacancies that intensify the adsorption of NO_2_ molecules. This results in an enhanced response at a lower gas concentration while also resolving the requirement of high operating temperatures, thus reducing power consumption [211,215,216].

### 5.5. Carbon Monoxide (CO)

CO is one of six primary air pollutants the World Health Organization (WHO) identified. CO is a colorless and odorless gas that is less dense than air. It is produced by the incomplete combustion of fossil fuels, gas from automobiles, aeroplanes, natural gas emissions, coal mines, fires, industrial waste, sewage leaks, solid fuel appliances, water heaters, open flames, and other biological activities [217]. This pollutant harms all natural resources, including the air, water, earth, and all living organisms. Therefore, several attempts have been made to reduce and prevent exposure to harmful pollutants. As CO is an industrial-based chemical gas that is widely used in fuel production and other applications, the WHO has set an exposure limit of 9 ppm for CO for 8 h. Mild exposure to CO can cause breathing problems, headaches, nausea, dizziness, and fatigue [218,219]. However, more deadly symptoms are triggered when a person is exposed to a greater CO concentration [220]. When exposed to CO gas, CO molecules displace the oxygen in the body, leading to poisoning. CO rise in blood carboxyhemoglobin (CO-Hb) is the best biomarker to characterize CO poisoning [221].

On the other hand, the most probable direct cause of mortality is the buildup of CO in tissues. Apart from gas chromatography, there is no dependable technology that can accurately assess the amount of CO present in tissues. A synthetic supramolecular molecule called hemoCD1, consisting of an iron(II) porphyrin and a cyclodextrin dimer, was utilized as a reagent in a simple colorimetric test to measure the concentration of CO in biological samples [221]. The test was validated in various organ tissues obtained from rats under standard settings and after exposure to CO. The kinetic profile of CO in blood and tissues after CO treatment revealed that CO buildup in tissues was avoided by circulating Hb, suggesting a protective function of Hb in CO intoxication. This finding was based on the kinetic profile of CO in blood and tissues after CO treatment. According to the findings, the buildup of CO-Hb and CO in tissues in vivo is accelerated when CO gas is inhaled. Even while air or O_2_ ventilation can return CO-Hb levels to baseline levels, it is challenging to eliminate CO from tissues, particularly in the brain. This is especially the case for accumulation. Furthermore, scientists have demonstrated how the CO content in tissues reaches a plateau rapidly during CO exposure, despite CO-Hb levels in the blood continuing to increase over time. This finding raises significant concerns regarding the role of Hb as a potential molecular shield that safeguards tissues from accumulating harmful levels of CO. The data on CO concentration in tissues and organs were obtained using a novel and simplified approach for CO detection based on hemoCD1, which was extensively examined and validated in the study [221].

Many recent studies have focused on smart wearable devices that can be attached to the skin or injected into the body to monitor various parameters and provide quick detection. A cotton textile was obtained from a laboratory coat coated with graphene/ZnO in another study. These samples were cleaned using ultrasonication and dried before adhering to a glass microscope slide with adhesive polyimide tape. Graphene was deposited via dip coating. In the next phase, the graphene-coated textiles were modified with ZnO seed layers that were deposited using a solution-based technique. To make the ZnO seed solution, Zn(NO_3_)_2.6_H_2_O was dissolved in 2-propanol, and then triethylamine (C6H_15_N) was added as a sol stabilizer, and the reaction was completed. Using the chemical bath deposition (CBD) process, ZnO nanorods were generated on the ZnO-seeded graphene-coated textiles [220]. This chemosresistive sensor was highly sensitive and selective at RT towards CO gas down to 10 ppm, with the shortest sensor response and recovery times achieved being 280 s and 45 s, respectively. When comparing graphene and graphene/ZnO sensors, a 40% enhancement in sensing response was achieved, supporting the superiority of the proposed hybrid material approach as a flexible CO chemiresistive sensor. As discussed previously, FETs are also utilized for gas sensing. A FET sensor using a Zn-doped In_2_O_3_ single nanowire that can detect gas concentrations ranging from 1 to 5 ppm (CO, NO, and NO_2_) has been demonstrated [219]. The Zn-doped In_2_O_3_ FET sensor was found to be the most sensitive to CO gas when tested at ambient temperature. The sensing response at ambient temperature is linked to the development of defects in the nanowire and a change in its conductivity due to Zn doping. The performance of the Zn-doped In_2_O_3_ nanowire sensor was about three times better than that of the undoped nanowire sensor [219]. In a different study, researchers reported a 20% improvement in the detection performance of CO when operating a SnO_2_ NW chemiresistive gas sensor at 103 °C [222]. This was achieved by using a wireless microheater that was optimized through the appropriate design of its resistive heating element and inductive antenna. The optimal sensing temperature varied depending on the type of gas [222].

### 5.6. Carbon Dioxide (CO_2_)

All aerobic organisms release CO_2_ as a waste product when they metabolize organic compounds to produce energy by respiration. CO_2_ is also released from organic materials when they decay or combust (e.g., forest fires or by humans for heating and other purposes). On the other hand, CO_2_ is utilized by plants as part of their metabolism for photosynthesis, releasing O_2_. Consequently, CO_2_ is necessary for the survival of life on earth. CO_2_ is also released into the environment from the extraction and burning of fossil fuels (such as coal, oil, and natural gas) and natural processes like volcanic eruptions. Other significant anthropogenic sources include biomass burning, deforestation, and cement production. Transportation, industry, and agriculture are the primary sources of air pollution in cities. As of May 2022, the global average concentration of CO_2_ in the atmosphere is 421 ppm. Compared to 280 ppm before the mid-18th century, this is an increase of 50% since the start of the Industrial Revolution ^[31]^. This is well above the threshold of 350 ppm required to prevent irreversible climate change. This increase is due to human activity, mainly due to the burning of fossil fuels. In addition, CO_2_ absorbs and emits infrared radiation. As a result, it plays a significant role in influencing Earth’s surface temperature through the greenhouse effect, making CO_2_ a major contributor to global climate change. This phenomenon has been linked to an increase in severe weather conditions, such as storms, droughts, and wildfires worldwide [223].

CO_2_ is a colorless, odourless greenhouse gas [224]. It is an asphyxiant gas and not classified as toxic or harmful. Nevertheless, high levels of CO_2_ can have a significant impact on human health, causing everything from tiredness and loss of concentration (100–2000 ppm), drowsiness and a stuffy lung feeling (10,000 ppm), to death (>40,000 ppm), with concentrations of 70,000 to 100,000 ppm causing suffocation, even in the presence of sufficient oxygen. This initially manifests as dizziness, headache, visual and hearing dysfunction, and unconsciousness within a few minutes to an hour. As a result, detecting CO_2_ gas is crucial. More efficient building insulation can minimize the consequences of climate change, but over-insulated buildings may not be healthy. Poor ventilation can lead to low oxygen levels and a build-up of CO_2_. Even low levels of CO_2_ can severely impact health and productivity. There is, thus, an increasing demand for miniaturized and precise gas sensors that can be effectively employed in large-scale sensor networks to monitor greenhouse gas concentration patterns. These sensors must be capable of monitoring fluctuations in greenhouse gas concentrations [225]. As a result, manufacturers of CO_2_ sensor modules are seeing a rise in demand for smart indoor air quality monitors that can detect rising CO_2_ levels and inform the user or trigger a system reaction. Wearable CO_2_ sensors in smart homes and workplaces enable precise measurement of CO_2_ levels. A recent study took advantage of the widespread use of facemasks for combating the spread of the severe acute respiratory syndrome coronavirus (SARS-CoV-2) virus that caused the COVID-19 disease. A filtered face mask FFP2 has a dead space volume (DSV), the volume of non-exchanged gas between O_2_ and CO_2_. A batteryless wearable mask was proposed to determine CO_2_ concentration. Wearing facemasks makes breathing more challenging and increases DSV, resulting in higher CO_2_ levels. A sensing system was placed into a FFP2 mask to determine the amount of CO_2_ gas indoors. This opto-chemical sensor could detect a concentration of CO_2_ of 140 ppm with a response time below 1 s. The sensor had an 8 h operational time and could be operated by an application downloaded on a smartphone, as shown in Figure 9 [226].

Since the increasing concentration of CO_2_ contributes to the greenhouse gas effect, it leads to a temperature increase in the environment. CO_2_ also causes metal corrosion and decreases the heating value of methane (CH_4_) when combined with water in the atmosphere. In another study, researchers proposed using metamaterials to detect dangerous levels of gases. Metamaterials are artificially made structures with permittivity and permeability properties not found in naturally occurring materials. They have generated significant interest due to their impressive electromagnetic (EM) response. Constructing metamaterials enables us to alter the electromagnetic characteristics of materials to achieve more favorable optical properties. In recent years, researchers have conducted extensive studies on metamaterial-based electromagnetic (EM) absorbers that are promising for sensing applications [227]. One such metamaterial-based gas sensor was proposed for CO_2_ sensing [225]. A photonic CO_2_ gas sensor, based on infrared evanescent field absorption, could detect gas concentrations as low as 5000 ppm. Symptoms such as headaches, drowsiness, a lack of focus, a slightly elevated heart rate, and slight nausea may occur at this level. A desirable configuration for a metasurface can be achieved by depositing a periodic array of silicon nano-cylindrical meta-atoms (MAs) on a metal layer (Au). This arrangement offers the opportunity to function as a perfect narrowband absorber.

Moreover, the metasurface can operate as a gas sensor if a specific thin layer of a functional host material is deposited on its surface. Depositing a functional host material allowed researchers to investigate the gas-sensing capabilities of the metasurface-based absorber. To detect CO_2_ gas, a polymer called Poly hexamethylene biguanide (PHMB) was utilized [225]. This polymer can absorb CO_2_ gas. When CO_2_ molecules bind with the PHMB layer, it alters the polarizability of the layer, leading to a change in its refractive index. The functionalization layers’ refractive index is affected by variations in CO_2_ gas concentration in the surrounding environment. This change can be explained by a redistribution of electron density of the polymer repeating units resulting from the CO_2_ molecules’ binding. The proposed gas sensor has a maximum sensitivity of 17.3 pm/ppm and can detect CO_2_ gas concentrations ranging from 0 to 524 ppm. The described sensor architecture has the potential to identify other harmful gases by utilizing suitable functional host materials [225]. In another recent study, researchers worked on a transcutaneous blood gas analysis (non-invasive) method for health monitoring. COVID-19, sleep apnea, and COPD will be revolutionized by a small wearable device that can continuously monitor transcutaneous blood oxygen and CO_2_ [228]. A wearable wristband device that utilized a tiny nondispersive infrared (NDIR) sensor to continuously and non-invasively monitor transcutaneous blood CO_2_ was proposed.

Additionally, a hydrophobic layer with high CO_2_ permeability was created to reduce the interference caused by humidity on the NDIR CO_2_ sensor’s functionality. This NDIR sensor demonstrated a linear and precise response and low power consumption. As blood gas is humidified and water is released from the body through sweat or diffusion, this solution provides a low-cost and non-heating method for measuring transcutaneous CO_2_ on a wearable platform [228].

### 5.7. Hydrogen Sulfide (H_2_S)

H_2_S is a colorless, poisonous, highly flammable, corrosive, and toxic gas that is produced during the processing and breakdown of organic materials, e.g., from bacteria and industrial production activities related to fossil fuels, natural gas and petroleum production, and refineries. It inhibits cellular respiration and can rapidly damage the organs, and cause convolutions, breathing difficulties, and even death, affecting the nervous system. At low levels (~10 ppm), it causes eye irritation, nausea, cough, shortness of breath, fatigue, dizziness, and headache over time; 50–100 ppm can lead to eye damage. Consequently, it is classified as a hazardous air pollutant as it causes adverse health effects, making it a dangerous gas.

Moreover, the odor of H_2_S diminishes beyond 100 ppm, which is considered a potentially fatal concentration, due to damage in the olfactory nerve [97,229,230]. An amount above 320 ppm leads to pulmonary oedema, and 530 ppm leads to strong nervous system stimulation and potential loss of breathing. In contrast, 5 min exposure to 800 ppm is lethal for half the population. At the same time, H_2_S is a signaling molecule and a mediator for several disease states and healthy physiological processes. For example, overproduction of H_2_S is related to cancer and Down syndrome, while underproduction is associated with vascular disease. Thus, a sensing solution for rapidly detecting low H_2_S concentrations is required. Commercially available solutions typically provide an alarm to the user at 5–10 ppm and a second alarm at 15 ppm.

Electrospun nanofibers offer several advantages, such as rapid and effortless production, a highly scalable process, and lightweight materials with a large surface area. In a recent study focused on detecting H_2_S, the creation of flexible capacitive sensors via electrospinning was demonstrated using a zirconium-based metal-organic framework (MOF) called NO_2_-UiO-66 [229]. Researchers developed a three-step process to create an electrode using electrospun polyacrylonitrile (PAN) as the sensing layer coated with CNTs on both sides. Trifluoroacetic acid (TFA) was utilized to ensure the even dispersion of NO_2_-UiO-66 on PAN nanofibers and to slow the crystallization process. This step is critical to enhancing the material’s adsorption capacity. CNTs were chosen as the electrode due to their high conductivity and air permeation properties. CNTs also offer additional flexibility to the electrospun fiber, making it the ideal electrode for a wearable sensor. The designed electrospun sensor could detect H_2_S, even at an ultralow LOD of 10 ppb, and is suitable for use between 0 °C and 180 °C. Additionally, it demonstrated high reproducibility, even after one month of folding. These properties suggest that the designed material is promising for use in environments requiring sensitive and precise detection with wearable sensors, such as gas masks [229].

Another approach to H_2_S detection was taken by printing disposable sensor labels for colorimetric detection, among other air pollutants such as ammonia and formaldehyde [230]. The concept relies on detecting air pollutants through a color change, observable using either a smartphone camera or a stationary reader. The sensor is constructed of a low-cost combination of plastic and paper, with a gas-sensitive layer containing a copper (II) complex of the azo dye 1-(2-pyridylazo)-2-naphtol (H-PAN) (Cu-PAN). Detection of H_2_S occurs when S-2 reacts with the Cu-PAN complex, causing the color to change from purple to red, then yellow. Although the color change is reversible, the sensor will return to its initial purple state if the pollutant is not within the detectable range. However, the reverse reaction of the chelation of H-PAN with CuS is slower than the forward reaction. What sets this single-use printed pollutant sensor apart is its ability to overcome long-term stability issues typically encountered with similar sensors. In another paper, researchers demonstrated a sensor that utilizes Quick Response (QR) code technology to enable the easy identification of pollutants by the user. The slow reverse reaction of the sensing material also makes it suitable for dosimeter applications. This product can be utilized in a variety of industries, including but not limited to mining and crude oil, due to its use of low-cost and easily obtainable paper–plastic composites [230] (Figure 10a). More recently, an Ag nanofilm was used as a chemiresistive sensing film, exploiting the fact that, upon exposure to H_2_S, Ag loses electrons and converts into Ag_2_S [231]. The applied temperature and the H_2_S concentration define the speed of this reaction. A temperature sensor was patterned next to the gas sensor. The device was operated at 300 °C, achieved an LOD of 1.4 ppm, and consumed 90 mW of power. At the same time, it demonstrated high selectivity towards H_2_S and a long-term stability of 28 days when exposed to 100 ppm. At an applied temperature of ~520 °C, the sensor can be recovered; otherwise, the authors estimated that the sensors can be used 533 times before it is exhausted.

### 5.8. Sulfur Dioxide (SO_2_)

SO_2_ is a dangerous toxic pollutant produced by burning fossil fuels in places such as oil refineries, power stations, and industrial plants. It is also generated from volcanic eruptions and forest fires. It is a widely used food preservative, particularly with dried fruit, due to its antimicrobial properties and aptitude to prevent oxidation. It is extensively used in winemaking as an antibiotic and antioxidant since it protects wine from bacteria, and oxidation can spoil it. It is widely used for chlorinated wastewater treatment. It can be found in small concentrations in the atmosphere, in the region of 15 ppb. It is one of the six air pollutants monitored in the US, with a limit of 75 ppb for one hour of exposure. Endogenous SO_2_ is essential in regulating cardiac and blood vessel function, and its deficient metabolism can lead to cardiovascular problems. Overexposure to SO_2_ can cause severe health problems in the eyes, lungs, and throat. This pollutant harms human health, due to its high solubility in water.

SO_2_ molecules can accumulate in rivers, soil, and clouds, resulting in acid rain. Acid rain disrupts the natural balance of wildlife and can damage building materials, such as limestone, ecosystem stability, and agricultural production [232]. These adverse effects of SO_2_ have stimulated research in developing sensitive, low-cost, and wearable sensors.

One study on SO_2_ detection demonstrated an electrospun wearable and capacitive sensor with high efficiency and flexibility [233]. Researchers combined electrospun polyvinylidene fluoride (PVDF) with a zirconium-based MOF (UiO-66-NH_2_). The manufactured SO_2_ capacitive sensors achieved detection within an extensive range, from 1 ppm to 150 ppm at RT. The azanide group (NH_2_^ˉ^) is essential in this work. Azanide can be used as a surface functionalization group and convert UiO-66 into a chemically resistant sensor.

A study on the preparation and performance of SO_2_ sensors based on MOF-manufactured flexible sensors focused on the mechanical properties of the sensor [234]. The technology behind this sensor is similar to the previously discussed SO_2_ sensor [233]. The substrate was polyvinyl alcohol (PVA), and the sensor was based on MOF-5. PVA was loaded with MOF-5 crystals previously modified with azanide groups via electrostatic spinning. The resulting sensor was low-cost and showed fluorescent activity upon detection of SO_2_. The lightweight and flexibility of the final device allowed mobility and flexibility, which are the most critical requirements for wearable sensors. Although it is easy to carry, the material shows a tensile strength of 1.42 MPa. Moreover, it has been revealed that decreasing the fiber diameter enhances the thermal stability of the sensor, making it suitable for use across a wide range of temperatures. Additionally, PVA nanofibers are convertible, and the sensor’s fluorescent probe can be substituted with other sensor types. These material properties could broaden the sensor’s applications beyond SO_2_ sensing [233] (Figure 10b). In another study, an RT UV-light-assisted chemiresistive SO_2_ gas sensor based on PANI and Ag nanoparticle-commodified tin dioxide nanostructures (Ag/PANI/SnO2) was presented [235]. When exposed to 50 ppm of the target gas, the sensor exhibited a response of 20.1, a response time of 110 s, and a recovery time of 100 s; the latter was achieved due to the UV irradiation (365 nm).

### 5.9. Ozone (O_3_)

Ozone formation in the troposphere occurs through a reaction between nitrogen oxides and volatile organic compounds released into the atmosphere by solar radiation. Ozone has a negative impact on human health, as it can cause respiratory and cardiovascular diseases and can affect the central nervous system. Ozone pollution in urban areas is related to heat-related mortality during heat waves. Moreover, exposure to even low ozone levels, below those currently regulated, can have chemical and toxicological effects due to its potent oxidative properties, leading to cellular-level oxidative damage. Above 0.1 ppm, it causes damage to mucous and respiratory tissues in animals and in plant tissue. As specified by the WHO, 50 ppb is the O_3_ exposure threshold for 8 h [236]. At the same time, it also interferes with photosynthesis, affecting the growth of certain plant species. These detrimental effects of ozone have prompted research into its rapid and accurate detection, even at very low concentrations. In 2020, the USA Environmental Protection Agency (EPA) established the safe limit for ozone exposure at 0.09 ppm/h. In addition to its harmful impacts, ozone is also utilized for disinfecting medical environments owing to its oxidative properties. At high altitudes and in the so-called ozone layer, its high concentrations in the range of 2–8 ppm protect life on Earth by preventing dangerous and damaging UV irradiation from reaching the planet’s surface. It is important to note that catalytic decomposition with solid catalysts can be achieved with noble metals, such as Pt, Rh, or Pd and transition metals, including Mn, Co, Cu, Fe, Ni, or Ag. Alternatively, it can be decomposed with heat (a prolonged process below 250 °C) or under UV irradiation. However, given the potential toxicity of ozone in public spaces, there is a need for portable sensors to detect its presence [237,238].

Researchers have developed an ozone sensor utilizing a UV sensor [239]. The developed sensor uses an aluminum-coated substrate-integrated hollow waveguide (iHWG) and a portable spectrophotometer for real-time ozone monitoring. The significance of this study lies in the noteworthy reduction in the size of ozone sensor technology. As previously discussed, even ozone gas below the maximum level set by regulatory agencies can have adverse effects. The developed sensor could detect ozone at levels as low as 29 ppb. A mercury lamp was used to detect ozone because of its strong emission in the UV-C range and narrow absorption at 254 nm (i.e., the absorbance range of ozone). By reducing the cost and size of ozone sensors, this work opens the door for smaller, more cost-effective ozone sensors with higher efficiency and greater accuracy, even at lower ozone levels [239]. In a different study, a CuCo_2_O_4_ nanomaterial based O_3_ chemiresistive sensor operating at 90 °C was proposed, achieving a response to 1 ppm of 11 at a RH of 0%, which increases with increasing RH (e.g., 24 and 27 for RH 50% and 70%, respectively) [237].

### 5.10. Hydrogen Fluoride (HF)

Hydrogen fluoride (HF) is widely utilized in various industries, such as electronics manufacturing and metal cleaning, as well as in the production of various pharmaceuticals and other chemicals, such as fluoropolymers, chlorofluorocarbons, aluminum fluorides, and petrochemicals. Additionally, it is employed as a rust removal and car washing agent for domestic use. With a pH level of approximately 5.5, HF is classified as a weak acid that can dissolve lipids. Its significant solubility in lipids enables it to swiftly penetrate tissues when contacted or inhaled, resulting in poisoning upon ingestion, skin burns, and blindness by rapid destruction of the cornea. Exposure to it necessitates immediate medical attention. In severe cases, prolonged exposure to HF can cause hypocalcemia and hypomagnesemia due to its affinity for calcium and magnesium in the body.

In contrast, it can cause death due to irregular heart rates or pulmonary oedema [240]. Due to the toxic properties of HF mentioned above, regulating, and detecting its presence is crucial. OSHA has set the permissible exposure limit for HF at 3 ppm, with a short-term exposure limit of 6 ppm. However, exposure to HF would quickly exceed these established limits in case of a leak or spill. For example, the thermal decomposition of hydrofluorocarbon (HFC)-based fire extinguishers and lithium-ion battery fires can result in high levels of HF exposure. In such cases, the rapid production of HF from HFC-based fire extinguisher decomposition can exceed the limits set by OSHA by a significant amount [240]. In 2021, researchers developed a robust and intelligent fabric surface that can visually sense HF in a scalable manner. The study introduced a smart cotton fabric that was functionalized with maleimide-based copolymers. The detection of HF was achieved through the chromatic capability of the acid/base switch. The visual color change occurred within tens of seconds and was induced by protonation and deprotonation of the maleimide in the copolymers due to acid/base reactions. This study’s significance lies in the prepared surfaces’ superhydrophobic nature, excellent antifouling properties, and mechanical and chemical stability. Such a material would demonstrate exceptional durability under harsh weather conditions, expanding its range of applications. Moreover, smart fabrics hold immense potential in detecting acid/base gases. The utilization of as-fabricated materials and the simple technology strategy pave the way for cost-effective and high-capacity monitoring systems [241]. An optical sensor for HF detection based on a microstructured-core photonic crystal fiber was recently proposed. The device was optimized computationally to achieve a wavelength of 1.33 μm, a relative sensitivity for the absorption lines of CH_4_ and HF gases of 44.47% [242].

### 5.11. Volatile Organic Compounds (VOCs)

VOCs are chemicals with a relatively high vapor pressure at RT and atmospheric pressure. VOCs are a complex and difficult-to-detect group of substances. This category encompasses alcohols, aldehydes, aromatic hydrocarbons, non-methane hydrocarbons (NMHCs), oxygenated organic compounds, halogenated hydrocarbons, and organic compounds that contain sulfur and nitrogen. Most VOCs are hazardous, possess strong odors, and can harm human health and the environment. They can cause a wide range of versatile respiratory, immune, and allergic effects, among other issues, such as liver, central nervous system and kidney damage, loss of co-ordination, nausea, headaches, irritation to the throat, eyes and nose, allergic skin reaction, dyspnea, fatigue, vomiting, visual disorders, memory impairment, and nose bleeds. In addition, their presence can affect the air quality inside and outside buildings; their indoor concentration can be two to five times greater than outdoors, or even thousands of times more when specific activities occur indoors [243]. These compounds can originate from various sources, including natural processes and human activities. However, they are mainly generated through manufacturing processes in the energy industries, fossil fuel production and use (through their incomplete combustion or evaporation), and the use of solvents in paints, inks, and coatings, which are crucial components in the production of ash and photochemical haze [9]. Other sources include compressed aerosol products, biofuels, cooking oils, bioethanol, and incomplete biomass combustion. Hydrocarbons (CxHy) are a type of gaseous pollutant that is commonly associated with traffic pollution. It is worth noting that VOCs are prone to undergo chemical reactions that can lead to the generation of toxic gases like NO_2_, which can contribute to air pollution [244,245].

Nevertheless, they play an essential role in communicating with animals and plants. Consequently, VOC sensors are crucial in the oil industry, food and agriculture, healthcare, safety, environmental monitoring, and other fields [246]. Therefore, wearable VOC electrical, optical, and gravimetric sensors are gaining increasing scientific and technological attention. Recently, Ti_3_C_2_T_x_ nanosheets and 2D TMDCs (such as WSe_2_) were combined to create a chemiresistive sensor that selectively detects oxygen-based VOCs. The Ti_3_C_2_T_x_/WSe_2_ hybrid sensor was found to have a reduced noise level, quick response and recovery times, and high adaptability for a wide range of VOCs. In addition, the sensitivity of the hybrid sensor to ethanol was increased by over 12 times, which is a significant improvement compared to unmodified Ti_3_C_2_T_x_. Inkjet printing was used to deposit the hybrid gas-sensing membrane on a polyimide substrate with an Au-interdigitated electrode structure composed of six pairs of fingers with an active electrode area of 8 mm × 8 mm, allowing measurements to determine the concentration of ethanol gas between 1 to 40 ppm [247].

A prototype for a wearable VOC sensor was designed and demonstrated based on the synergistic effect of mechanochromic–vapochromic luminescence [244]. The synthesis pathway was developed based on crystal analysis and emissions from molecules in crystalline states. These molecules adopt a planar conformation and avoid π–π stacking through side-inserted molecules. Additionally, the mechanical structure of these crystals forces the molecules into coupled π-dimers to quench the light emission. However, upon detection of VOC, intermolecular interactions are rebuilt, and π-dimers are isolated, giving light emission an advantage. Such a molecular-level design allows for VOC detection at the ppm level.

In summary, a solid-supported sensor and wearable device were presented that leverage the principle of aggregation-induced emission (AIE) molecular systems. The AIE of choice was (E)-1-(((4-acetylphenyl)iminio)methyl) naphthalen-2-olate (AINO), which enables high emission in aggregated states, enabling the detection of VOCs in the atmosphere. The outstanding feature of this design is its easy encapsulation into clothing and gloves, as the sensor is fabricated on cellulose, making it suitable for daily use [244] (Figure 10c).
Figure 10(**a**) Printed QR code label for H_2_S sensing [230]. Reprinted from Engel, L.; Benito-Altamirano, I.; Tarantik, K.R.; Pannek, C.; Dold, M.; Prades, J.D.; Wöllenstein, J. Printed sensor labels for colorimetric detection of ammonia, formaldehyde and hydrogen sulfide from the ambient air. *Sens. Actuators B Chem.*
**2021**, *330*, 129281, Copyright 2020, with permission from Elsevier. (**b**) Processing and fabrication steps for the capacitive gas sensor [233]. Reprinted from Zhang, X.; Zhai, Z.; Wang, J.; Hao, X.; Sun, Y.; Yu, S.; Lin, X.; Qin, Y.; Li, C. Zr-MOF Combined with Nanofibers as an Efficient and Flexible Capacitive Sensor for Detecting SO_2_. *ChemNanoMat*
**2021**, *7*, 1117–1124, © 2023 Wiley-VCH Gmb, with permission from John Wiley and Sons. (**c**) Preparation of AINO-impregnated cellulose sensor [244]. (CY: Cotton yarn; BSA: Bovine serum albumin; BLA: β-lactoglobulin; rGO/M_BSA_/CY: Reduced graphene oxide/Bovine serum albumin monomer/cotton yarn composite; DGY: Dopamine–graphene yarn). Reprinted from Hu, J.; Liu, Y.; Zhang, X.; Han, H.; Li, Z.; Han, T. Fabricating a mechanochromic AIE luminogen into a wearable sensor for volatile organic compound (VOC) detection. *Dye. Pigment.*
**2021**, *192*, 109393, Copyright 2021, with permission from Elsevier.
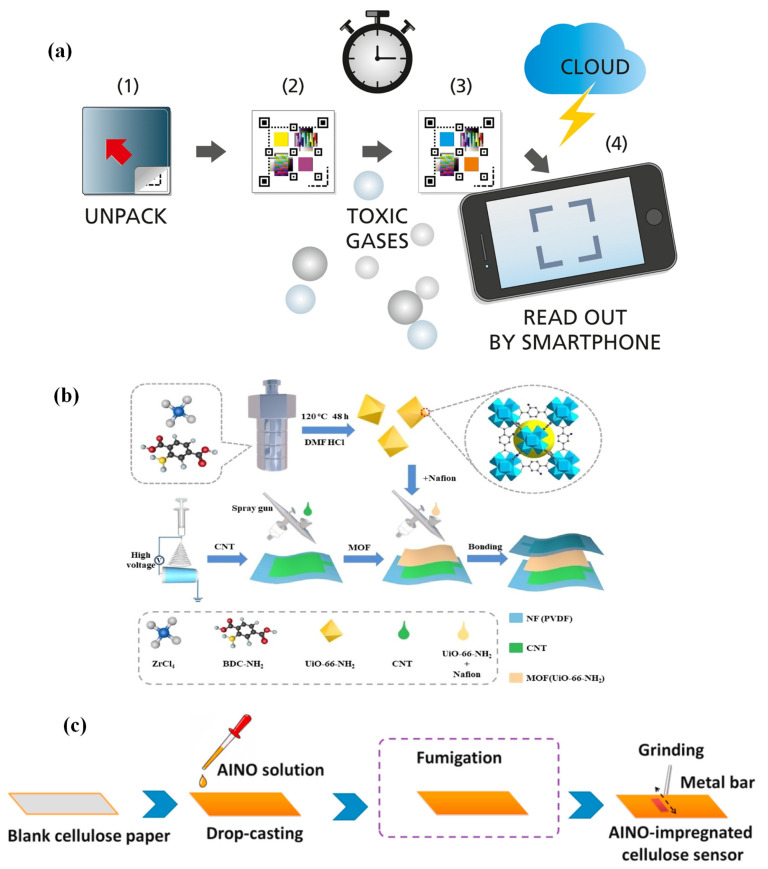



Research focusing on personalized environmental quality monitoring developed a wearable sensor to detect several factors, including VOC and NO_2_, light quality, and acoustic and thermal comfort [245]. The sensor has been integrated into a wristband, which uses Bluetooth to transmit information to a smartphone, to quickly detect, monitor, and display results to users via a mobile application. The hardware was enclosed in a 3D-printed cage-like casing. This technology holds great promise for the future, with a user-friendly interface directly targeted at end-users [245]. Phthalocyanine (Pc)-functionalized graphene chemiresistors have also been proposed. A graphene sensor array was implemented by developing sensors functionalized with different amounts of metal-free phthalocyanine and copper phthalocyanine (CuPc) [248]. The characteristic response pattern of each device for specific analytes was different, allowing the detection and differentiation of different VOCs (acetone, ethanol, formaldehyde, and toluene). LODs in the low ppm range were achieved. In another chemiresistive example, laser-patterned porous carbon/ZnO nanostructure composite inks were created on flexible PET substrates to detect acetone and toluene [249]. Laser carbonization allowed the in-situ formation of ZnO nanoparticles from the Zn(NO_3_)_2_ contained in ink also composed from a citric acid/urea-based carbon network. Optimizing the initial Zn(NO_3_)_2_ concentration allowed for porosity and polarity tuning. At a concentration of 5 wt% Zn, the response toward acetone was increased (−21.5% response at 2.5% acetone), which is attributed to a significantly higher porosity. When using pre-synthesized ZnO nanorods as additives, a relatively high selectivity toward toluene (−0.8% response at 0.4% toluene) was accomplished.

A summary of what has been discussed thus far on different gaseous pollutants detected by nano-based wearable gas sensors is provided in Table 1.

## 6. Challenges in Optimization of Sensor Performance

### 6.1. Circuit Integration and Miniaturization

A critical part of a sensing system is the electronic instrumentation used to interrogate and control a sensor. Dedicated circuitry also has the capability of addressing shortcomings that sensors may have. For example, they can provide averaging, filtering, and computational functions to improve sensor signal quality and signal-to-noise-ratio (SNR), remove offsets, provide stable biasing conditions, or extract the necessary information. Consequently, electronic instrumentation can enhance and optimize sensor performance. Using generic off-the-shelf commercially available electronic components, while convenient and relatively easy with fast production and prototyping cycles, will lead to sizeable overall device size and power consumption (discussed in more detail below). The reason is that these are generic chips designed to address many applications and are, thus, not optimized for a particular application in mind. At the same time, microchips containing all necessary functions will typically be unavailable. In addition, within academic research, the novelty and innovation in developing electronic systems using off-the-shelf commercial components are very low. To enhance overall system integration, provide additional functionalities, minimize power consumption, and potentially improve speed and bandwidth, the development of custom application-specific integrated circuits (ASICs) using advanced deep sub-micron CMOS technologies is required. Doing so also allows the realization of complete systems-on-chip (SoC) implementations. Mixed signal circuit design is, thus, essential, while the implementation of most processing and computation tasks, where possible, should be performed in the analogue domain. The solid-state circuit system communities have played a pivotal role in these developments. It is clear from the above discussions that ASIC development is the only way forward, especially for wearable applications, where power consumption and system size are crucial. Packaging and bonding technologies are essential for integrating ASICs into flexible, stretchable wearable devices. ASICs are relatively thick and rigid, based on traditional silicon technologies. Thinning the silicon chips to make them somewhat flexible and easier to integrate with thin-film flexible and stretchable devices is, thus, attracting interest in creating a hybrid technology [250,251,252]. Greater miniaturization and integration can be achieved if the sensors and electronics are on the same substrate and exploit the ASIC manufacturing and design processes to realize sensors. This may solve many issues related to the packaging and bonding of the ASIC to the sensing device, as they are now the same device [150,154].

In the coming years, edge computing and on-node sensing information processing will play a critical role in wearable sensor nodes, networks, and the IoT. Neuromorphic and in-memory computing (memristive, spintronic, and phase-change) technologies, especially where analogue processing dominates, are expected to lead these developments, allowing for ultra-low power consumption and high integration and miniaturization [253,254,255]. These developments will be helpful for wearable sensors and autonomous robotic systems that are sensor-enabled. Statistical processing and machine-learning approaches are already being implemented in sensing applications [256], and their implementation on silicon is the next step toward miniaturized intelligent wearable sensing nodes. Complex algorithms running on relatively large and power-hungry devices are expected to be realized in much smaller, more energy-efficient devices. Such approaches will further enhance decision support and readout integrity by applying advanced computation methods on the sensing node and in real time to analyze noisy sensor signals and perform data fusion (e.g., Kalman filters) [257].

With miniaturization and integration, chemiresistive micro-hotplate devices have attracted significant interest. Miniaturization of the sensing element and hotplate reduces power consumption (from 100 s of mW to tens of mWs). Implementing these on CMOS allows the co-integration of the sensor with the required analogue and digital electronics using established technologies and the microelectronics industry’s related economies for scale and scope [258]. This high miniaturization and co-integration also allow large arrays of miniaturized sensors on the same substrate. Challenges are related to the post-processing of the CMOS die to enable the realization of the hotplates (e.g., front etching techniques to achieve a suspended membrane or back etching for a closed membrane) and also the deposition of the gas-sensitive material on the CMOS die, as these are typically CMOS-process-incompatible materials. Thus, care must be taken during the post-processing steps to ensure that the sensor readout and control electronics are not damaged [259].

### 6.2. Real-Time Sensing

One of the main challenges of sensors is conducting real-time measurements. For real-time monitoring, many strategies have been created and introduced. Meanwhile, due to their growing importance in the industry 4.0 revolution, wireless sensor networks (WSNs) and IoT-based systems are the most prominent technologies created for indoor air quality monitoring (IAQ) [260]. Machine-learning solutions can empower real-time IAQ monitoring. This is further enabled via their low-power and computationally efficient implementation into new-generation microprocessors for edge computing through the rise of mobile devices and platforms, the IoT and big data analytics, and cloud computing, where computation can occur remotely. Air quality may now be readily monitored and regulated in real time due to the emergence of IoT-based handheld IAQ-monitoring devices. Real-time sensing is also crucial for firefighters and factory workers involved in industrial safety processes. A smart fire detector, for example, should respond before smoke appears and warn whether chemicals are about to burn. Real-time smart gas monitoring meets the difficulty of computational time and data transfer, necessitating the collecting of sufficient gas data from each sensing location and fast processing of gas data to interpret the results. Real-time sensing requires the deployment of many gas sensors, as well as reliable processing resources, to train a model [261]. Numerous IoT-based IAQ-monitoring systems and devices involving open-source data-processing and communication technologies have been commercialized. The sensor’s detecting qualities, such as sensitivity, selectivity, and reaction time, may be improved by adjusting and altering the charge transfer between the analyte and sensing material. However, the condition that IAQ sensors identify contaminants at sufficiently low concentration levels is more difficult to meet [260].

A 2D metal oxide nanosheet (NS)-based humidity sensor with a wearable wristband-type sensing module has been proposed as a practical demonstration of real-time fast breath humidity monitoring during normal breathing. It is necessary to collect real-time breath humidity data to calibrate gas sensors against interference signals in breath components and to reliably monitor a patient’s physiological information for non-invasive and point-of-care diagnostics applications. Metallic ruthenium (Ru) oxide NSs and semiconductor manganese (Mn) oxide NSs were prepared separately on a flexible substrate patterned with interdigitated sensing electrodes to investigate the different humidity-sensing behaviors. Resistance changes were measured after the NSs were exposed to various humidity levels [262]. Real-time rapid breath humidity monitoring during normal breathing has also been demonstrated using 2D metal oxide NSs. For this, a 2D NS-based humidity sensor in conjunction with a stretchable detecting module was worn on the wrist. The wearable sensor module wirelessly transmits sensor data to a smartphone. Breath enters the Ru oxide NS sensor, and real-time resistance changes are transmitted to a smartphone for viewing. The results demonstrated that the oxide NSs might be used directly to detect humidity in real-life situations since Mn and Ru oxide NSs had recovery durations of only 24 and 8 s, respectively [262].

Wearable technology demand is increasing exponentially, necessitating the development of enhanced smart wearable devices with low power consumption, real-time readout, biocompatibility, and self-healing capabilities. Even though there has been significant progress in developing wearable environmental sensors, several obstacles and challenges are yet to be overcome. In terms of materials, a better understanding of the characteristics of nanoparticles is required to achieve biocompatibility and self-healing capabilities, as well as high performance and low operating costs. Furthermore, biosynthesis-based green techniques can be exploited for the on-site cleanup of nanomaterials from environmental pollutants [263].

### 6.3. Repeatability and Reusability

Sensor metrics related to repeatability and reusability are often ignored, as there is a focus on describing a novel methodology, transduction mechanism, and approach. Nevertheless, a novel technology that can have an actual impact needs to address issues with sensing repeatability and sensor reusability, among other metrics. These are often factors limiting the commercial success of a proposed sensing approach. Other important parameters are the manufacturability and large-scale bulk fabrication of a sensor (thus, repeatability in the sensor manufacturing process, i.e., throughput), fabrication, operational costs, and maintenance. These are often considered once the technology has matured beyond a validated prototype and at high technology readiness levels (TRLs), but should form an essential part of the process from conception in order to guarantee favourable results. Repeatability is related to concepts discussed in the previous sections, such as hysteresis and drift. If the sensor response drifts continuously over time, the repeatability of the measurement is affected. When cycling through the range of values of a target analyte, if the sensor does not always lead to the same values, again, repeatability is not guaranteed. These are crucial in real-life scenarios, where a mixture of gases will be present, and the various concentrations will be unknown and varying. Sensing materials critical for transducing a physical event into an electrical signal must be reusable and not saturate within the range of interest. The repeatability and reusability of micro-hotplate-based gas sensors are also related to the ability of the hotplate and sensing layer to sustain the required elevated temperatures over time, especially over multiple operating cycles. As discussed in other sections, thermal stress can be translated into mechanical stress, forming cracks and pinholes. At the same time, the elevated temperatures can also cause microwire thermal elements to fuse and delaminate [264,265].

### 6.4. Power Consumption

Power consumption must be considered when designing portable electronic devices (point-of-need, point-of-care, etc.), especially wearables. Wearable systems are expected to be as unintrusive and hassle-free as possible to facilitate end users’ wide acceptance rate and seamlessly integrate into their daily routines. Power consumption will dictate battery size, which needs to be kept small to enhance wearability, portability, and overall system integration. At the same time, it also dictates the frequency of battery recharging and charging times. To a large extent, power consumption is dictated by the implemented electronic instrumentation and whether wireless data transmission is required, as well as the frequency of sensor readout and data transmission (continuous vs. periodic). ASIC implementations with deep submicron technology nodes must be used to push the power supply rails to a minimum (below 1 V) while exploiting weak inversion operation where possible in the analogue CMOS front-end electronics and analogue signal processing and computation. At the same time, the sensing modality can also dictate power consumption. For example, optical sensing methods are generally more power-consuming than purely electrical ones. Many sensing approaches use direct current (DC) electrical measurements; thus, bandwidth is not an issue, and power consumption can be kept low. In contrast, many gas sensors use micro-hotplates exploiting Joule heating, which can be energy-consuming [250]. Such metal-oxide-based sensors rely on chemical reactions requiring high temperatures to improve their efficiency, leading to increased power consumption in the range of 100 s of mW to 1 W. Thermal isolation using micromachining techniques can reduce their power consumption by a few mW [250]. Consequently, a single gas sensor can consume tens or hundreds of mW. This will limit the number of sensing elements and targets possible for many applications, necessitating large batteries and/or leading to a short battery life, which is unsuitable for wearable multiparametric applications. Pulsed temperature operation (PTO) is a mode of operation commonly used with such gas sensors to extend the system’s battery life and reduce power consumption. Essentially, the device is turned on and off (hence, the term ‘pulsed’), and the ratio of the on to the total cycle duration is inversely proportional to the power saved. PTO duty cycles above 20% are typically used. Unfortunately, PTO is associated with reduced sensor performance, e.g., slower response time with long transient periods before the sensor response stabilizes. Researchers have studied and exploited various characteristics in the sensor transient response to extract useful information, such as the minimal, maximal, and steady-state device resistances [250]. Recently, a μW-range PTO scheme was proposed [250]. This low-power approach was evaluated by detecting ethylene, acetaldehyde, and ammonia gases using three Au-doped SnO_2_ and one SnO_2_ device, each with a separate 80 μm diameter micro-hotplate. A 1.61 V supply to the heaters induced 400 °C temperature, allowing a 1.5 ms thermal time constant and a 14.5 mW power consumption for each element. The sensors were integrated into a PCB with off-the-shelf components, radio-frequency identification (RFID) capabilities, and an on-PCB inductor. Under these conditions, a 25 mAh battery can supply the system for only 40 min or 2.7 h if only one device is used. Low-power operation modes degrade sensor sensitivity. The temperature can be controlled using pulse-width modulation (PWM) in the excitation signal at 30 kHz instead of DC, with 1.8 V pulses and duty cycles between 10% to 0.05%. This depends on several processes on the device, such as ensuring that the device has reached a steady-state temperature (which is possible with the modes explored, 5 ms required vs. 32.5 ms used). Nevertheless, the ionic species on the sensor surface involved in the chemical reaction with the target analyte require some time to reach a new equilibrium and take place [250]. These slow processes are not adequately understood and are challenging to establish theoretically. It has been found that they cannot reach the stationary state with very short temperature pulses, limiting sensitivity, which experimentally changes logarithmically with the power applied to the sensor heater. Nevertheless, only a moderate sensitivity decline was found within the proposed power-saving modes. Even so, the sensitivity is in the same order of magnitude with continuous temperature stimulation operation. Mixed conclusions were drawn for the LOD, as there seemed to be a dependency on the type of target volatile, and, thus, it needs to be explored case-by-case. The NH_3_ LOD remained constant; for acetaldehyde, it dropped five times, while for ethylene, it increased. The most significant degradation is related to the sensors’ response time, which can be several hours (between 10–20 h for aggressive power consumption reduction towards less than 200 μW) compared to 10 min. This can be detrimental in wearable applications.

It is clear from the above that much effort has been focused on maintaining the operating temperature high to ensure selectivity, sensitivity, and LOD are high. Micro-hotplate technology enabled by microfabrication techniques is one approach to maintaining high-temperature operation and reducing overall power consumption. This reduces the power consumption from 100 s of mWs to 10 s of mWs [266].

Modification of the overall geometry of the heating element (shape, track width, thickness, and length), as well as the material, and reducing the device’s thermal conductivity will affect the overall power consumption of the element, as these are based on Joule heating, which is proportional to the electrical power and thus the resistance of the device and the voltage applied to the device or the current flowing through the device. However, increasing the electrical resistance of the heating element to reduce the power consumption must be carried out with caution to ensure that the component (most often a wire) can sustain the current injected and the temperature induced, and that it does not fail. Using cavities and insulating material optimization (e.g., Si_3_N_4_ or SiO_2_) can also aid in reducing power consumption [264]. One approach uses suspended structures or examples achieved through micro-electro-mechanical systems (MEMS) processes. It has been found that reducing the number of bridges to the suspended structure and their width also reduces power consumption. Nevertheless, others have focused on reducing power consumption by operating sensors at RT, which hampers gas sensors’ sensitivity, selectivity, LOD, and response time. Thus, efforts have been made to enhance these using, for example, additional membranes, hybrid materials, and noble-metal functionalization [264,265].

Another approach is to exploit energy-harvesting approaches. For example, piezoelectric and triboelectric nanogenerators have been demonstrated that allow RT operation [267]. ZnO can generate a piezoelectric output acting as the power source and the sensing signal. Incorporating metals such as Cu, Pd, Au, and Pt into ZnO enhances catalytic reactions and sensing performance. ZnO-oxide-based heterostructures, such as ZnO_2_/SnO_2_, ZnO/In_2_O_3_, and ZnO/NiO, enhance sensitivity, leading to a stronger output signal [267]. Gases adsorbed on the surface of triboelectric nanogenerators affect the charging effectiveness of these devices, which can be exploited as a sensing mechanism [267]. Self-heating devices are a promising class of devices. MOS nanowire (NW) materials are typically preferred for such devices. NWs act as the pathways for electron transfer, and their length relates to the temperature generated. The diameter of the nanowires is also associated with the latter [267]. Essentially, increasing the resistance of the nanowire will push the power consumption down. This approach leads to μWs of power consumption due to the limited thermal capacitance of the NWs [267]. SnO_2_ NWs with diameters in the 95 nm region that can be grown with a vapor-liquid-solid (VLS) method are one example that has demonstrated a power consumption of 25 nW and the response of 2.1–2.5 ppm for NO_2_ [267]. CVD has also been used to create 80 nm NWs of SnO_2_ for C_2_H_5_OH sensing [267]. Decorating such NWs with Ag nanoparticles has allowed for the low temperature and, thus, low power detection (<20 mW) of H_2_S with a response of up to 21.2 ppm at 2 mW and a response/recovery time of 18/980 s [267]. ZnO decorated with Pt or Pd with voltages 1–20 V applied to them is also a promising approach for the RT detection of toluene and benzene [267]. At 5 V, the power consumption was only 208 μW and 139 μW, respectively. The power consumption in these sensors depends on the electrode sensing area and the design of head-to-head electrodes [267]. Au/SnO_2_-ZnO core-shell NW are the most promising materials for low-power self-heating gas sensors. Nevertheless, NW-based devices require complex multistep fabrication processes with low yield and are, thus, unsuitable for large-scale bulk fabrication [37,155,160,265,267,268].

### 6.5. Gas-Sensitive Material Deposition

The various gas-selective membranes can be deposited on the sensing surfaces using a plethora of manufacturing approaches. This subject was recently reviewed in detail [265], and, here, we will thus only discuss this in summary. Drop casting, followed by baking, is probably the most straightforward method for realising thick films with hard-to-define thickness. Liquid phase deposition (LPD) is a low-temperature growth method that leads to dense films characterized by a slow response. An effective technique is atomic layer deposition (ALD), which allows ultra-thin films with precise thickness and composition even at low temperatures. ALD is, however, a slow and high-cost specialized process. Nevertheless, LPD and ALD are suitable for SnO_2_ deposition, a standard gas-sensing material in chemiresistors [265].

CVD is another common technique to obtain gas-sensing thin films, such as NiO and graphene, with favourable crystallinities and uniformities suitable for large-area deposition; however, impurities can be quickly introduced in the process. Sputtering is an established method for thin-film deposition ideal for the mass production of high-quality and highly repeatable sensing films, e.g., SnO_2_, WO_3_, ZnO, CuO, and MoS_2_. It requires a high vacuum, and the adhesion of the deposited films is increased. In situ growth of gas-sensing nanomaterials is another low-cost approach suitable for mass production. This is often used with ZnO-based materials to create nanowires and nano-branched geometries. The adhesion of the generated films is good, but the steps required to prepare the solutions used in the process can be high [265]. Self-assembly of sensing membranes directly on electrodes and substrates is a powerful approach. It is based on electrostatic interactions to realise highly ordered thin films with a fast response time. An advantage of in situ and self-assembly methods is that they do not require expensive equipment and are relatively cheap processes suitable for mass-fabricating porous gas-sensing membranes. Another simple method is DEP, a low-cost, simple RT method with good repeatability [265]. Defining the frequency and amplitude of the excitation signal can be a challenge. It allows the arrangement of the gas-sensitive material along the direction of the electric field gradient on the surface of, for example, a hotplate. This approach has been used to deposit ZnO and InO nanowires from dispersions. Following DEP, an annealing step is typically used at elevated temperatures (e.g., 400 °C) [265].

Spray pyrolysis is another simple method belonging to the additive manufacturing family of techniques for the realization of thick, porous films by spraying inks over heated substrates. The adhesion of the generated films using this method can be poor. Another additive approach is inkjet printing [265]. This approach has been used to print-on-demand thick films from inks of, for example, Pd-modified SnO_2_ nanofibers and SnO_2_ nanoparticles onto hotplates. Inkjet printing, however, being a direct-write technique, can be a slow process unsuitable for large-scale fabrication; depending on the resolution, accuracy, and inkjet-printing method used, it can be both a low-cost and high-cost technique. Stencil printing and its related method, screen printing, are simple, low-cost techniques for printing both electrodes (e.g., interdigitated) and gas-sensing membranes [265]. These techniques require relatively thick pasty inks to be used, especially if small features with excellent resolution are needed. Screen printing has the advantage of being able to print closed geometries, e.g., circles, that stencils cannot print. Both techniques are suitable for the large-scale bulk fabrication of devices. Spray coating can be combined with stencil printing to create sensing films. The thickness and uniformity of thick films deposited with additive methods can be hard to control and define [265].

### 6.6. Selectivity and Sensitivity

Selectivity is an essential aspect of any sensing technology. A sensor needs to provide sensitivity, or, at least, the primary sensitivity, from only one specific event, e.g., the change in the concentration of one target gas within a mixture of gases. If the sensor is sensitive to more than one analyte, then it may not be possible to distinguish what the recorded response is from and to reach realistic conclusions. However, it may be that different analytes lead to different responses in the measured output signal, such as different ranges or different transient behaviors that can lead to safe conclusions. Nevertheless, this is not ideal, as interpreting the measured data can be challenging and may lead to ambiguities. The ratio of sensor signal response the target gas reaches to that achieved with other interfering gases is often used to define selectivity [269]. The above can be described as intrinsic sensor selectivity.

Principal component analysis, machine learning, and other statistical approaches can be exploited to address the above issues and enhance measurement interpretation [270]. Such data-processing approaches can be categorized as analysis selectivity enhancement approaches, that, when possible, can enhance the selectivity of the sensor with low intrinsic selectivity. However, their capability to operate in small portable devices and at the component level, e.g., in a low-power microcontroller or a custom integrated circuit, with ultra-low power consumption for on-node data processing, is an area of significant interest and current development. It is expected to influence the pervasiveness of these sensors in wearable applications. Furthermore, using gas sensor arrays and analyzing sensing patterns among many such elements employing different temperatures or different sensing materials and membranes and employing pattern recognition approaches to the recorded data is another approach to enhancing sensing accuracy [269,270]. In any case, enhancing selectivity/specificity enhances the confidence in the conclusions drawn when interpreting sensor data and reduces false positives/negatives.

The methods exploited to enhance the selectivity of gas sensors can be roughly divided into two categories: (i) physical and (ii) chemical modifications [269]. The former is related to changing the device’s physical properties, such as surface structure and morphology (e.g., roughness, porosity, and geometry), or using membranes that essentially act as filters. The latter involves using materials that recognize a target molecule through chemical processes chosen based on the target molecule’s chemical properties, required reactions, or adsorption abilities [269]. Chemical modifications lead to excellent selectivity, and several approaches involving both families of modification approaches can be exploited [269]. Chemical modification methods can be categorized themselves, as well as into strategies based on catalyst decoration (e.g., using oxidation, dissociation, and adsorption catalysts), composite formation (e.g., using conductive, imprinted, or non-conjugated polymers), or surface functionalization (e.g., using active chemical, organometallic compounds, or self-assembled monolayers, SAMs) [269]. For example, oxidation catalysts can promote the oxidation of a target gas, increasing the reaction rate between surface-adsorbed oxygen and the target (e.g., Cr-doped NiO for toluene and o-Xylene, and ZnO nanofibers for H_2_) [269].

On the other hand, dissociation catalysts indirectly increase the surface reaction rate. For example, Ag-loaded metal oxides, such as SnO_2_ and TiO_2_, lead to high ethanol selectivity. The increased alkalinity leads to ethanol decomposition into acetaldehyde that, due to higher polarity and reactivity, leads to enhanced sensing performance. Another example is noble metals, such as Pt or Pd, for catalyzing hydrogenation reactions. The preferential adsorption affinity of a target to the catalyst surface augments the target-sensing reaction. Examples include Ag-ZnO-rGO for acetylene and Cu_2_O-loaded graphene for H_2_S. Disadvantages of catalysts include catalyst poisoning, susceptibility to humidity, and the requirement for high temperatures, which can be addressed by using, for example, UV irradiation instead of heat, using hydrophobic materials, and employing protective layers. Nanocomposites of PANI with MOSs, graphene, or CNTs are typical for detecting NH_3_, while porous SnO_2_ with poly-3-hexythiopene (P3HT) is selective to NO_2_ [269].

Polymers can swell in the presence of specific gases. This can be exploited, for example, both for resistive and capacitive sensing. One example of the family of nano-conjugated polymers is polyetherimide (PEI) composites with carbon black (CB) that have been modified to provide increased nitrogen to oxygenated sites and have been found helpful for detecting medium-length aldehydes. Adsorption of aldehydes to the nitrogenated sites leads to the swelling of the polymer [269]. Molecularly imprinted polymers (MIPs) are materials with template-induced cavities that allow target recognition by the shape and size of the molecule, in addition to functional groups and adsorption affinity. Binding the target gas to the template leads to the expansion of the polymer and, thus, a considerable change in conductivity. Chemical modification methods aim to increase the chemical affinity between the sensor surface and the target gas by exploiting the chemical properties of the gas. One example of active chemicals is rGO, a p-type semiconductor. rGO modified with sulfanilic acid is further p-doped as the added acid is electron-withdrawing, and NO_2_ enhances this, leading to enhanced conductivity [269]. SWCNTs non-covalently modified with pyrene/cobalt phthalocyanine also attained selective detection of H_2_S [271]. Another example of NO_2_ detection is the functionalization of WO_3_ with (3-aminopropyl) triethoxysilane (APTES), while APTES-modified silicon nanowires have demonstrated specificity to trinitrotoluene (TNT) [269]. Biomimetic SAMs use oligopeptides, mimicking particular human olfactory receptors and are sensitive to heat and irradiation [269].

For example, RT operation in resistive metal-oxide sensors can be challenging as their selectivity is severely limited in low temperatures. Significant efforts have been made towards achieving this goal, as RT operation would significantly reduce their power consumption, primarily in the 100 s of mW range [270]. Some approaches have, thus, been examined to enhance RT selectivity, some of these mentioned earlier. These include using permeable membranes, combining n- and p-type metal oxides to form heterostructures, nanocomposites, or core-shell structures, doping semiconducting oxides with metal ions, or functionalizing surfaces with various polymers or metal oxides, and combinations of all the above [270]. Physical modification of membranes is the critical element in this direction, particularly zeolite-based materials and 2D nanomaterials, such as graphene and GO. Porous graphene with angstrom-sized pores demonstrates selective molecular sieving and properties superior to polymer and silica membranes. GO sheets stack non-uniformly, allowing permeability to small gases without requiring the formation of pores. Based on the above, ZnO metal oxide nanowires with < 50 nm nanoporous GO membranes have been proposed to reduce the permeability of specific gases. This allowed highly selective ultra-low power (200 nW) H_2_ gas sensing. It is evident, thus, that there is an interdependency between selectivity and power consumption [270].

### 6.7. Stability and Response/Recovery Time

Sensor response stability (drift), response and recovery time, and hysteresis are essential metrics defining a sensor’s operation and capability to be used in specific applications. The response time of a sensor needs to be shorter than the transient response of the analyte being measured. The sensor signal should stabilize and minimally drift within this timeframe to allow monitoring of this transience. If the measurement changes before the sensor response stabilize to take a measurement, the acquired data will not be valid. In addition, the hysteresis needs to be smaller than the required LODs to avoid measurement ambiguities and false negatives/positives regarding thresholds. Specific sensors, such as electrochemical sensors, often need conditioning in specific gaseous or liquid environments before use. Other sensors based on the change of material properties, such as analyte-responsive polymer/hydrogels that allow sensing through changes of shape (volume), will typically require longer times to respond, stabilize, and recover. Resistive-typed gas sensors employing micro-hotplates must establish a stable operational temperature. This can be in the range of ms [250].

In addition, the chemical reactions on the surface of the material being used as the sensing surface are related to their time constants and kinetics that need to stabilize, which are slow processes. These are not adequately understood and are typically examined and established experimentally [250]. The use of methods for reducing power consumption, such as PTO in micro-hotplate resistive sensors, can significantly negatively influence the response time of these sensors from a few minutes to tens of hours [250]. Stability is also related to the actual endurance and lifetime of the device. For example, excessive heating over long periods will lead to mechanical deformation of a micro-hotplate due to the induced thermal stress, and the heating element can also fuse [264]. This mechanical deformation can also cause the devices to crack, thus damaging the sensing materials and causing sensor failure [264]. Geometrical optimization, e.g., through finite element simulations, can reduce these mechanical stresses and thus enhance the endurance and lifetime of these devices [264]. The choice of materials used can have a meaningful impact on the lifetime and stability of the device. For example, ceramics have a smaller coefficient of thermal expansion than silicon; thus, the ratio of the coefficient of thermal expansion of a Pt heating wire to that of a ceramic will be smaller than with silicon as the substrate [264,265]. Consequently, thermal stress is smaller. Adhesion of Pt to ceramics can be good, but care must be taken to ensure delamination and pinholes/porosity are not induced at elevated temperatures. Using intermediate adhesion-promoting layers such as Ti addresses these [264,265]. Using suspended structures realised using cavities while allowing lower power consumption leads to structures with reduced stability over time compared to standard closed-film devices.

## 7. Conclusions and Outlook

This review presented an update to almost all aspects of available smart wearable (nano)sensing devices for the environmental monitoring of gaseous pollutants, with a focus on the challenges in the optimization of sensor performance which is, currently, a considerable gap in the literature. Different sensing modalities and approaches, as well as materials and manufacturing methods, were thoroughly reviewed, also discussing many recent examples from the published academic literature for different gases of interest. As there is an urgent demand for fabricating reliable sensing platforms to detect and quantify these pollutants accurately, devised multidisciplinary approaches should be proposed. On the other hand, these approaches should constantly implicate the concerns and opinions of analytical chemists, biochemists, ecological and environmental researchers, toxicologists, risk management specialists, and governmental authorities in order to be upgraded to an industrial level. Distinctive features of various novel nanomaterials, especially their ability to accommodate necessary functionalization, have made them promising candidates for using sensors for gaseous pollutants. Wearable gas sensors will progress technically over the following years due to their great potential toward real-time multiparametric non-intrusive smart wearable sensing. This is further supported by the numerous applications where such sensors are valuable, that range from body monitoring for work safety to theranostics, and the added benefit they can have, which should support user acceptability and widespread deployment.

Moreover, to turn the challenge of commercialization of these advanced sensors into an opportunity, a thorough marketing endeavor is required so that people trust these high-tech devices enough to use them in daily life activities. To achieve this, the sensing device should be simple and comfortable enough to be directly managed and controlled by the user and to seamlessly be integrated into daily routines without requiring significantly more effort by the users. Moreover, technology developers, employers, and industry stakeholders must agree to compromise on the additional expenses of providing these platforms to users.

From the perspective of electronic engineering and material science, there are still unsolved problems in fabricating a conformable skin-friendly sensing device, as excellent biocompatibility, elasticity, and stretchability should be added to the substrates to be used on human skin. However, some flexible substrates lack the required surface energy to deal with surface adhesion issues. For instance, chemical functionalization and oxygen plasma treatment can be used to improve the adhesion of the substrate to other components of stretchable sensors. Important challenges are also related to the power consumption of such sensors, which becomes a critical and limiting parameter for wearable applications. Selectivity, sensitivity, and multi-parametric sensing are crucial aspects for a robust gas sensor that can be successfully commercially launched, aspects that also become challenging to achieve in a wearable format. Sensor lifetime, robustness, and contamination issues are also challenges crucial for wearable sensors, as well as motion artifacts that could arise from sensitivity to mechanical perturbations. Since, in some cases, used sensors should be disposable, sensor fabrication costs should be reduced as much as possible, while biodegradability and ecological impact should also be considered. Co-integration in a miniaturized and wearable (flexible or stretchable) format of all aspects required for a wearable implementation, including battery or energy scavenging, sensor, and front-end (analog) and back-end (digital), as well as radio frequency (RF, antenna, transceiver, etc.) components, is the current challenge that needs to be addressed, together with the above-mentioned aspects, for the next generation of wearable gas sensors. Such work will potentially lead towards the development of hybrid co-packed systems, and, thus, several advancements are required toward this direction and the heterogeneous integration of different devices and technologies. It might also be possible to fabricate near-zero-cost gas sensors (e.g., PB types) to detect gaseous pollutants. In terms of manufacturing technologies, for the high resolution, miniaturization, throughput, and quality of the end product, standard microfabrication techniques should be preferred. However, current advanced ion additive manufacturing approaches should not be overlooked. While often inferior in terms of manufacturing resolution, they come with certain advantages. These include the lower cost of fabrication and manufacturing instruments, and the fact that a clean room is not needed. Techniques such as inkjet, screen, stencil, and extrusion-based printing, laser engraving and carbonization, stamping and transfer printing, and their variations and their combination can be exploited to create devices, while they can also be combined, if needed, with microfabrication approaches for critical stages of the manufacturing process [272,273,274,275,276,277,278,279,280,281]. Each of these methods comes with their constraints in terms of material properties they are most suitable to be used with, manufacturing temperatures, final device dimensions, etc., so materials need to be carefully selected and optimized. As a result, manufacturing different system parts with different techniques might be required, leading to the aforementioned requirement for heterogeneous integration. In terms of materials, MOS materials, carbon nanomaterials, and catalysts play an important role in gas sensors and their future. The selection of these and their combination needs careful thinking and needs to be tailored to the specific application. The final form factor and device flexibility and stretchability are of prime importance for wearable applications, as well as power consumption and mechanical robustness. A self-powered, low-power-consumption system should, thus, be the focus in the next generation of wearable gas sensors and TENG technologies; in addition, sweat biofuel cells are promising roots towards that direction [282,283,284]. For deployment directly worn on the body, skin adhesion should be considered. In addition, sensor lifetime and reusability, especially following long wearable use and repeated cycles of removing and reattaching on the body, will prove critical. For devices meant for clothing integration, the possibility of reversible attachment to clothes could be a solution to avoid damage during washing. Crucial aspects of gas sensor performance and wearability are also related to the operational temperature of the sensor, its response and recovery time, and its sensitivity to relative humidity. The effect of humidity on gas sensor response has been highlighted throughout the paper. One approach to evaluate and properly interpret sensor response could be the co-integration of humidity sensors [285,286]. Alternatively, device, material, and machine-learning approaches can be explored [287,288,289,290,291,292]. Room-temperature-operated MOS sensors with remarkable performance are emerging [293]. As in several technological domains, the recent explosion in the development, use, and perturbation of machine-learning approaches, such techniques can immensely impact the area of gas sensors. Several papers where machine-learning approaches are used to aid gas sensors have been recently published [148,294,295,296,297].

These small nano-based gas sensors can also impact the economy and employment. Companies can easily monitor toxic gases in their warehouses and working areas by applying a smart control system. In fact, with the help of smart wearable sensing technology, which minimizes human errors, employees can work in a safe environment where gas leaks can be successfully detected, thereby preventing catastrophic incidents in factory equipment maintenance, and ensuring the efficient storage of goods and products with no loss. Such solutions can also be utilized in modern forest management (e.g., fire prevention) in the fabrication of forester or firefighter clothes and outfits, as wildfires are the most common form of natural disasters in which large amounts of, for example, CO, CO_2_, NO, and SO_2_ are released into the atmosphere. More research is required to provide a brighter vision of the potential of this technology in accelerating the development and application of novel smart wearable sensors in the near future.

## Figures and Tables

**Figure 1 sensors-23-08648-f001:**
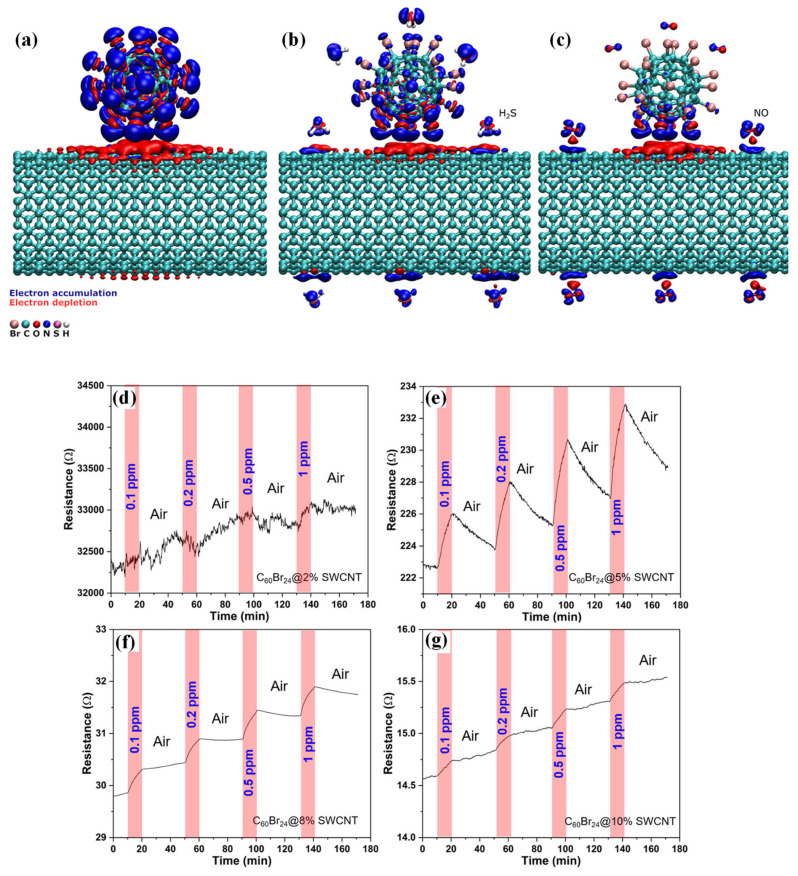
Atomic structure of the interface between C_60_Br_24_ and SWCNTs (**a**) without analytes and with several adsorbed (**b**) H_2_S and (**c**) NO molecules. Real-time response–recovery curves of sensors based on C_60_Br_24_/SWCNT composites having different SWCNT loadings: (**d**) 2, (**e**) 5, (**f**) 8, and (**g**) 10 wt%. Images reused from [34], with no changes. CC-BY 4.0 licence (https://creativecommons.org/licenses/by/4.0/, accessed on 8 October 2023).

**Figure 2 sensors-23-08648-f002:**
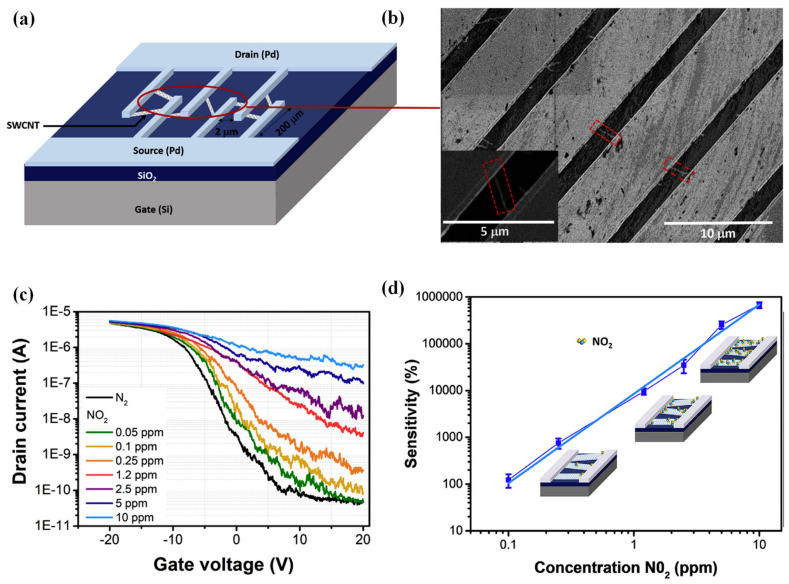
(**a**) Scheme of the device architecture, (**b**) SEM micrograph showing individual SWCNTs connecting the source and drain electrodes, (**c**) transfer characteristics under various NO_2_ concentrations on SWCNT-FET after 15 min of gas flow stabilization, (**d**) response of the SWCNT-FET gas sensor at RT as a function of NO_2_ concentration [37]. Reprinted from Sacco, L.; Forel, S.; Florea, I.; Cojocaru, C.-S. Ultra-sensitive NO_2_ gas sensors based on single-wall carbon nanotube field effect transistors: Monitoring from ppm to ppb level. *Carbon*
**2020**, *157*, 631–639, Copyright 2019, with permission from Elsevier.

**Figure 3 sensors-23-08648-f003:**
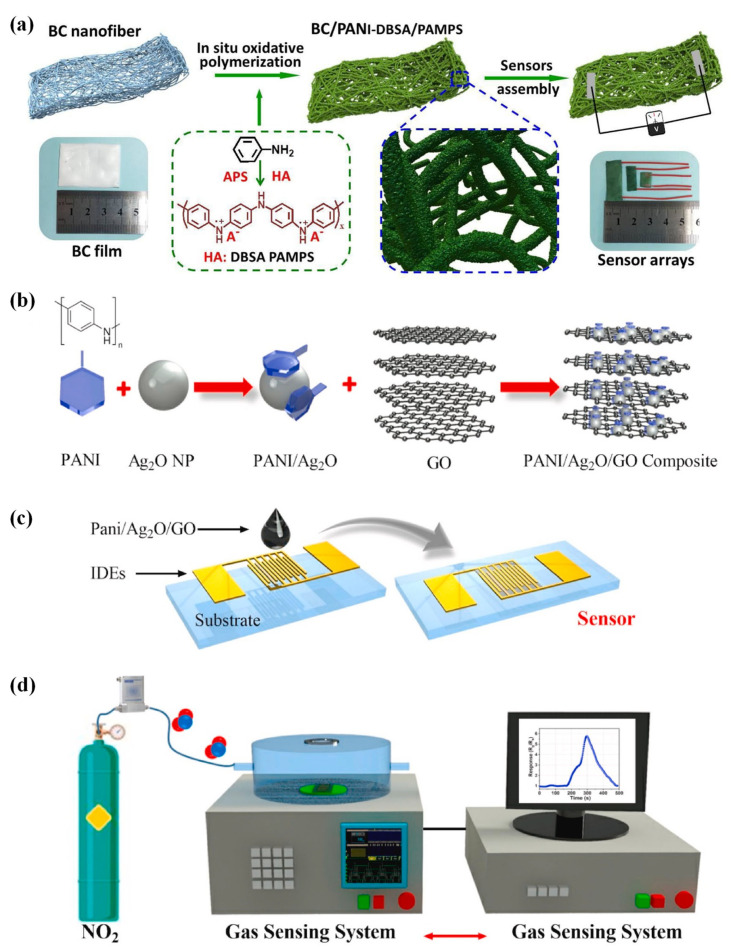
(**a**) Illustration of the manufacturing process for a DBSA and PAMPS co-doped BC/PANI (BC/PANI-DBSA/PAMPS) NH_3_ sensor is shown [52]. Reprinted from Yang, L.; Yang, L.; Wu, S.; Wei, F.; Hu, Y.; Xu, X.; Zhang, L.; Sun, D. Three-dimensional conductive organic sulfonic acid co-doped bacterial cellulose/polyaniline nanocomposite films for detection of ammonia at room temperature. *Sens. Actuators B Chem.*
**2020**, *323*, 128689, Copyright 2020, with permission from Elsevier. (**b**) Production of PANI/Ag_2_O nanoparticles and PANI-Ag_2_O-GO composite, (**c**) manufacturing of the gas sensor, and (**d**) analysis of the NO_2_-sensing performance of the produced sensor [53]. Reprinted from Umar, A.; Ibrahim, A.A.; Algadi, H.; Albargi, H.; Alsairi, M.A.; Wang, Y.; Akbar, S., Enhanced NO_2_ gas sensor device based on supramolecularly assembled polyaniline/silver oxide/graphene oxide composites. *Ceram. Int.*
**2021**, *47*, 25696–25707, Copyright 2021, with permission from Elsevier.

**Figure 4 sensors-23-08648-f004:**
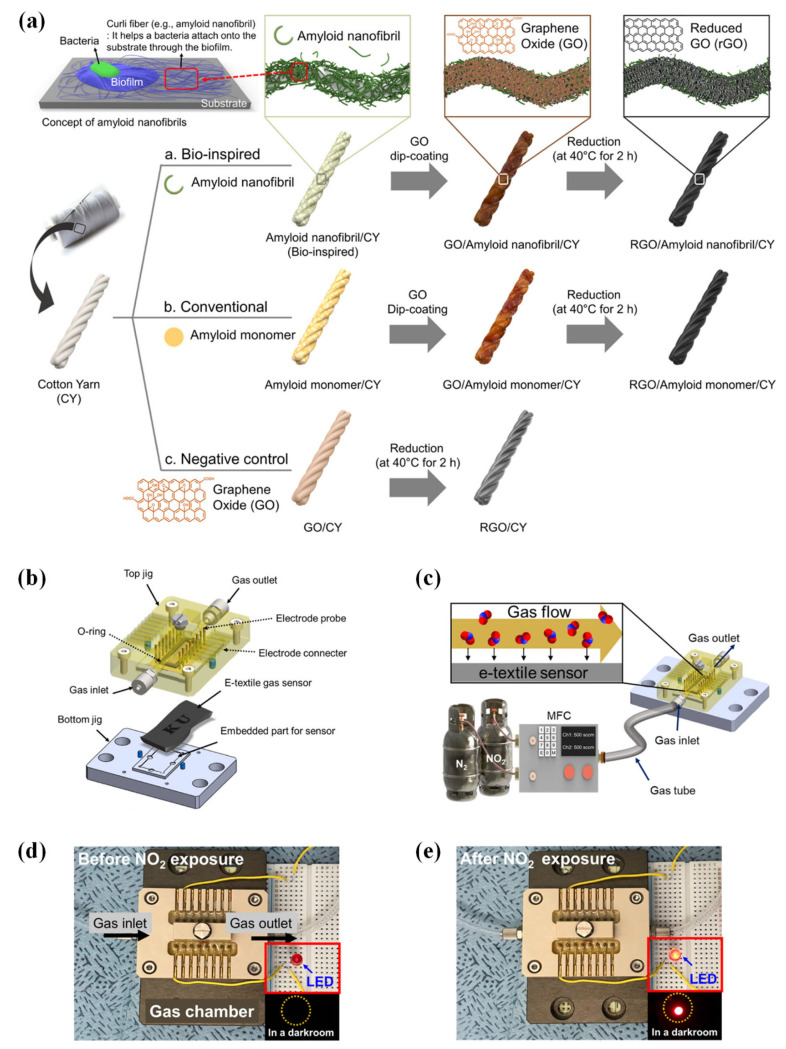
(**a**) Schematic illustration of the fabrication process of (a) RGO/amyloid nanofibril/CY, (b) RGO/amyloid monomer/CY, and (c) RGO/CY. LED-based e-textile gas sensor using the RGO/FBLG/CY (tu = 30 min) for practical application. (**b**) Schematic of the assembly and operation of the gas chamber. The e-textile gas sensor was embedded in the gas chamber before NO_2_ exposure. (**c**) Flow rate and concentration of NO_2_ were controlled through an MFC. NO_2_ was exposed to the e-textile gas sensor embedded in the gas chamber. (**d**) Red LED is not lit before NO_2_ exposure. (**e**) Red LED light is turned on after exposure to NO_2_ (10 ppm) for 4 min [103]. Reprinted (adapted) with permission from Lee, S.W.; Lee, W.; Kim, I.; Lee, D.; Park, D.; Kim, W.; Park, J.; Lee, J.H.; Lee, G.; Yoon, D.S. Bio-Inspired Electronic Textile Yarn-Based NO_2_ Sensor Using Amyloid–Graphene Composite. *ACS Sens.*
**2021**, *6*, 777–785. https://doi.org/10.1021/acssensors.0c01582. Copyright 2020 Americal Chemical Society.

**Figure 5 sensors-23-08648-f005:**
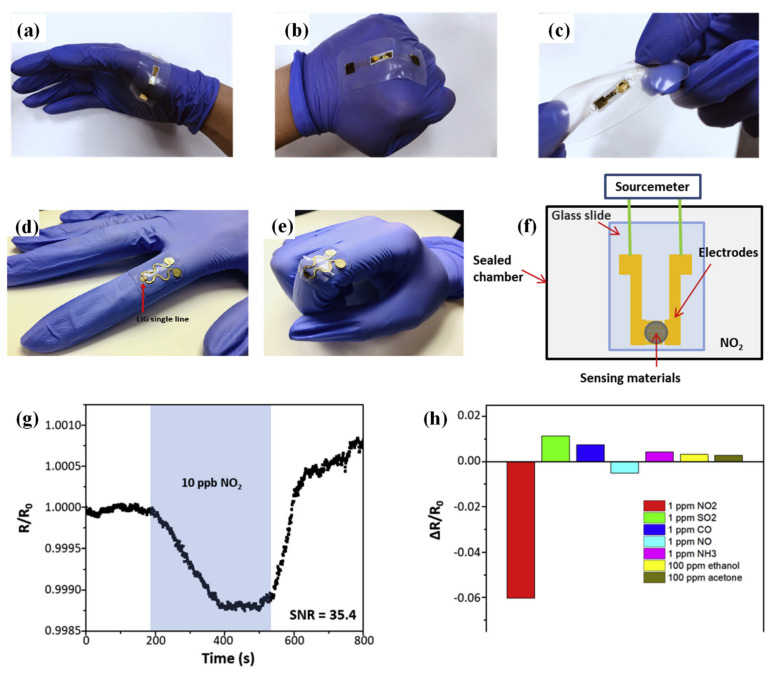
Demonstration of the stretchable gas sensor (**a**) deformed by conforming to the hand, (**b**) bending to the fist, and (**c**) twisting. Images of the stretchable wearable rGO/MoS_2_-based NO_2_ gas sensor (**d**) before and (**e**) after bending of the finger. (**f**) Schematic illustration of the testing setup for the gas sensor, including a sealed chamber, interdigitated electrode, and a source meter (SM). (**g**) The measurement of the gas sensor to NO_2_ of 10 ppb demonstrated an excellent SNR of 35.4, promising the detection of NO_2_ with ppt concentrations. (**h**) The selectivity of the stretchable MoS_2_@rGO gas sensor to NO_2_ [116]. Reprinted from Yi, N.; Cheng, Z.; Li, H.; Yang, L.; Zhu, J.; Zheng, X.; Chen, Y.; Liu, Z.; Zhu, H.; Cheng, H. Stretchable, ultrasensitive, and low-temperature NO_2_ sensors based on MoS_2_@rGO nanocomposites. *Mater. Today Phys.*
**2020**, *15*, 100265, Copyright 2020, with permission from Elsevier.

**Figure 6 sensors-23-08648-f006:**
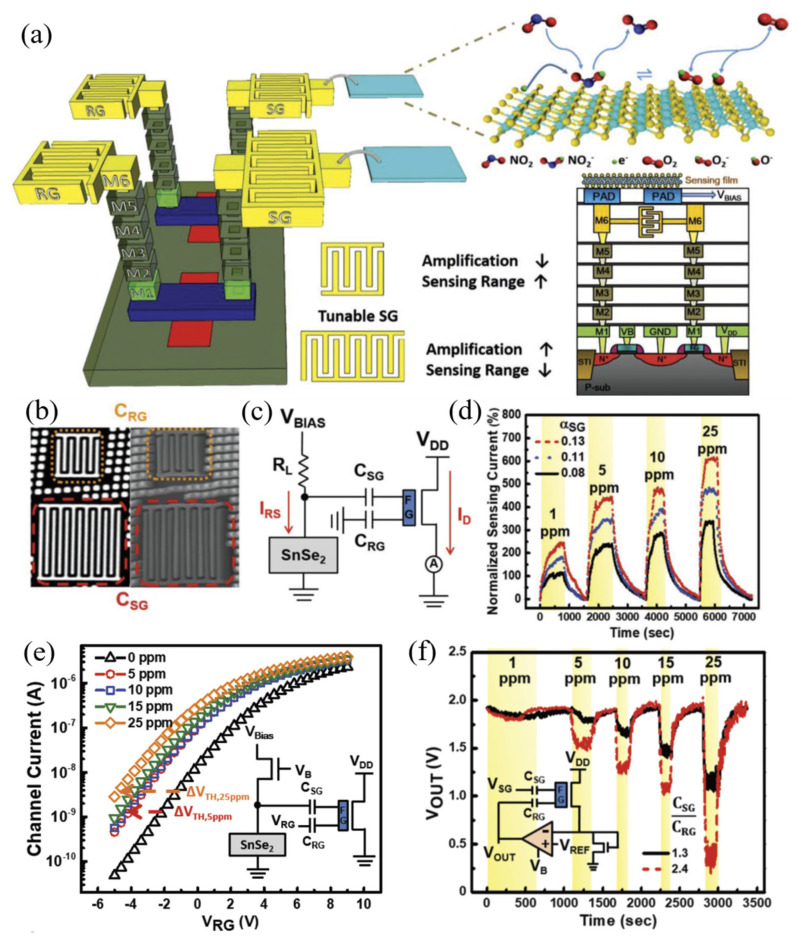
(**a**) Diagram of the proposed structure in the FGMOS sensing architecture. (**b**) The finger-type interdigitated MOM structure. (**c**) FGMOS-based (MOM-coupled NMOS) sensing circuit amplifies the resulting potential due to resistive changes in the SnSe_2_ film arising from NO_2_ exposure. (**d**) The effect of using different sensing gate coupling ratios to the measured current using the circuit of (**b**). (**e**) The change in V_TH_ shift at αSG/αRG = 3.5 when exposed to 5, 10, 15, and 25 ppm NO_2_. When the 2D SnSe_2_ layered film is exposed by NO2, it contributes a larger voltage on floating gate, leading to a smaller VTH on read gate of the floating-gate device. (**f**) The response of the self-balanced readout circuit using two different coupling ratios and different NO_2_ gas concentrations [150]. Reprinted from Tan, P.-H.; Hsu, C.-H.; Shen, Y.-C.; Wang, C.-P.; Liou, K.-L.; Shih, J.-R.; Lin, C.J.; Lee, L.; Wang, K.; Wu, H.-M.; Chiang, T.-Y.; Chih, Y.-D.; Chang, J.; King, Y.-C.; Chueh, Y.-L., Complementary Metal–Oxide–Semiconductor Compatible 2D Layered Film-Based Gas Sensors by Floating-Gate Coupling Effect. *Adv. Funct. Mater.*
**2022**, *32*, 2108878. © 2023 Wiley-VCH GmbH, with permission from John Wiley and Sons.

**Figure 7 sensors-23-08648-f007:**
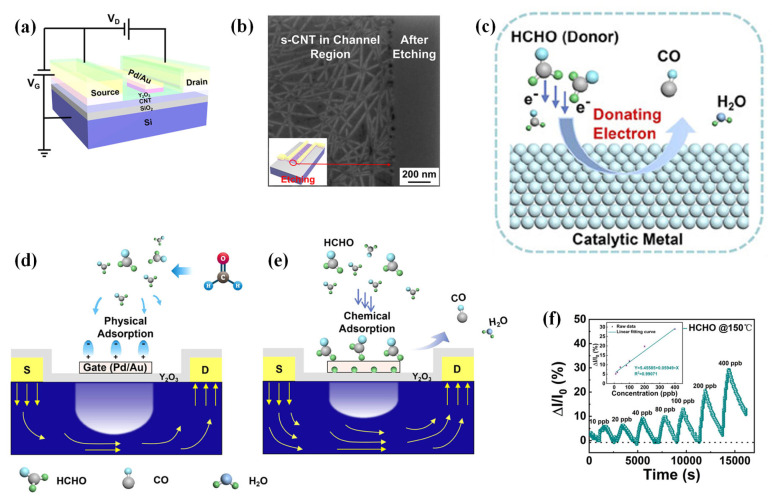
(**a**) Schematic diagram of the device, illustrating its various material layers and the biasing used for its characterization. (**b**) Image of the percolated CNT film acting as the semiconducting channel of the device. (**c**) Illustration of the sensing approach at the catalytic metal gate for detecting HCHO. Illustration of the sensing mechanism (**d**) at RT, primarily based on physisorption at the metal gate and (**e**) at 150 °C, based mainly on chemisorption. (**f**) Dynamic response and recovery of the CNT-FET HCHO sensor at 150 °C and, inset, the linear fitting curve demonstrating its sensitivity as a function of concentration [155]. Reprinted (adapted) with permission from Liu, C.; Hu, J.; Wu, G.; Cao, J.; Zhang, Z.; Zhang, Y. Carbon Nanotube-Based Field-Effect Transistor-Type Sensor with a Sensing Gate for Ppb-Level Formaldehyde Detection. *ACS Appl. Mater. Interfaces*
**2021**, *13*, 56319. https://doi.org/10.1021/acsami.1c17044. Copyright 2021 American Chemical Society.

**Figure 8 sensors-23-08648-f008:**
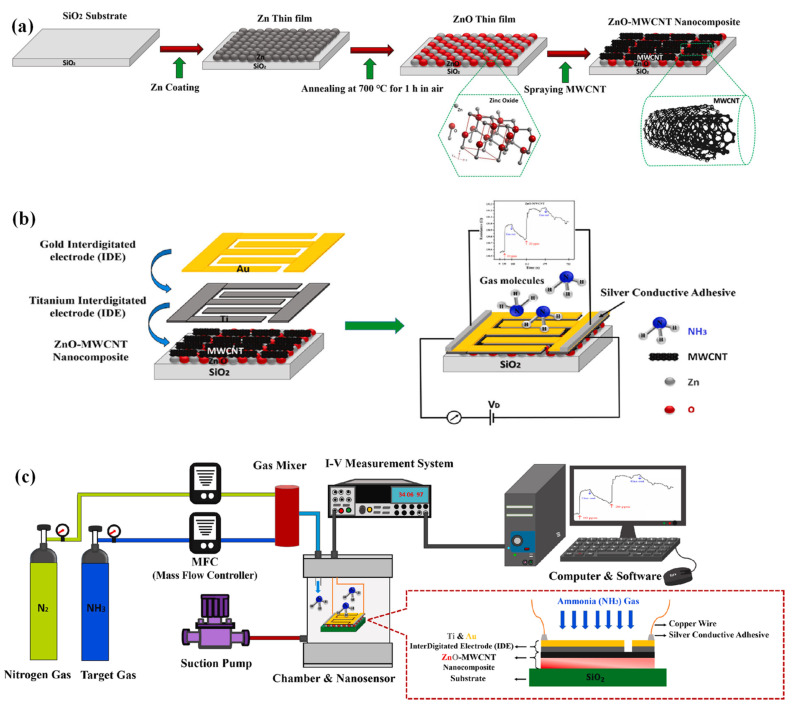
(**a**) The process of ZnO-MWCNT nanocomposite manufacturing. (**b**) Steps of Au and Ti IDE fabrication. (**c**) Schematic of the experiments [185]. Reprinted from Vatandoust, L.; Habibi, A.; Naghshara, H.; Aref, S.M. Fabrication of ZnO-MWCNT nanocomposite sensor and investigation of its ammonia gas sensing properties at room temperature. *Synth. Met.*
**2021**, *273*, 116710, Copyright 2021, with permission from Elsevier.

**Figure 9 sensors-23-08648-f009:**
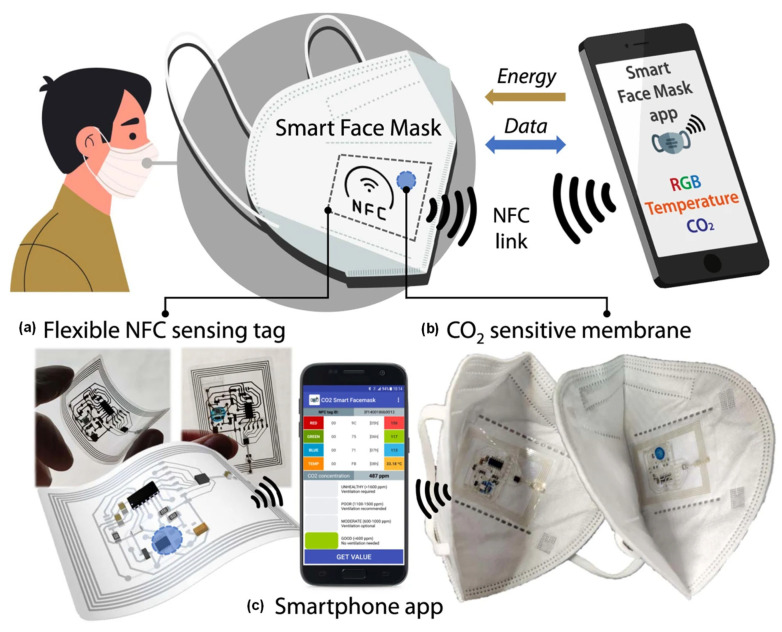
Overview of the NFC-based smart facemask for real-time wireless CO_2_ measurement. (**a**) NFC sensor tag that is flexible. (**b**) CO_2_ sensor membrane on a flexible tag connected to the inside layer of a regular FFP2 facemask. (**c**) Custom-developed smartphone applications provide power and enable bidirectional communication. Images reused from [226], with no changes. CC-BY 4.0 licence (https://creativecommons.org/licenses/by/4.0/, accessed on 8 October 2023).

**Table 1 sensors-23-08648-t001:** Gaseous pollutants detected by nano-based wearable gas sensors.

Gaseous Pollutant	Sensing Material	Substrate	Type of Sensor	Form Factor	Connectivity	Sensing Linear Range	Limit of Detection	Sensitivity	Ref.
H_2_S	CNTs/SnO_2_/CuO	Silica	Chemiresistor	Interdigitated electrodes (IDE)	Electronic	10–80 ppm	10 ppm	19%	[40]
NH_3_						NA	NA	4.1%	
CO						NA	NA	0.2%	
SO_2_						NA	NA	0.1%	
NH_3_	Pyrene/MWCNT (25%)SWCNT (75%)	Polyethene terephthalate (PET)	Chemiresistor	IDE on ceramic	Electronic	NA	40 ppm	NA	[41]
NO_2_						NA	10 ppm	NA	
NO_2_	Polystyrene brush graphene	SiO_2_/Si	Field-effect transistor (FET)	IDE	Electronic	0.1–1 ppm	4.8 ppb	45.11%	[44]
NH_3_						NA	NA	13.38%	
CO_2_						NA	NA	0.28%	
NO_2_	SiGNS-400	Au	Chemiresistor	IDE on a ceramic	Electronic	18 ppb–300 ppm	18 ppb	21.5%–50 ppm	[45]
NH_3_	Graphene	Polyvinylidene fluoride (PVDF)	Vertical graphene (VGr) FET	Silicon chip	Electronic	0–100 ppm	86 ppb	0.0195 ppm	[46]
NH_3_	BSA and PAMPS co-doped PANI layer	Bacterial cellulose nanofiber	Chemiresistor	Nanocomposite film	Wired	10–150 ppm	200 ppb	NA	[52]
NO_2_	PANI/Ag_2_O/GO	Interdigitated electrodes	Chemiresistor	Nanocomposite film	Electronic	5–50 ppm	25 ppm	5.85 for 25 ppm	[53]
CO	PPy/TiO_2_	Cu-interdigitated electrodes	Chemiresistor	Nanocomposite film	Electronic	1–320 ppm	1 ppm	93 at 270 ppm	[55]
BTEX	Polyethylene	Paper	Cantilever	Film	Phone camera	0–150 ppm	3–5 ppm	0.134 (±0.0027) μM ppm^−1^	[93]
NH_3_	Ppy/Au	Polyvinyl pyrrolidone	Chemiresistor	Flexible porous fiber	Wireless	1–800 ppm	1 ppm	2.3 ppm^−1^	[95]
H_2_S	Ppy/WO_3_	Cellulose acetate	Chemiresistor	Nanofibers and nanofilms	Electronic	1–50 ppm	1 ppm	NA	[96]
NO_2_	rGO/Triazine	PET	Chemiresistor	Film	Electronic	Up to 1000 ppm	2.2 ppb	452.6 ppm^−1^	[98]
NO_2_	Graphene/GO	Polyester	Chemiresistor	Sheets	Electronic	0–100 ppm	NA	0.34 μA/ppm	[106]
NH_3_								0.16 μA/ppm	
NO_2_	GO	Amyloid nanofibrils	Chemiresistor	Nanofibrils	Electronic	0–100 ppm	3–5 ppm	20.6 ± 0.07 nA/ppm	[103]
NO_2_	Dopamine/Graphene	Cotton	Chemiresistor	Nanofiber	Electronic	0–100 ppm	1 ppm	0.02 μA/ppm	[104]
NH_3_	5′,9′-dihydro-3H-spiro[isobenzofuran-1,15′-pyrano [2,3-b:6,5-b’]dicarbazol]-3-one (RhYK)	Polyester	Colorimetric/pH-sensitive	Stretchable textile	Visual	0–100 ppm	5 ppm	NA	[105]
NO_2_	SnO_2_/Pt/Ti	SiO_2_/Si	Electrochemical	Film	Electronic	10–400 μM		1.7 μM	[150]
NO_2_	SnSe_2_	Si	FGMOS	Silicon chip	Wired	1–25 ppm	1 ppm	15.42% ppm^−1^	[151]
NH_3_								102.3 mV ppm^−1^	
HCHO	10/10 nm-thick Pd/Au film	Si	Gate-sensing FET	Silicon chip	Wired	10–800 ppb	10–20 ppb	2.9–5.4% ppb^−1^	[155]
NO_2_	n-doped graphene	Polyethylene naphthalate (PEN)	Chemiresistor and channel-sensing FET	Flexible film	Wired	0.83 ppq–0.4 ppm	0.83 ppq	−23% ppm^−1^	[156]
SO_2_									
NO_2_	IGZO	Si	Channel-sensing FET	Silicon chip	Wired	100 ppb–5 ppm	100 ppb	~30% ppm^−1^	[158]
NH_3_	IDT-BT with TCNQ and ABN	PEN	Channel-sensing FET	Flexible film	Wired/Wireless	NA	10 ppm	23.14% ppm^−1^	[159]
NO_2_	SWCNT	Si	Channel-sensing FET	Silicon chip	Wired	0.56–10 ppm	0.92 ppm	1.97% ppm^−1^	[160]
NO_2_	MoS_2_	Si	Channel-sensing FET	Silicon chip	Wired	25 ppb–1 ppm	25 ppb	12.5% ppm^−1^	[161]
NO_2_	MoS_2_	Si	Channel-sensing FET	Silicon chip	Wired	1–256 ppm	1 ppm	NA	[162]
NH_3_	MoO_x_/Graphene	Si	Channel-sensing FET	Silicon chip	Wired	12–96 ppm	0.31–2.2 ppm	0.05–0.1% ppm^−1^	[163]
NH_3_	Hydroxyl-functionalized graphene quantum dots (OH-GQDs)	Alumina	Chemiresistor	Ni IDE	Electronic	10–500 ppm	NA	0.1459 ppm^−1^	[180]
NH_3_	Graphene nanowalls	Si	FET	Ti/Au electrodes	Electronic	2–50 ppm	2 ppm	4.53% at 10 ppm	[181]
NH_3_	Polyaniline nanofibers/SWCNT	Glass	Chemiresistor	Ag electrodes	Electronic	2–15 ppm	10 ppm	2.4% ppm	[182]
NH_3_	CNT	Quartz tuning fork (QTF)	Photoelectric	Piezoelectric transducer	Light-induced thermoelastic spectroscopy (LITES)	1–300 ppm	7.27 ppm for rGO	33.76 mV	[186]
	GO							34.81 mV	
	rGO							47.89 mV	
NH_3_	Meta-toluic acid/GO	Si/SiO_2_	Chemiresistor	Nanosheets	Electronic	100–3000 ppm	NA	32.7% at 100 ppm	[183]
NH_3_	Pyrrole/phthalocyanine/MWCNT	Au	Chemiresistor	IDE	Electronic	1–100 ppm	11.3 ppb	26.2% to 50 ppm	[184]
NH_3_	ZnO/MWCNT	SiO_2_	Chemiresistor	Ti/Au IDE	Electronic	10–20 ppm	10 ppm	1.022 at 10 ppm	[185]
NH_3_	Sulfur-containing C60 fullerene	Glass	Organic field-effect transistor (OFET)	IDE	Electronic	NA	1 ppm	NA	[187]
NH_3_	Poly(3,4-ethylene dioxythiophene): poly(styrene sulfonate) (PEDOT: PSS)/iridium oxide (IrO_x_)	PET	Chemiresistor	Flexible film	Bluetooth	17–7899 ppm	8 ppm	0.8% ppm^−1^	[188]
CO	Hybrid graphene/ZnO	Flexible cotton fabric	Resistive	Flexible	Visual	10–90 ppm	9 ppm	NA	[220]
CO_2_	Hydroxypropyl methylcellulose	PET	Opto-chemical	Flexible	Wireless smart phone	500–48,000 ppm	221 ppm	100 ppm	[226]
H_2_S	NO_2_-UiO-66/CNT	Polyacrylonitrile	Chemiresistor	Flexible	Electronic	0–100 ppm	10 ppb	10 ppb	[229]
NH_3_	Copper (II) complex of the azo dye 1-(2-pyridylazo)-2-naphtol (Cu-PAN)	Paper	Colorimetric/pH sensitive	Printed paper	Wireless/Phone camera	5–100 ppm	>100 ppm	1.3%/ppm	[230]
Formaldehyde							NA	NA	
H_2_S									
SO_2_	UiO-66-NH_2_	Polyvinylidene fluoride	Chemiresistor	Nanofibers	Electronic	1–150 ppm	1 ppm	85% at 50 ppm	[233]
O_3_	NA	Aluminum-coated iHWG	Photoelectric	Hand-held	USB connection	0.4–21 ppm	29 ppb	NA	[239]
VOC	Mechanochromic AIE luminogen	Cellulose	Mechanochromic	Wearable	Visual	NA	5 ppm	NA	[244]
VOC	NA	NA	Chemiresistor	Wearable	Wireless	0–10 ppm	5 ppm	NA	[245]
HF	Maleimide	Cotton	Chemiresistor	Fabric	Visual	NA	NA	NA	[241]
NH_3_	Au-doped SnO_2_	Si	Micro-hotplate chemiresistive	Silicon chip on RFID tag PCB	RFID	2–70 ppm	2 ppm	0.2–0.8 ppm^−1^	[250]

NA: Not applicable.

## Data Availability

Data sharing not applicable.

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
