# Peer review of "Wearable Nano-Based Gas Sensors for Environmental Monitoring and Encountered Challenges in Optimization"

_sensors, 2023, doi:10.3390/s23208648_

Round 1

Reviewer 1 Report

The novelty of this review has not been sufficiently emphasized in the current version.

  The authors need to provide the contributions of this study more specifically.

Author Response

Response to reviewer comments

We would like to thank the editors and the reviewer for their time and effort in helping us to improve our manuscript. Please find below our point-by-point response to the reviewer comments. Changes made to the manuscript have also been highlighted with yellow in the main text of the paper for your convenience.

Reviewer 1

We the authors of this review paper would like to thank you for your time to review our manuscript and help us to improve this paper, it is very much appreciated.

Comment 1:  The novelty of this review has not been sufficiently emphasized in the current version.

Response: We have revised the last paragraph of the Introduction to address this comment and further emphasize on novelty of this review paper; The realm of gas-sensing has been extensively explored in several reviews [12–15]. Yet, a comprehensive examination of wearable gas-sensing devices, along with their associated challenges, remains conspicuously absent, particularly from a holistic standpoint. This gap is noteworthy, especially as the future trajectory is evidently leaning towards the integration of gas-sensing with smart wearable personalized technologies. Notably, there has been a review on the recent advancements in flexible room-temperature gas sensors grounded on metal oxide semiconductors [16]. Furthermore, cutting-edge methodologies solely cantered on metal oxide-based heterostructures for room temperature gas sensors have been discussed [17]. Additionally, a detailed analysis has been undertaken regarding gas sensors and the dynamics that impact their sensing mechanisms, with a special emphasis on metal oxide semiconductors [18]. The uniqueness of our paper lies in its broad perspective, bridging the existing literature gap and highlighting the novel integration of wearable technology in the gas-sensing domain.

This paper presents a cutting-edge review of advancements made since 2020 in the realm of smart wearable (nano)sensors tailored for environmental monitoring of gaseous pollutants. Notably, this comprehensive survey underscores the pioneering nature of these wearables in tackling key challenges associated with their development and commercialization. These challenges encompass sensor sensitivity, selectivity, wearability, integration, miniaturization, cost implications, power consumption, and the quest for high-performance, biocompatible power supplies. Such power sources must closely align with the sensing device in terms of weight, flexibility, durability, and other vital attributes [19].

This review is systematically structured into distinct sections for ease of understanding:

  1. Gas Sensing Materials: This section delves into a variety of materials, including 2D nanostructures, carbon nanomaterials, conducting polymers (CPs), nanohybrids, and metal oxide semiconductors (MOS).
  2. Wearable Substrates & Electrodes: Here, different substrates such as paper based (PB) mediums, polymeric materials, textiles, and stretchable electronics have been explored that serve as pivotal components in wearable sensors.
  3. Sensor Types: covers an in-depth analysis of diverse sensors – from colorimetric, chem-optical, and electrochemical to transistors and chemiresistors. Emphasis is laid on the critical role these sensors play in monitoring environmental gaseous pollutants, including but not limited to ammonia (NH3), nitric oxide (NO), nitrous oxide (N2O), nitrogen dioxide (NO2), carbon monoxide (CO), CO2, H2S, sulfur dioxide (SO2), ozone (O3), hydrocarbons (CxHy), hydrogen fluoride (HF). The significance of deploying proficient detection methods for these pollutants is accentuated.
  4. Challenges in Gas Sensor Optimization: Distinct from existing literature, this section offers a meticulous discussion on challenges that have remained largely unexplored. Topics such as circuit integration, real-time sensing, repeatability, power consumption, gas-sensitive material deposition, and stability, among others, are examined in detail.

The uniqueness of this review paper lies in its holistic approach, addressing numerous facets of gas sensor technology in a singular, comprehensive document. We envision that this will significantly benefit researchers and industry experts, steering them towards pioneering the next generation of gas sensors.

Comment 2:  The authors need to provide the contributions of this study more specifically.

Response: As mentioned in our response to your previous comment and in the edited section of the Introduction of our paper, the contributions of this review are:

  1. A uniquely broad perspective, bridging the existing literature gap and highlighting the novel integration of wearable technology in the gas-sensing domain. This review is systematically structured into distinct sections covering aspects related to:
    1. Gas Sensing Materials.
    2. Wearable Substrates & Electrodes.
    3. Sensor Types.
    4. Challenges in Gas Sensor Optimization.
  2. A cutting-edge review of advancements made since 2020 in the realm of smart wearable (nano)sensors tailored for environmental monitoring of gaseous pollutants.
  3. A uniquely holistic approach, addressing numerous facets of gas sensor technology in a singular, comprehensive document.

We envision that this will significantly benefit researchers and industry experts, steering them towards pioneering the next generation of gas sensors.

Reviewer 2 Report

In the manuscript titled “Wearable Nano-based Gas Sensors for Environmental Monitoring and Encountered Challenges in Optimization” (Manuscript ID: sensors-2611509), the authors provide a review of recent developments in smart wearable nano-sensors for the detection of multiple gaseous species. The paper discusses some current developments in the gas sensing field and appears to be useful for readers. However, I have some questions that should be addressed. Therefore, I recommend a major revision of the manuscript. Please find additional comments and suggestions for the authors below.

1.      Although no critical grammar issues were identified, I recommend a detailed English review to improve sentence structure and enhance clarity and readability.

2.      The manuscript presents several structural issues that require revision:

a) It is observed multiple times that a Figure divides a paragraph (e.g., Figures 3 and 4).

b) Some paragraphs are excessively long (e.g., Introduction, section 5.3, and the first paragraphs of sections 4.3 and 4.5) or too short (e.g., the first paragraph on page 8).

c) The title of section 2.1 should be corrected.

d) Section 2.4 is not properly formatted.

d) Authors must double-check that all acronyms are defined before using them.

3.      The numbering of figures in the manuscript must differ from those in the cited article to avoid misunderstanding through citation (lines 651 and 652 on page 18).

4.      The authors discuss 2D material applications but overlook the use of one-dimensional (1D) nanostructures as gas sensors. Please consider adding a section on this topic.

5.      The purpose of this review should be clearly stated in the abstract and/or introduction section. It should also clearly demonstrate the aspects that differ from previously reported reviews. The authors should also mention the gaps in the field and their suggestions.

Although no critical grammar issues were identified, I recommend a detailed English review to improve sentence structure and enhance clarity and readability.

Author Response

Reviewer 2

We, the authors, sincerely appreciate the time and effort you dedicated to reviewing our manuscript, which has greatly contributed to enhancing its quality.

Comment 1: Although no critical grammar issues were identified, I recommend a detailed English review to improve sentence structure and enhance clarity and readability.

Response: We have meticulously proofread and enhanced this manuscript to ensure its clarity and readability.

Comment 2: The manuscript presents several structural issues that require revision:

  1. a) It is observed multiple times that a Figure divides a paragraph (e.g., Figures 3 and 4).
  2. b) Some paragraphs are excessively long (e.g., Introduction, section 5.3, and the first paragraphs of sections 4.3 and 4.5) or too short (e.g., the first paragraph on page 8).
  3. c) The title of section 2.1 should be corrected.
  4. d) Section 2.4 is not properly formatted.
  5. e) Authors must double-check that all acronyms are defined before using them.

Response: Thank you for highlighting these issues.

  • We have moved the figures such that they are located after the end of a paragraph. Thank you for pinpointing this out.
  • We have edited the problematic text sections as suggested. We have split the sections into multiple paragraphs and also split log sentences. These changes will improve the readability and flow of out paper, thank you.
  • The title of section 2.1 has been corrected.
  • The formatting of section 2.4 has been corrected. This issue was probably introduced when the journal’s editing team transferred our manuscript to the journal’s template. Together with the figure placement they will be dealt with before publication, should the paper get accepted. Thank you for highlighting this out, we will ensure such issues ae not present in the final document.
  • We have gone through the manuscript and we have ensured that all acronyms are defined that that they are first defined before being used.

Comment 3: The numbering of figures in the manuscript must differ from those in the cited article to avoid misunderstanding through citation (lines 651 and 652 on page 18).

Response: Thank you for highlighting this. The original figure number from the referenced paper was included by accident. This has now been deleted.

Comment 4: The author discusses 2D material applications but overlook the use of one-dimensional (1D) nanostructures as gas sensors. Please consider adding a section on this topic.

Response: While the use of one-dimensional (1D) nanostructures as gas sensors is not addressed as a standalone topic, section 2.2 titled 'Carbon nanomaterials' provides a detailed review of CNTs, particularly in subsection 2.2.2. Additionally, section 2.5 discusses the application of 1D ZnO nanostructures for NO2 detection, as cited in references 66 and 67. Recent studies in the field of gas sensing have also focused on the use of one-dimensional semiconductors. By increasing the surface area, these studies have shown enhanced adsorption of analyte molecules, subsequently improving the gas sensing response of the device, as discussed in reference 68. We are now mentioning that these are 1D materials highlighting their advantages.

Comment 5: The purpose of this review should be clearly stated in the abstract and/or introduction section. It should also clearly demonstrate the aspects that differ from previously reported reviews. The authors should also mention the gaps in the field and their suggestions.

Response: to better reflect the purpose of this review paper the abstract and introduction have been revised as following:

“Abstract: With a rising emphasis on public safety and quality of life, there is an urgent need to ensure optimal air quality, both indoors and outdoors. Detecting toxic gaseous compounds plays a pivotal role in shaping our sustainable future. This review aims to elucidate the advancements in smart wearable (nano)sensors for monitoring harmful gaseous pollutants, such as ammonia (NH3), nitric oxide (NO), nitrous oxide (N2O), nitrogen dioxide (NO2), carbon monoxide (CO), carbon dioxide (CO2), hydrogen sulfide (H2S), sulfur dioxide (SO2), ozone (O3), hydrocarbons (CxHy), and hydrogen fluoride (HF). Differentiating this review from its predecessors, we shed light on the challenges faced in enhancing sensor performance and offer a deep dive into the evolution of sensing materials, wearable substrates, electrodes, and types of sensors. Noteworthy materials for robust detection systems encompass 2D nanostructures, carbon nanomaterials, conducting polymers, nanohybrids, and metal oxide semiconductors. A dedicated section dissects the significance of circuit integration, miniaturization, real-time sensing, repeatability, reusability, power efficiency, gas-sensitive material deposition, selectivity, sensitivity, stability, and response/recovery time, pinpointing gaps in the current knowledge and offering avenues for further research. Concluding, we provide insights and suggestions for the prospective trajectory of smart wearable nanosensors in addressing the extant challenges.”

In addition, the following statement has been added to the introduction section:

“The realm of gas-sensing has been extensively explored in several reviews [12–15]. Yet, a comprehensive examination of wearable gas-sensing devices, along with their associated challenges, remains conspicuously absent, particularly from a holistic standpoint. This gap is noteworthy, especially as the future trajectory is evidently leaning towards the integration of gas-sensing with smart wearable personalized technologies. Notably, there has been a review on the recent advancements in flexible room-temperature gas sensors grounded on metal oxide semiconductors [16]. Furthermore, cutting-edge methodologies solely centered on metal oxide-based heterostructures for room temperature gas sensors have been discussed [17]. Additionally, a detailed analysis has been undertaken regarding gas sensors and the dynamics that impact their sensing mechanisms, with a special emphasis on metal oxide semiconductors [18]. The uniqueness of our paper lies in its broad perspective, bridging the existing literature gap and highlighting the novel integration of wearable technology in the gas-sensing domain.

This paper presents a cutting-edge review of advancements made since 2020 in the realm of smart wearable (nano)sensors tailored for environmental monitoring of gaseous pollutants. Notably, this comprehensive survey underscores the pioneering nature of these wearables in tackling key challenges associated with their development and commercialization. These challenges encompass sensor sensitivity, selectivity, wearability, integration, miniaturization, cost implications, power consumption, and the quest for high-performance, biocompatible power supplies. Such power sources must closely align with the sensing device in terms of weight, flexibility, durability, and other vital attributes [19].

This review is systematically structured into distinct sections for ease of understand-ing:

  1. Gas Sensing Materials: This section delves into a variety of materials, including 2D nanostructures, carbon nanomaterials, conducting polymers (CPs), nanohybrids, and metal oxide semiconductors (MOS).
  2. Wearable Substrates & Electrodes: Here, different substrates such as paper based (PB) mediums, polymeric materials, textiles, and stretchable electronics have been explored that serve as pivotal components in wearable sensors.
  3. Sensor Types: covers an in-depth analysis of diverse sensors – from colorimetric, chem-optical, and electrochemical to transistors and chemiresistors. Emphasis is laid on the critical role these sensors play in monitoring environmental gaseous pollutants, including but not limited to ammonia (NH3), nitric oxide (NO), nitrous oxide (N2O), nitro-gen dioxide (NO2), carbon monoxide (CO), CO2, H2S, sulfur dioxide (SO2), ozone (O3), hydrocarbons (CxHy), hydrogen fluoride (HF). The significance of deploying proficient detection methods for these pollutants is accentuated.
  4. Challenges in Gas Sensor Optimization: Distinct from existing literature, this section offers a meticulous discussion on challenges that have remained largely unexplored. Topics such as circuit integration, real-time sensing, repeatability, power consumption, gas-sensitive material deposition, and stability, among others, are examined in detail.

The uniqueness of this review paper lies in its holistic approach, addressing numerous facets of gas sensor technology in a singular, comprehensive document. We envision that this will significantly benefit researchers and industry experts, steering them towards pioneering the next generation of gas sensors.”

Reviewer 3 Report

The authors undertook to write a review describing various types of sensors, various sensitive materials, and various analytical tasks. To write such a review, it is necessary to have extremely high competence and be a professional in many fields of science. Did the authors manage to cope with the task? In part, however, many problems were described at an insufficiently high level. Some phrases are inserted from the cited articles without deep understanding. Some links do not refer to the original articles, but to other reviews. Whether the level of this review is high enough for publication in the journal of the first quartile?

Here are some examples:

1. <Regarding gas-sensitive materials, metal oxide semiconductors (MOS, such as ZnO, SnO2, and In2O3, etc.) have been extensively studied for gas detection (Kumar et al., 2015b, Xie et al., 2021, Yuan et al., 2019).>

It may give the impression that these authors first guessed to use ZnO, SnO2, and In2O3 as gas-sensitive materials, although these materials have been used for a long time and have long been well described.

2. <Zinc oxide (ZnO), as a commonly used n-type semiconductor, has plenty of advantages such as cost-efficiency, non-toxicity, high response, and good stability, resulting in a promising material in NO2 detection (Ahmad et al., 2019). Jeon et al. fabricated resistive-type semiconductor gas sensors based on ZnO nanofilms that show high sensitivity to NO2 (~ 84.2%) compared to NH3 (~ 45.9%), CH4 (~ 35.8%), H2 (~ 29.2%), and C3H8 (~ 31.2%) at the optimal operating temperature of 250 °C (Jeon et al., 2021).> 

The phrase is mechanically taken out of the context of the cited article and does not carry useful information. The reader can conclude that ZnO is definitely preferable to other metal oxide materials. It would be correct to give responses for different metal oxide materials under comparable conditions in order to draw such conclusions. In addition, information about the magnitude of the response does not make sense at all without providing information about the sensor temperature and gas concentration.

3. <MOSs (e.g., CuO, ZnO, TiO2, SnO2 and In2O3, etc.) have also been exploited as a sensing interface in surface acoustic wave (SAW)-based detection of gases such as NOx, NH3 and H2S [72]>

In reference [72], the MOS-sensor and surface acoustic wave (SAW) are separated, not glued together. How correct is it in the review to refer not to the original article, but to another review [72]? Besides, I failed to find such information in [72].

4. <The most common family of materials used for chemiresistive gas sensors, are MOSs, such as SnO2, ZnO, In2O3, TiO2, CeO2, Fe2O3, WO3, CdO, CuO>

CeO2 is used in sensors (for example, electrochemical), but rarely in chemiresistive, except in the form of catalytic additives. The same can be said about Fe2O3, CdO, CuO. They are commonly used in other types of sensors. To write about CeO2, Fe2O3, CdO, CuO that this is <the most common family of materials used for chemiresistive gas sensors> is a clear exaggeration.

5. The authors write about selectivity without separating the concepts of sensor selectivity and analysis selectivity, which can also be achieved by low-selective sensors.

6. The authors should indicate the preferred applications for different types of sensors, as well as indicate the prospects for the development of sensory research based on the collected data.

Author Response

Reviewer 3

We would like to thank you for your time and effort you invested in reviewing our manuscript.

Comments and Suggestions for Authors: The authors undertook to write a review describing various types of sensors, various sensitive materials, and various analytical tasks. To write such a review, it is necessary to have extremely high competence and be a professional in many fields of science. Did the authors manage to cope with the task? In part, however, many problems were described at an insufficiently high level. Some phrases are inserted from the cited articles without deep understanding. Some links do not refer to the original articles, but to other reviews. Whether the level of this review is high enough for publication in the journal of the first quartile?

Here are some examples:

Comment 1: <Regarding gas-sensitive materials, metal oxide semiconductors (MOS, such as ZnO, SnO2, and In2O3, etc.) have been extensively studied for gas detection (Kumar et al., 2015b, Xie et al., 2021, Yuan et al., 2019).> It may give the impression that these authors first guessed to use ZnO, SnO2, and In2O3 as gas-sensitive materials, although these materials have been used for a long time and have long been well described.

Response: Section 2.5 that is dedicated to MOS materials originally mentioned in its 1st paragraph the following sentences:

“The use of metal oxide gas sensors goes back to 1962, and it was realised that the resistance of such films changes in the presence of CO2, toluene and propane gases at a specific temperature. This chemiresistive characteristic of semiconductor materials has made them ideal candidates for gas sensing. Initially, MOS gas sensors suffered from high power consumption and complicated fabrication processes, which hindered their application. However, in 1968 gas sensors based on SnO2 were commercialized.”

We have now supported the above staements with references and added this sentence as well:

“As a result, MOS materials and their use in gas sensing have been extensively studied over several decades.”

We believe that the above should be sufficient to establish to the unfamiliar readership that these are well known materials used in gas sensing over many many years and that have been well studied.

One of the primary goals of a review paper is to offer readers a thorough overview of the field. Consequently, we felt it was crucial to incorporate both traditional and well-established materials from the domain. This inclusion is particularly pertinent as metal oxide semiconductors (MOSs) like ZnO, SnO2, and In2O3 are not only cost-effective but also exhibit high sensitivity and are straightforward to fabricate (reference 65).

Comment 2: <Zinc oxide (ZnO), as a commonly used n-type semiconductor, has plenty of advantages such as cost-efficiency, non-toxicity, high response, and good stability, resulting in a promising material in NO2 detection (Ahmad et al., 2019). Jeon et al. fabricated resistive-type semiconductor gas sensors based on ZnO nanofilms that show high sensitivity to NO2 (~ 84.2%) compared to NH3 (~ 45.9%), CH4 (~ 35.8%), H2 (~ 29.2%), and C3H8 (~ 31.2%) at the optimal operating temperature of 250 °C (Jeon et al., 2021).> The phrase is mechanically taken out of the context of the cited article and does not carry useful information. The reader can conclude that ZnO is definitely preferable to other metal oxide materials. It would be correct to give responses for different metal oxide materials under comparable conditions in order to draw such conclusions. In addition, information about the magnitude of the response does not make sense at all without providing information about the sensor temperature and gas concentration.

Response: Thank you for your insightful comments. We recognize the inadvertent implications the statement might have made regarding the superiority of ZnO over other metal oxide materials. We have now revised the segment to ensure that it provides a more balanced viewpoint, emphasizing the need for comparative studies and considering essential parameters like operating temperature and gas concentration. We believe this revised version offers readers a more nuanced and informed perspective on the topic. Your feedback was invaluable, and we appreciate the opportunity to clarify and improve our work.

" Nanostructured ZnO-based MOS gas sensors, representative of n-type semiconductors, offer certain advantages including non-toxicity, cost-efficiency, rapid response, and stability. These attributes make them notable candidates in the realm of gas sensing, especially for wearable devices. 1D ZnO nanostructure-based sensors have demonstrated the ability to detect gases like NO2, NH3, CH4, H2, and C3H8, though the sensitivity largely depends on specific operating temperatures [73,74]. However, it is crucial to highlight that the choice of the best material is influenced by application-specific requirements. Comparing performance ZnO with other metal oxide materials under standardized conditions would be imperative for comprehensive analysis. In addition to ZnO-based 1D nanomaterials, advancements in gas sensing also focus on other one-dimensional semi-conductors, such as Pd-doped 1D tungsten oxide nanowires [75]. Overall, metal oxide nanowires have been gaining attention for their stable chemical properties and considerable surface-to-volume ratio. By increasing the surface area, these studies have shown enhanced adsorption of analyte molecules, subsequently improving the gas sensing re-sponse of the device. . For a holistic understanding, it is essential to consider parameters like sensor temperature and gas concentration when evaluating the magnitude of the response.”

Comment 3: <MOSs (e.g., CuO, ZnO, TiO2, SnO2 and In2O3, etc.) have also been exploited as a sensing interface in surface acoustic wave (SAW)-based detection of gases such as NOx, NH3 and H2S [72]>In reference [72], the MOS-sensor and surface acoustic wave (SAW) are separated, not glued together. How correct is it in the review to refer not to the original article, but to another review [72]? Besides, I failed to find such information in [72].

Response: We have added several review apers covering the subject to support this statement and removed reference 72 from being referenced here. Also we have changed the sentence sightly:.

“As reviewed elsewhere, MOSs (e.g., CuO, ZnO, TiO2, SnO2 and In2O3, etc.) have also been exploited as a sensing interface in surface acoustic wave (SAW)-based detection of gases such as NOx, NH3 and H2S”

Comment 4:  <The most common family of materials used for chemiresistive gas sensors, are MOSs, such as SnO2, ZnO, In2O3, TiO2, CeO2, Fe2O3, WO3, CdO, CuO>CeO2 is used in sensors (for example, electrochemical), but rarely in chemiresistive, except in the form of catalytic additives. The same can be said about Fe2O3, CdO, CuO. They are commonly used in other types of sensors. To write about CeO2, Fe2O3, CdO, CuO that this is <the most common family of materials used for chemiresistive gas sensors> is a clear exaggeration.

Response: Thank you for your constructive feedback on our manuscript. We have revised the statement regarding the materials used in chemiresistive gas sensors to better represent their application and relevance. The revised section now specifically emphasizes the common materials like SnO2, ZnO, In2O3, TiO2, and WO3, while also clarifying the roles of CeO2, Fe2O3, CdO, and CuO.

"The MOS family, including materials such as SnO2, ZnO, In2O3, TiO2, and WO3, is frequently used in chemiresistive gas sensors. While CeO2, Fe2O3, CdO, and CuO are also employed in various sensor applications, their primary use in chemiresistive gas sensors is often as catalytic additives rather than primary sensing materials."

Comment 5: The authors write about selectivity without separating the concepts of sensor selectivity and analysis selectivity, which can also be achieved by low-selective sensors.

Response: We recognize the distinction between sensor selectivity and analysis selectivity. In this context, within the most of the paper we focused on sensor selectivity, defining it as the sensor's ability to discriminate the target from interfering molecules and produce a target-specific response. In section 6.6 we defined selectivity and how this can be enhanced. To address your comments and to make the distinction more evident, per your suggestion, we have added the following highlighted text in section 6.6.:

Selectivity is an essential aspect of any sensing technology. A sensor needs to provide sensitivity, or at least the primary sensitivity, from only one specific event, e.g., the change in the concentration of one target gas within a mixture of gases. If the sensor is sensitive to more than one analyte, then it may not be possible to distinguish what the recorded response is from and to reach realistic conclusions. However, it may be that different analytes lead to different responses in the measured output signal, such as different ranges or different transient behaviors that can lead to safe conclusions. Nevertheless, this is not ideal, as interpreting the measured data can be challenging and may lead to ambiguities. The ratio of sensor signal response the target gas reaches to that achieved with other interfering gases is often used to define selectivity [265]. The above can be described as the intrinsic sensor selectivity.

Principal component analysis, machine learning and other statistical approaches can be exploited to address the above issues and enhance measurement interpretation  [266]. Such data processing approaches can be categorized as analysis selectivity enhancement approaches, that when possible can enhance the selectivity of sensor with low intrinsic selectivity.

Comment 6: The authors should indicate the preferred applications for different types of sensors, as well as indicate the prospects for the development of sensory research based on the collected data.

Response: We have discussed the preferred applications for different types of sensors in section 5 titled "Environmental Gaseous Pollutants Monitoring". This section explores critical gaseous contaminants and their significance, specifically tailored for environmental sensing applications. For each pollutant, recent examples are highlighted. Consequently, we believe these xtensive discussions are already sufficient to address the first part of your comment. For a more comprehensive view on the future prospects and direction of sensory research, please refer to section 7 titled "Conclusions and Outlook". We have further enhanced this section by adding the following sections:

Important challenges are also related with the power consumption of such sensors, which becomes a critical and limiting parameter for wearable applications. Selectivity, sensitiv-ity and multi-parametric sensing are crucial aspects for a robust gas sensor that can be successfully commercially launched, aspects that also become challenging to achieve in a wearable format. Sensor lifetime, robustness and contamination issues are also challenges crucial for wearable sensors, as well as motion artifacts that could arise from sensitivity to mechanical perturbations. Since, in some cases, used sensors should be disposable, sensor fabrication costs should be reduced as much as possible, while biodegradability and ecological impact should also be considered. Co-integration in a miniaturized and wear-able (flexible or stretchable) format of all aspects required for a wearable implementation, including battery or energy scavenging, sensor, front-end (analog) and back-end (digital) as well as radio frequency (RF, antenna, transceiver, etc.) components is the current challenge that needs to be addressed together with the above mentioned aspects, for the next generation of wearable gas sensors.

We hope this addresses your concerns and provides clarity on the subject matter.

Reviewer 4 Report

This review (sensors-2611509) focuses on wearable nano-based gas sensors for environmental monitoring. This topic is a hot research field and has a wide range of readers, including in the fields of nanomaterials, gas sensors and wearable electronics. But there are some issues to be addressed or clarified before possible publication. My specific comments are as follows:

1. “wearable” needs to be defined and discussed or clarified. Do wearable sensors need to meet flexibility? When the sensor size is small enough, it seems that the rigid sensor also meets the wearable requirements.

2. Introduction: The authors should not neglect the previous review papers in this field and suggest discussing and distinguishing them to highlight the necessity.

3. “2.1. 2D nanostructures” instead of “2.1.2. D nanostructures”

4. “4. Sensor types”:Self-powered triboelectric type and mixed potential type should not be ignored. As a review paper, it is recommended to provide relevant parameter definitions for different gas sensors, referring to ref. 157.

5. “4.2. Optical”: It is recommended to delete this section as it is difficult to comply with wearability.

6. Table 1: The table content is not displayed completely.

7. “7. Conclusions and outlook” need to be strengthened further. The influence of humidity needs to be discussed, and may refer to Small 2023, 19, 2303631. High performance metal oxide-based room temperature gas sensor. Temperature and humidity compensation algorithm, etc. In addition, readers would like to see more constructive strategies. For example, principles of material selection, materials and device manufacturing processes, humidity resistance and application prospects.

8. References list: The literature on this subject is not complete, and it is recommended to further search the literature. In addition, it may be necessary to consider setting certain restrictions on the publication time of the cited literature, such as focusing on the past 5 years.

9. Check the reference format of the journal, such as providing complete titles.

10. English needs polishing.

Minor editing of English language required

Author Response

Reviewer 4

We truly appreciate the effort and dedication you put into evaluating our manuscript, which has greatly enhanced its quality.

Comment 1: “wearable” needs to be defined and discussed or clarified. Do wearable sensors need to meet flexibility? When the sensor size is small enough, it seems that the rigid sensor also meets the wearable requirements.

Response: Thank you for the valuable feedback. To further elaborate on the definition of “wearable”, the following statement has been added to the manuscript:

“In terms of their mechanical properties, sensors can generally be categorized as either rigid, flexible, or stretchable depending on the properties of the main substrate of the sensor. All three categories can be used in wearable applications, depending on the re-quirements and specifications (e.g. lifetime, duration of use, mechanical perturbation that can be tolerated) of the application and the body area they are intended to worn on. Flexible and stretchable sensors can be composed of several rigid parts interconnected with flexible/stretchable electrical interconnects in an island-bridge approach. Rigid sub-strates and devices can be thinned to make them flexible to some extent, while use of in-trinsically rigid devices can be worn on body areas not susceptible to mechanical defor-mations (bending) such as the lower back, the head, and areas away from joints. In addi-tion, making a device sufficiently small compared to the scale of mechanical deformation and motion, can render a rigid device wearable and insensitive to the motion of the body. For example, see Section 4.4 on microelectronics approaches. When device scales are larger than the body curvature, and not much mechanical deformation due to body mo-tion is expected, flexible devices are needed. However, when large deformations are ex-pected as in joints and during exercise routines and sports, stretchability is required. In addition, large rigid devices, due to the significant difference in their mechanical proper-ties with skin and the underlying tissue, will lead to injuries for the wearer, while the rigid device will also become damaged and will malfunction. Consequently, when the overall sensing system size cannot be sufficiently small, the development of low-cost, flexible, or stretchable wearable, and high-performance flexible gas sensors is necessary. Wearable sensors can also be integrated into clothing or accessories in addition to being worn di-rectly on the skin, depending on the needs of the intended use.”

Comment 2: Introduction: The authors should not neglect the previous review papers in this field and suggest discussing and distinguishing them to highlight the necessity.

Response: Thank you for pointing out the oversight. We have now included discussions on the previous review papers in this field. References 12-18 have been added to the Introduction to highlight the distinctions and emphasize the necessity of our work in the context of the existing literature.  We have also included the following text in the introduction to further highlight the uniqueness of our review paper:

“The realm of gas-sensing has been extensively explored in several reviews [12–15]. Yet, a comprehensive examination of wearable gas-sensing devices, along with their as-sociated challenges, remains conspicuously absent, particularly from a holistic standpoint. This gap is noteworthy, especially as the future trajectory is evidently leaning to-wards the integration of gas-sensing with smart wearable personalized technologies. Notably, there has been a review on the recent advancements in flexible room-temperature gas sensors grounded on metal oxide semiconductors [16]. Further-more, cutting-edge methodologies solely centered on metal oxide-based heterostructures for room temperature gas sensors have been discussed [17]. Additionally, a detailed analysis has been undertaken regarding gas sensors and the dynamics that impact their sensing mechanisms, with a special emphasis on metal oxide semiconductors [18]. The uniqueness of our paper lies in its broad perspective, bridging the existing literature gap and highlighting the novel integration of wearable technology in the gas-sensing domain.

This paper presents a cutting-edge review of advancements made since 2020 in the realm of smart wearable (nano)sensors tailored for environmental monitoring of gaseous pollutants. Notably, this comprehensive survey underscores the pioneering nature of these wearables in tackling key challenges associated with their development and com-mercialization. These challenges encompass sensor sensitivity, selectivity, wearability, integration, miniaturization, cost implications, power consumption, and the quest for high-performance, biocompatible power supplies. Such power sources must closely align with the sensing device in terms of weight, flexibility, durability, and other vital attributes [19].

This review is systematically structured into distinct sections for ease of under-standing:

  1. Gas Sensing Materials: This section delves into a variety of materials, including 2D nanostructures, carbon nanomaterials, conducting polymers (CPs), nanohybrids, and metal oxide semiconductors (MOS).
  2. Wearable Substrates & Electrodes: Here, different substrates such as paper based (PB) mediums, polymeric materials, textiles, and stretchable electronics have been explored that serve as pivotal components in wearable sensors.
  3. Sensor Types: covers an in-depth analysis of diverse sensors – from colorimet-ric, chem-optical, and electrochemical to transistors and chemiresistors. Emphasis is laid on the critical role these sensors play in monitoring environmental gaseous pollutants, including but not limited to ammonia (NH3), nitric oxide (NO), nitrous oxide (N2O), ni-trogen dioxide (NO2), carbon monoxide (CO), CO2, H2S, sulfur dioxide (SO2), ozone (O3), hydrocarbons (CxHy), hydrogen fluoride (HF). The significance of deploying proficient detection methods for these pollutants is accentuated.
  4. Challenges in Gas Sensor Optimization: Distinct from existing literature, this section offers a meticulous discussion on challenges that have remained largely unex-plored. Topics such as circuit integration, real-time sensing, repeatability, power con-sumption, gas-sensitive material deposition, and stability, among others, are examined in detail.

The uniqueness of this review paper lies in its holistic approach, addressing nu-merous facets of gas sensor technology in a singular, comprehensive document. We envi-sion that this will significantly benefit researchers and industry experts, steering them towards pioneering the next generation of gas sensors.”

Comment 3: “2.1. 2D nanostructures” instead of “2.1.2. D nanostructures”

Response: The section title has been corrected to "2.1. 2D nanostructures" as per the reviewer's suggestion. Thank you for pointing it out.

Comment 4:  “4. Sensor types”:Self-powered triboelectric type and mixed potential type should not be ignored. As a review paper, it is recommended to provide relevant parameter definitions for different gas sensors, referring to ref. 157.Couldn’t find any paper on mixed potential WEARABLE gas sensors.

Response: Thank you for your valuable feedback. In accordance with your suggestion, we have included a section specifically dedicated to Self-powered triboelectric wearable gas sensors (Section 4.6). Mixed potential sensors were briefly discussed for completeness in section 4.3. We acknowledge the lack of literature on mixed potential WEARABLE gas sensors. We have highlighted this in the relevant section of the paper by including this sentence, which issupported by relevant recent papers as well:

“Primarily portable and benchtop mixed potential sensors have been demonstrated. Investigation towards wearable and room-temperature realizations of such gas sensing devices could be a promising research direction.”

When discussing the various papers we review in our manuscript, we discuss and present the various gas performance metrics such as selectivity, sensitivity, response/recovery time where possible. Some of these have also been used in the table of the paper. At the end of the introduction we have included the following sentence to provide an introduction to the reader in terms of how we will be discussing, comparing and presenting various sensors reviewed in the paper, as per your request:

“In the following sections, various materials, device architectures and sensing approaches are reviewed and are discussed and presented with regards to gas sensing parameters that mainly include sensor response, sensitivity, response/recovery times, selectivity, de-tection limit, and stability.”

Comment 5: “4.2. Optical”: It is recommended to delete this section as it is difficult to comply with wearability.

Response: We appreciate the concern regarding wearability. While optical-based gas sensors might initially seem challenging in terms of wearability, they have been highlighted in this section due to their distinct advantages. These include fast response times for immediate real-time detection and minimal drift because they rely on the absorption of gas molecules at specific wavelengths. Their inherent specificity and sensitivity to gases also mean that with proper design, there's reduced potential for cross-response to other gases, ensuring reliability. We believe that by emphasizing the potential for wearability improvements and the unparalleled benefits they provide, it underscores their relevance in the context of this review potentially stimulating further research towards new materials and devices in an effort to enhance the wearability of this type of sensors. The main challenges are related to the overall system size, its flexibility/stretchability as well as the power consumption of such approaches. We have further emphasize this in the paper. To highlight this we have added the following paragraph at the end of the section:

“Generally speaking, optical approaches are not the most suitable sensing techniques for wearable applications, due to their overall size, especially when considering a complete system, their power consumption and aspects related to wearability (flexibility, stretchability, mechanical robustness, etc.). However, they are discussed herein for completeness and in an effort to stimulate future research into the wearability of such sensing approaches.”

Comment 6: Table 1: The table content is not displayed completely.

Response: Apologies for the oversight. This was probably done by the journal’s editing team, when moving our manuscript to the journal’s template. We have corrected the issue, and Table 1 now displays its content completely. We will ensure this is not an issue in the final manuscript.

Comment 7:  “7. Conclusions and outlook” need to be strengthened further. The influence of humidity needs to be discussed, and may refer to Small 2023, 19, 2303631. High-performance metal oxide-based room temperature gas sensor. Temperature and humidity compensation algorithm, etc. In addition, readers would like to see more constructive strategies. For example, principles of material selection, materials and device manufacturing processes, humidity resistance and application prospects.

Response: Thank you for your comment. Issues related to relative humidity as well as operational temperature have been discussed throughout the paper, as they are critical. However indeed, they were not highlighted in section 7. These are discussed now in summary to address the issue raised. The suggested recent Small paper has been added in relevant discussions in section 7, as per your request. Different manufacturing approaches are now presented and separated as standard microfabrication approaches and additive manufacturing techniques, and several recent (2023) examples have been cited. Recent TENG and biofuel cell reviews have been added to point the reader towards self-power implementations. We have added the following text:

“In terms of manufacturing technologies, for high resolution, miniaturization, throughput and quality of end-product standard microfabrication techniques should be preferred. However, current advanced ion additive manufacturing approaches should not be over-looked. While often inferior in terms of manufacturing resolution, they come with certain advantages. These include lower cost of fabrication and manufacturing instruments and no need for a clean room. Techniques such as inkjet, screen, stencil and extrusion-based printing, laser engraving and carbonization, stamping and transfer printing, their varia-tions and their combination can be exploited to create devices, while they can also be combined if needed with microfabrication approaches for critical stages of the manufac-turing process. Each of these methods come with their constraints In terms of material properties most suitable to be used with, manufacturing temperatures, final device di-mensions etc., so materials need to be carefully selected and optimized. As a result man-ufacturing different system parts with different techniques might be required, leading to the aforementioned requirement for heterogeneous integration. In terms of materials MOS materials, carbon nanomaterials and catalysts play an important role in gas sensors and their future. Selection of these and their combination need careful thinking and needs to be tailored to the specific application. The final form factor and device flexibility and stretchability are of prime importance for wearable applications, as well as power con-sumption and mechanical robustness. Self powered, low-power consumption system should thus be the focus in the next generation of wearable gas sensors and TENG tech-nologies as well as sweat biofuel cells are promising roots towards that direction. For deployment directly worn on the body, skin adhesion should be considered. In addition, sensor lifetime and reusability, especially following long wearable use and repeated cy-cles of removing and reattaching on the body will prove critical. For devices meant for clothe integration, the possibility of reversible attachment to the clothes could be a solu-tion to avoid damage during washing. Crucial aspects to gas sensor performance and wearability are also related to the operational temperature of the sensor, its response and recovery time, and its sensitivity to relative humidity. The effect of humidity to gas sensor response has been highlighted throughout the paper. One approach to evaluate and properly interpret sensor response could be the co-integration of humidity sensors. Al-ternatively, device, material and machine learning approaches can be explored. Room temperature operated MOS sensors with remarkable performance are emerging. As in several technological domains, the recent explosion in the development, use and pertur-bation of machine learning approaches, such techniques can immensely impact the area of gas sensors. Several papers were machine learning approaches are used to aid gas sensors have been recently published.”

Comment 8: References list: The literature on this subject is not complete, and it is recommended to further search the literature. In addition, it may be necessary to consider setting certain restrictions on the publication time of the cited literature, such as focusing on the past 5 years.

.

Response: Thank you for your insightful comments. In response to the feedback, the scope of the literature review has been refined to emphasize developments from the past five years (from 2018 onwards) in the realm of smart wearable (nano)sensors for environmental monitoring of gaseous pollutants. We acknowledge that the field is rapidly evolving, and while we have made comprehensive efforts to include the most recent and relevant studies, it's worth noting that the literature is extensive. Thus, readers are encouraged to explore the subject further for a more exhaustive understanding. In response to several reviewer comments, we have added several recent papers.

Comment 9: Check the reference format of the journal, such as providing complete titles.

Response: Thank you for bringing this to our attention. We have carefully checked and updated the reference format in accordance with the journal's guidelines, ensuring that complete titles are provided for each reference. We have also cross-verified the references to ensure their accuracy and adherence to the specified format.

Round 2

Reviewer 1 Report

Accept in present form

Reviewer 2 Report

The authors answered some questions satisfactorily; despite the work still needs improvement, I recommend the publication.

English language better.

Reviewer 3 Report

the authors corrected some inaccuracies, the paper became better

Reviewer 4 Report

The response and revised manuscript are satisfactory, and it is recommended to accept.